# Information-Driven Design of Imaging Systems

**Henry Pinkard**          **Leyla Kabuli**          **Eric Markley**

**Tiffany Chien**          **Jiantao Jiao**          **Laura Waller**

Department of Electrical Engineering and Computer Sciences, University of California, Berkeley

## Abstract

Imaging systems have traditionally been designed to mimic the human eye and produce visually interpretable measurements. Modern imaging systems, however, process raw measurements computationally before or instead of human viewing. As a result, the information content of raw measurements matters more than their visual interpretability. Despite the importance of measurement information content, current approaches for evaluating imaging system performance do not quantify it: they instead either use alternative metrics that assess specific aspects of measurement quality or assess measurements indirectly with performance on secondary tasks. We developed the theoretical foundations and a practical method to directly quantify mutual information between noisy measurements and unknown objects. By fitting probabilistic models to measurements and their noise characteristics, our method estimates information by upper bounding its true value. By applying gradient-based optimization to these estimates, we also developed a technique for designing imaging systems called Information-Driven Encoder Analysis Learning (IDEAL). Our information estimates accurately captured system performance differences across four imaging domains (color photography, radio astronomy, lensless imaging, and microscopy). Systems designed with IDEAL matched the performance of those designed with end-to-end optimization, the prevailing approach that jointly optimizes hardware and image processing algorithms. These results establish mutual information as a universal performance metric for imaging systems that enables both computationally efficient design optimization and evaluation in real-world conditions.

A video summary of this work can be found at:

`https://waller-lab.github.io/EncodingInformationWebsite/`

## 1   Introduction

The increasing utilization of computational processing in imaging systems removes the constraint that they must produce human-interpretable measurements. Traditional imaging system designs produce measurements that contain the specific visual patterns recognizable to human observers [1]. However, these patterns are unnecessary for modern algorithms like deep neural networks, which can utilize information in measurements regardless of how it appears: they can either transform measurements into forms that make sense to humans, as computational imaging systems do [2], or they can analyze measurements directly and bypass humans altogether.

This shift fundamentally changes what limits performance: not *how well* systems encode information into human-perceptible patterns, but *how much* information they encode, regardless of its visual appearance.

39th Conference on Neural Information Processing Systems (NeurIPS 2025).

Existing metrics to quantify hardware performance fail to capture information content comprehensively. Traditional measurement metrics like resolution, signal-to-noise ratio, sampling rate, and field-of-view characterize individual system aspects separately, making it difficult to assess their combined effect on overall performance or compare systems that make different trade-offs among these factors.

As a result, performance is often assessed by computationally decoding measurements to perform secondary tasks. This decoder-based approach uses tasks like image reconstruction or object classification. It quantifies success by comparing algorithm outputs against ground truth using metrics like mean squared error or classification accuracy [2–4]. This approach furthermore enables system design in simulation through "end-to-end" methods that jointly optimize both encoder parameters (like lens shapes or sensor configurations) and decoder algorithms (like image reconstruction). [5–11].

However, decoder-based evaluation and end-to-end optimization have a few limitations. First, decoder-based evaluation metrics require full ground truth knowledge of the object or some of its properties, which limits their application to simulations and laboratory settings that may not generalize to real-world conditions. Second, this joint evaluation conflates encoder and decoder performance, since low performance could result from either insufficient encoded information or the decoder's failure to utilize it. Finally, neural network-based decoders require substantial compute and may create backpropagation challenges in end-to-end optimization [11, 12].

Directly quantifying information in measurements provides an alternative approach that avoids the limitations of decoders. Mutual information quantifies how much object information survives the encoding and measurement processes [13–37, 3]. Equivalently, it quantifies how much a measurement reduces an observer's uncertainty about the object.

Mutual information also unifies traditionally separate aspects of measurement quality, enabling meaningful comparison between systems that make different trade-offs among these factors (**Sec. S1.2**). For example, it captures the combined effect of resolution, signal-to-noise ratio, sampling, and quantization on information preservation [17, 25, 38].

Despite these benefits, practical estimation of mutual information has been challenging. Previous approaches fall into two categories, each with significant limitations.

The first approach overestimates information because it models an imaging system as an unconstrained communication system [13, 15–18, 23, 28, 37]. While communication systems assume encoders can transform any input into any output without physical constraints, imaging systems cannot achieve this because lenses, sensors, and other hardware impose fundamental limitations. These theoretical calculations can therefore produce highly inaccurate estimates (**Fig. S6**).

The second approach explicitly models the objects being imaged, relying on assumptions about object properties that are difficult to verify in practice [30–35]. These object models must also be developed case-by-case for each object type, limiting their generality.

To address these limitations, we developed a framework for estimating mutual information directly from measurement of unknown objects. Our approach requires only two inputs: a dataset of measurements and a model of the system's noise. By fitting models to actual measurements rather than assuming object properties or ignoring encoders' physical constraints, the method readily adapts to diverse imaging modalities and performance trade-offs. Our framework enables computationally efficient optimization when designing systems through simulation, as well as quantitative evaluation of real imaging systems operating in the field.

This paper is organized as follows: **Section 2** presents the theoretical framework for estimating mutual information from measurements. **Section 3** describes implementation through three probabilistic models and validates their accuracy. **Section 4** demonstrates that information estimates predict system performance across four imaging applications: color photography, radio astronomy, lensless imaging, and microscopy. **Section 5** introduces our method for automated imaging system design through gradient-based optimization, Information-Driven Encoder Analysis Learning (IDEAL). **Section 6** considers implications and future research directions.

## 2 Mathematical framework

This section shows how to overcome the inherent statistical and computational challenges of mutual information estimation by progressively decomposing the estimation problem using key properties of imaging systems. First, deterministic encoding makes it possible to focus on the relationship between clean images and noisy measurements rather than the more complex object-measurement relationship. Second, the known statistical structure of noise in imaging systems enables decomposing the estimation problem into two simpler subproblems. Finally, the constraint of only having measurement data (on real imaging systems) informs the selection of estimation approaches for each subproblem.

To formalize this approach, we model imaging systems as a chain of probability distributions (**Fig. 1**). Consider photographing trees: a probability distribution over *objects* captures how trees vary in size, shape, and species. An *encoder* (a camera with particular lens and position) maps each possible tree to a noiseless *image* on the sensor. Detection noise corrupts these ideal images, creating the *measurements* that sensors actually record [13, 22, 30, 39, 40] (**Sec. S1**).

Mutual information provides a metric for comparing encoder designs by quantifying how well measurements can distinguish different objects. Information losses occur at two stages: during encoding (e.g. from finite resolution or sampling) and during measurement (from noise).

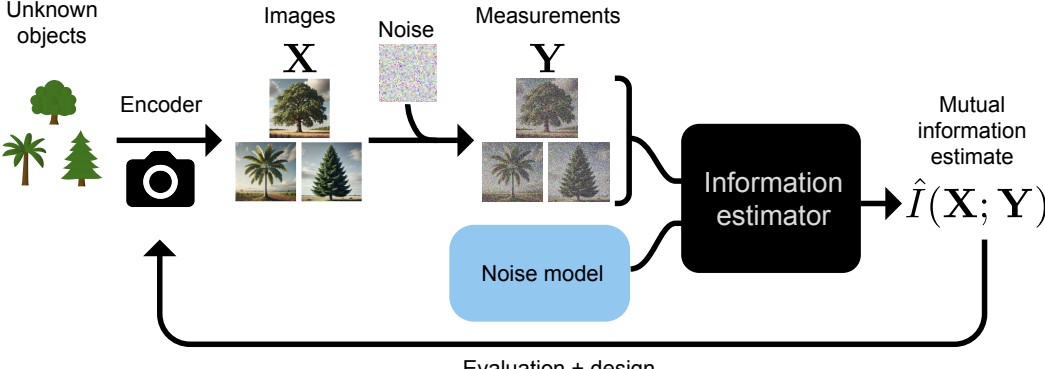

Figure 1: **Information estimation for imaging systems.** An encoder (e.g., an optical system) maps unknown objects to noiseless images $\mathbf{X}$. Noise corrupts these images to produce measurements $\mathbf{Y}$. The information estimator uses these measurements and a noise model to quantify how well measurements distinguish noiseless images (and thus objects under deterministic encoding). The estimate $\hat{I}(\mathbf{X};\mathbf{Y})$ enables both encoder evaluation and design optimization.

Mathematically, we model images and measurements as probability distributions over vectors. An image with $D$ pixels is a vector $\mathbf{x} = (x_1, x_2, \ldots, x_D)$, where $x_k$ represents the energy at the $k$th pixel. The probability distribution $p(\mathbf{x}) = p(x_1, x_2, \ldots, x_D)$ describes the likelihood of each possible image. Similarly, a noisy measurement is a vector $\mathbf{y} = (y_1, y_2, \ldots, y_D)$ with distribution $p(\mathbf{y})$.

Quantifying object information in measurements becomes tractable through the widely-used assumption of deterministic imaging systems. The object-measurement mutual information $I(\mathbf{O};\mathbf{Y})$ requires working with the joint distribution of objects and measurements, which is unknown in real systems. However, deterministic encoders, which always produce the same noiseless image for each object, enable estimation of $I(\mathbf{X};\mathbf{Y})$ instead. This deterministic encoder assumption applies broadly in practice and holds by construction in simulated design.

Intuitively, this equality holds because all variations in noiseless images come from variations in objects when encoders are deterministic. Thus, any image information that survives the measurement process must have also come from the object. Mathematically:

**Deterministic Information Transfer Theorem.** For a Markov chain $\mathbf{O} \to \mathbf{X} \to \mathbf{Y}$, if $\mathbf{X} = f(\mathbf{O})$ for some deterministic function $f$, then $I(\mathbf{O};\mathbf{Y}) = I(\mathbf{X};\mathbf{Y})$.

(See supplement for proof (**Sec. S1.1**)).

Even the more accessible $I(\mathbf{X};\mathbf{Y})$ estimation problem requires handling the high-dimensional joint distribution $p(\mathbf{x}, \mathbf{y})$. Estimating mutual information between high-dimensional variables from

samples is notoriously challenging [41–50]. Without strong assumptions, estimation requires sample sizes that grow exponentially with dimensionality, suffers from high bias and variance, and lacks formal guarantees [51, 46].

Furthermore, the majority of existing estimation methods assume access to a dataset of paired samples $\{\mathbf{x}^{(i)}, \mathbf{y}^{(i)}\}_1^N$. While these are available in simulated systems, real systems only provide noisy measurements $\{\mathbf{y}^{(i)}\}_1^N$ without corresponding noiseless images.

Fortunately, imaging systems have well-characterized noise processes $p(\mathbf{y} \mid \mathbf{x})$ from physical models or calibration [40]. This knowledge, combined with the deterministic encoder property, enables decomposing $I(\mathbf{X}; \mathbf{Y})$ into entropy terms that can be estimated from noisy measurements alone:

$$I(\mathbf{X}; \mathbf{Y}) = H(\mathbf{Y}) - H(\mathbf{Y}|\mathbf{X}) \tag{1}$$

$H(\mathbf{Y})$ quantifies total variation in measurements from both objects and noise. $H(\mathbf{Y}|\mathbf{X})$ quantifies variation from noise alone. Their difference $I(\mathbf{X}; \mathbf{Y})$ isolates noiseless image-induced variation in measurements, which equals the object information preserved when encoders are deterministic.

These entropy terms can be expressed as expectations:

$$H(\mathbf{Y}) = \mathbb{E}\left[-\log p(\mathbf{Y})\right] \tag{2}$$
$$H(\mathbf{Y}|\mathbf{X}) = \mathbb{E}\left[-\log p(\mathbf{Y} \mid \mathbf{X})\right] \tag{3}$$

The $H(\mathbf{Y}|\mathbf{X})$ term further simplifies under the assumption of independent noise across pixels, reducing the conditional entropy $H(\mathbf{Y}|\mathbf{X})$ from a high-dimensional integral to a sum of single-pixel entropies: $H(\mathbf{Y}|\mathbf{X}) = \sum_{i=1}^D H(Y_i|X_i)$ (**Sec. S2.3.1**). Imaging systems commonly satisfy this assumption [40].

The $H(\mathbf{Y})$ term requires learning from data. The true distribution $p(\mathbf{y})$ is unknown but its entropy $H(\mathbf{Y})$ can nonetheless be upper bounded by fitting a parametric model $p_\theta(\mathbf{y})$ to a training set of $N$ measurements $\{\mathbf{y}^{(i)}\}_1^N$ and computing the cross-entropy on a held-out test set of $M$ additional measurements:

$$H(\mathbf{Y}) \leq \mathbb{E}\left[-\log p_\theta(\mathbf{Y})\right] \approx -\frac{1}{M} \sum_{i=1}^M \log p_\theta(\mathbf{y}^{(i)}) \tag{4}$$

This upper-bounding approach reliably estimates entropy in complex real-world distributions without knowing the true probability [52].

Furthermore, cross-entropy provides a principled way to compare models. Since cross-entropy between true and estimated distributions always exceeds true entropy, models achieving lower cross-entropy values produce more accurate estimates.

Subtracting our estimate of $H(\mathbf{Y}|\mathbf{X})$ from this upper bound on $H(\mathbf{Y})$ yields our estimate of the mutual information $\hat{I}(\mathbf{X}; \mathbf{Y})$.

## 3  Probabilistic models for information estimation

Practical implementation of our framework requires choosing probabilistic models to fit both measurements and noise processes. For measurements, users face a fundamental trade-off: expressive models capture complex distributions accurately but demand substantial data and compute, while simpler models train faster using fewer resources but may overestimate information. To address this trade-off, we developed and tested three models spanning the expressivity-efficiency spectrum: a stationary Gaussian process, a full Gaussian process, and an autoregressive PixelCNN model [53–55]. For noise modeling, users must select an appropriate model and (on real systems) fit its parameters using only noisy measurements.

This section details the three measurement models and validates their performance through experiments that establish practical guidance for model selection in different scenarios.

**Measurement model assumptions.** Our models differ in two key assumptions: whether measurements follow a multivariate Gaussian distribution and whether pixel statistics are translation-invariant (stationary). Violated assumptions yield worse fits that overestimate information. For example, Gaussian models cannot capture many types of statistical structure that are present in real measurement distributions, such as the bimodal statistics of measurements of MNIST handwritten digits (**Fig. S15**). Meanwhile, stationary models fail with position-dependent artifacts like sensor defects or illumination gradients.

We processed images as patches rather than full images for computational efficiency. Larger patches capture more spatial relationships (**Fig. S12**) but require more computation; smaller patches provide more training samples and permit smaller models.

**Model types.** We tested three models with different expressivity-efficiency trade-offs.

Table 1: **Comparison of probabilistic models for information estimation.** Training times measured on $20 \times 20$ patches using an NVIDIA RTX A6000 GPU. Stationary Gaussian training time range reflects optional optimization for numerical stability.

| Model | Data Efficiency | Training Time | Expressivity | Gaussian | Stationary |
|-------|-----------------|---------------|--------------|----------|------------|
| Stationary Gaussian | Highest | $\sim$0.1–10s | Lowest | ✓ | ✓ |
| Gaussian (full cov.) | Medium | $\sim$0.1s | Medium | ✓ | |
| PixelCNN | Lowest | $\sim$100s | Highest | | |

**Estimator consistency.** We validated convergence behavior on both simulated data with known ground truth and real imaging data. On simulated data from a stationary Gaussian process with additive noise, all three models converged to the analytically-calculated mutual information, with the stationary Gaussian converging fastest since the test data matched its assumptions (**Fig. S16a**). On real imaging data, all models converged to stable, though different, values. PixelCNN consistently achieved the lowest entropy bounds, indicating the most accurate estimates, while Gaussian models plateaued at higher values due to their inability to capture non-Gaussian statistics (**Fig. S16b,c**). Based on these convergence results, we used $\sim$1,000–10,000 patches of $\sim 20 \times 20$ pixels for subsequent experiments, balancing computational efficiency with estimation accuracy.

**Model selection.** Based on our validation results, we recommend selecting models according to available resources and data characteristics. The stationary Gaussian provides reliable estimates with limited data, PixelCNN maximizes accuracy when computational resources and data are abundant, and the standard Gaussian balances ease of computation with reliable performance. While PixelCNN generally provides the most accurate estimates, the performance difference may be smaller for data with approximately Gaussian statistics, making simpler Gaussian models reasonable when computational resources are limited. Fitting multiple models and selecting the lowest entropy estimate yields the most reliable results when feasible.

**Noise model validation.** Noise processes in imaging systems proved far simpler to model than measurement distributions, making $H(\mathbf{Y}|\mathbf{X})$ estimation straightforward. While this required identifying the correct noise model and fitting its parameters, the structure of imaging noise allows the framework to remain robust to somewhat inaccurate noise modeling. Object-independent noise (thermal, dark current) affects all measurements equally, so any model error simply shifts all estimates by the same constant, preserving encoder rankings. Object-dependent noise (shot noise) arises from well-understood quantum processes with established mathematical models, facilitating accurate characterization.

We tested one key challenge specific to shot noise and confirmed our framework handles it successfully: shot noise parameters depend on the noiseless image that real systems cannot observe. We validated that accurate entropy estimation remained possible using only noisy measurements. Our estimators succeeded across a wide range of photon counts, deviating from the true value only below $\sim$20 photons per pixel (**Fig. S18**).

# 4 Validation across imaging applications

Theoretical results establish that mutual information limits achievable decoder performance [56–59], but *estimates* of information inevitably contain error. If information estimates accurately capture what limits real decoder performance, they should correlate with decoder-based evaluation across diverse imaging systems. We tested this relationship by comparing information estimates against decoder performance across four imaging systems spanning different modalities, noise regimes, and tasks, with decoders ranging from classical signal processing to modern neural networks.

Across all four systems, information estimates predicted decoder performance, validating their use for additional settings where decoders face computational challenges or cannot be used because ground truth data is unavailable.

**Color photography.** Digital cameras encode color on monochrome sensors using color filter arrays in front of their pixels. These arrays filter incoming light so that each pixel (traditionally) detects only red, green, or blue light, and demosaicing algorithms decode the raw measurements to reconstruct full-color images at the full sensor resolution. Beyond the traditional Bayer pattern (a $2\times2$ grid with two green, one red, and one blue pixel) (**Fig. 2a, bottom**), recent end-to-end learned designs incorporate white (i.e. no color filter) pixels together with neural-network-based decoders [60, 61] to improve reconstruction quality.

Our information estimates predicted reconstruction quality across three color filter designs, enabling the evaluation of designs without reconstruction algorithms or ground truth data (**Fig. 2a**). We tested the traditional Bayer pattern, a random arrangement of red, green, blue, and white filters, and a learned arrangement of these filters [61]. Using natural images [62, 63] with simulated photon shot noise, we estimated information content for each design and reconstructed full-color images using neural network demosaicing [61]. Higher mutual information consistently predicted better reconstruction across all quality metrics—mean squared error (MSE), peak signal-to-noise ratio (PSNR), and structural similarity index measure (SSIM) (**Fig. S20**).

**Black hole imaging.** Radio telescope arrays can achieve the angular resolution of an Earth-sized telescope by combining data from global sites, enabling images like the Event Horizon Telescope's M87 black hole image [64]. Selecting optimal telescope locations for next-generation arrays remains computationally challenging: each site's value depends on all others, and reconstructing images for every possible configuration is computationally intractable [65]. Thus, current approaches use simplified metrics that can miss important imaging features [65].

Our information estimates predicted reconstruction quality across telescope array configurations, enabling telescope site selection without computationally expensive image reconstruction (**Fig. 2b**). Using four-telescope subsets from the original Event Horizon array, we simulated measurements with additive Gaussian noise and compared information estimates to the error in black hole images generated by an iterative inverse problem optimization. Higher information estimates consistently produced more accurate reconstructions (**Fig. S20**).

**Lensless imaging.** Lensless imagers replace traditional lenses with light-modulating masks, offering simple hardware, wide field-of-view, and the ability to encode depth and time information in single measurements [66]. Their encoded measurements bear no visual resemblance to scenes and vary dramatically between designs, requiring computational reconstruction for both image recovery and performance evaluation.

Our information estimates predicted reconstruction quality across three optical designs, enabling evaluation of unconventional optics without image reconstruction (**Fig. 2c**). We tested a traditional lens, random microlens array [67], and Gaussian diffuser [68] using natural images with simulated photon shot noise at various light levels. Higher information estimates consistently correlated with better reconstruction accuracy across all designs and noise conditions (**Fig. S20**).

**Coded illumination microscopy.** LED array microscopy replaces standard illumination with programmable light sources to flexibly generate different contrast modes at low cost by varying angle, intensity, and coherence [69, 70]. Optimal illumination patterns depend heavily on imaging tasks, making theoretical evaluation difficult and necessitating empirical comparison through task-specific decoders [71, 72].

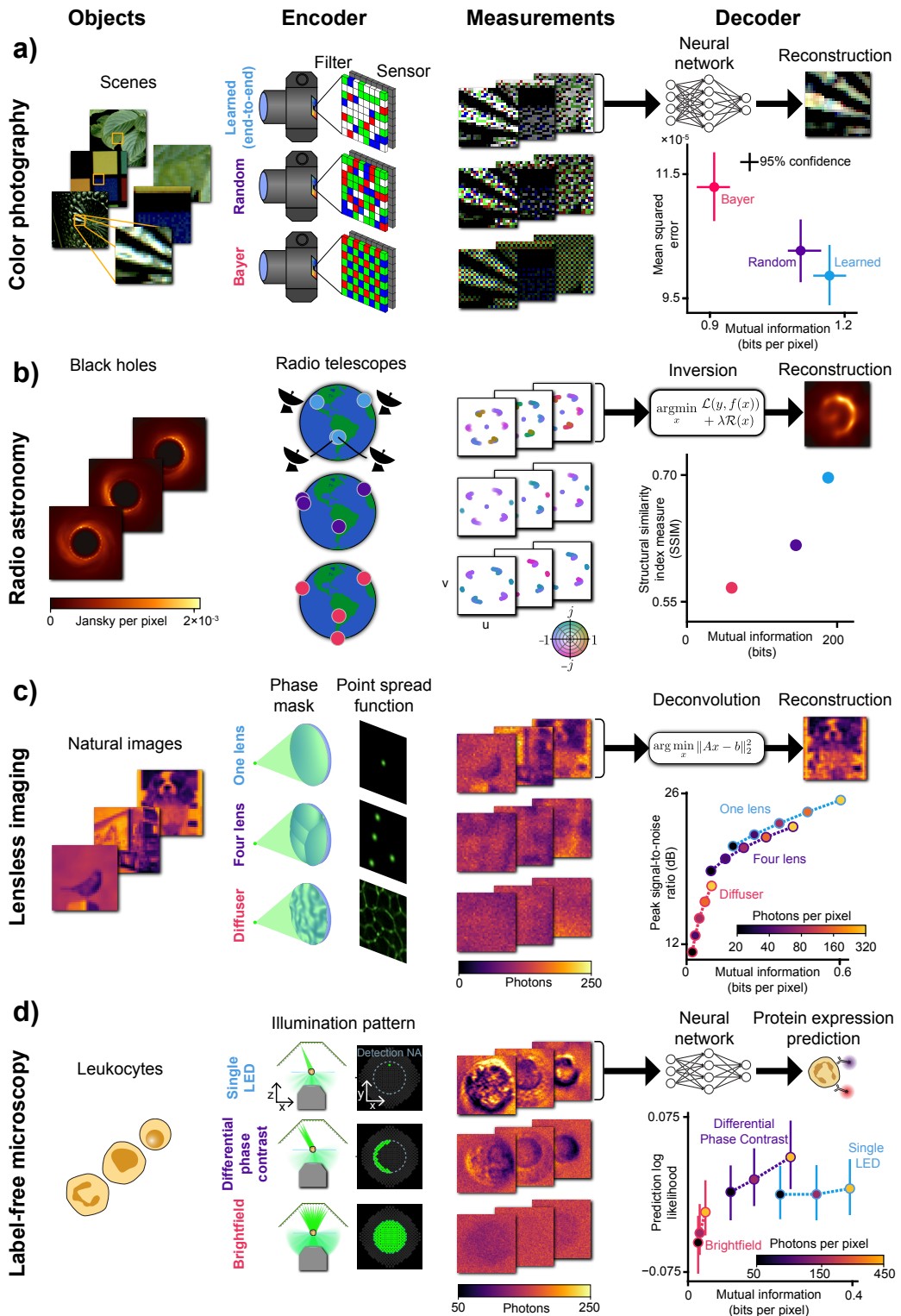

Figure 2: **Information estimates predict decoder performance across four imaging applications.** Each row shows representative objects, three encoder designs, example measurements, and the relationship between information estimates and a representative decoder performance metric (see **Fig. S20** for all metrics). **a)** Color photography: Bayer, learned, and random filter arrays with neural network demosaicing [61]. **b)** Radio astronomy: Three telescope array configurations with inverse problem reconstruction. **c)** Lensless imaging: Lens, microlens array, and diffuser with Wiener deconvolution. **d)** LED array microscopy: Brightfield, differential phase contrast, and single-LED illumination with neural network protein expression prediction.

Our information estimates correlated with protein prediction accuracy across three illumination patterns, enabling evaluation of microscopy designs without time-consuming and expensive protein labeling experiments (**Fig. 2d**). We tested brightfield, differential phase contrast [73, 74], and single-LED illumination measurements of white blood cells [75] with added simulated photon shot noise to equalize photon counts. Higher information estimates generally predicted better neural network classification performance, though single-LED showed inflated estimates due to optical imperfections (**Sec. S4.1**).

### 4.1 Task-specific information

While information estimates strongly predict reconstruction performance, they may correlate less strongly with specialized tasks that rely only on specific features of measurements. Systems that preserve specific task-relevant information can outperform those capturing more total information indiscriminately [31, 76].

Our experiments confirmed that specialized tasks show weaker correlations between information estimates and decoder performance. We tested this by switching from reconstruction to a classification task for the lensless imaging system (**Fig. S22**). This classification task uses only $\sim$1% of the information in measurements (**Sec. S4.2**). We found that designs with similar total information achieved different classification accuracies, likely reflecting different amounts of task-specific information captured (**Fig. S20**). A similar pattern may explain the microscopy results (**Fig. 2d**), where single-LED illumination yielded less accurate protein predictions than differential phase contrast despite higher information estimates.

These findings motivate extensions that optimize task-specific rather than total information. By decomposing information into task-specific and task-irrelevant components ($I(\mathbf{Y}; \mathbf{O}) = I(\mathbf{Y}; \mathbf{T}) + I(\mathbf{Y}; \mathbf{O}|\mathbf{T})$), future work could target specialized tasks while maintaining decoder-independent evaluation (**Sec. S2.8**).

## 5 Encoder design via information maximization with IDEAL

Having validated that information estimates can assess encoder quality directly from measurements, we tested their ability to guide computationally efficient design optimization. To realize this capability, we developed Information-Driven Encoder Analysis Learning (IDEAL), which iteratively improves encoder designs through gradient ascent on information estimates (**Fig. 3a**). IDEAL requires only a differentiable encoder model and a dataset or model representing the objects to be imaged, avoiding the memory and compute requirements and backpropagation challenges of complex neural network decoders.

IDEAL successfully optimized color filter arrays, creating encoders that capture more information and enable better reconstruction. Testing on photography filters with red, green, blue and white pixels, IDEAL progressively improved filter designs compared to a random initial filter (**Fig. 3b**). To validate the optimization, we trained decoders on measurements from designs at different optimization stages, and found that reconstruction accuracy increased from beginning to end (**Fig. 3c**). Furthermore, our PixelCNN estimator independently confirmed that information content increased throughout optimization, validating that the faster Gaussian model used as our objective successfully guided the design process. IDEAL achieved comparable information content and decoder error to end-to-end optimization [61], which jointly trains both encoders and decoders. This encoder-only approach offers advantages in simplicity, memory, and runtime, which are studied in greater depth in subsequent work [77].

## 6 Discussion

In this work, we developed a broadly applicable and computationally efficient method that comprehensively captures imaging system performance by quantifying the information content of measurements. The method applies broadly because it uses only measurement data and noise characterization, without requiring any knowledge or assumptions about objects or system-specific mathematical models. It captures comprehensive performance because mutual information naturally synthesizes resolution, noise, sampling, and all other factors that affect the ability to distinguish different objects into a single

**a) Information-Driven Encoder Analysis Learning (IDEAL) for color filter design**

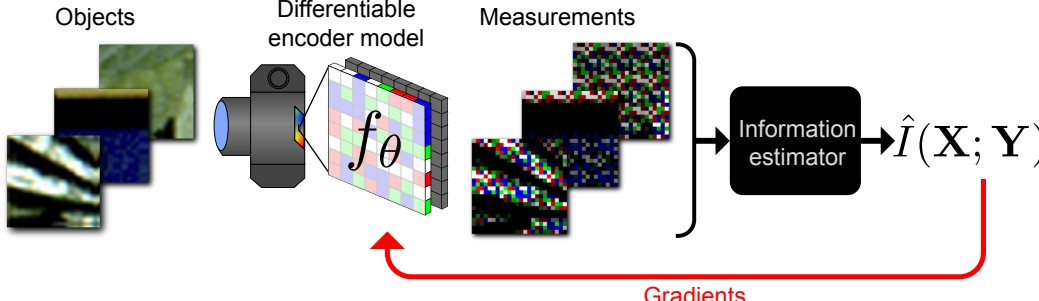

**b) Optimization process**

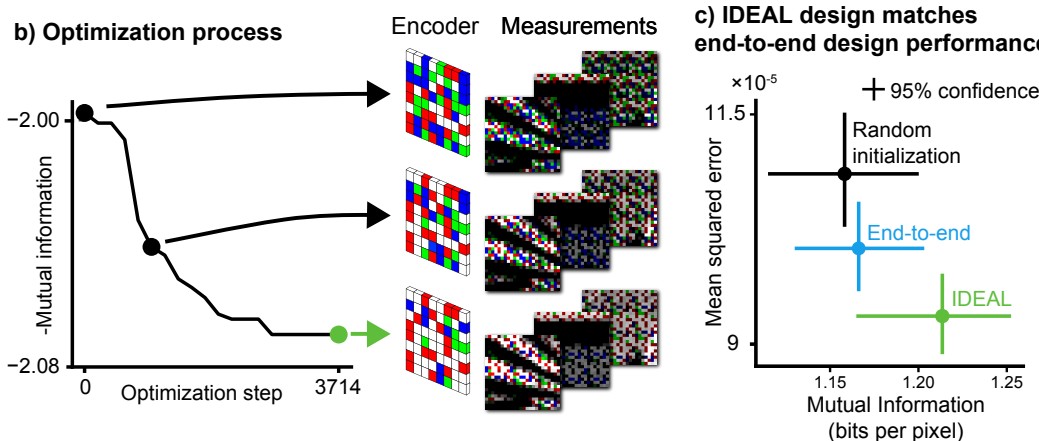

Figure 3: **Information-Driven Encoder Analysis Learning (IDEAL) designs encoders through gradient ascent on information estimates. a)** IDEAL framework applied to color filter design: gradient feedback from information estimates updates parameters of a differentiable filter model to maximize information capture. **b)** Estimated information increases monotonically during optimization. **c)** The final IDEAL-optimized encoder matches the performance of jointly optimizing encoders and decoders end-to-end [61] in terms of both downstream reconstruction error and measurement mutual information, while avoiding decoder complexity during training. We further study IDEAL in comparison to end-to-end in subsequent work [77].

metric. It achieves computational efficiency by evaluating encoders directly, bypassing the traditional approach of using decoder algorithms to perform secondary tasks and then measuring success on those tasks.

We validated that our approach realizes these advantages while matching traditional decoder-based evaluation and design strategies across color photography, radio astronomy, lensless imaging, and microscopy. Information estimates consistently predicted which designs would enable decoders to reconstruct unknown objects most accurately. Additionally, we demonstrated that information estimates can guide automated design of imaging systems using IDEAL (Information-Driven Encoder Analysis Learning), achieving comparable performance to the prevailing approach of end-to-end optimization through an encoder and decoder simultaneously.

Information-based evaluation offers computational advantages that suggest promising capabilities for designing previously intractable imaging systems in simulation. By avoiding decoders, the approach eliminates the memory requirements of large neural networks and the training complexity of joint optimization, such as vanishing gradients that arise during backpropagation through complex architectures [11, 12]. This reduced computational burden enables more effective exploration of design spaces and may allow for tackling new problems where end-to-end design becomes computationally intractable. We explore IDEAL's capabilities and limitations more extensively across diverse imaging systems in subsequent work [77].

Beyond simulated design, our method creates entirely new possibilities for rigorously evaluating imaging systems operating in real-world conditions. Current evaluation approaches each have significant limitations: subjective visual assessment of raw measurements or algorithm outputs suffers from observer bias and inconsistency; heuristic quality metrics applied to algorithm outputs are unreliable for powerful neural network decoders that can hallucinate convincing but incorrect details [78]; assessments like signal-to-noise ratio capture only isolated system aspects rather than overall capability; and standardized test objects like resolution targets may not represent the diversity of real-world objects. Our information-based approach addresses these limitations by providing an objective, unified metric that quantifies measurement quality without requiring knowledge or assumptions about the objects being imaged. Our approach thus enables rigorous and comprehensive evaluation of imaging systems capturing unknown objects in their intended environments.

More broadly, our method may aid in discovering signal in measurements that appear unintelligible to human observers. Information theory suggests that optimal encoders produce measurements that appear random or noisy to maximize entropy and ensure small object changes remain distinguishable [79, 80]. Such measurements would naturally appear unintuitive to human observers who are selectively sensitive to specific visual features [1]. Laser speckle patterns hint at this possibility: once routinely suppressed as unwanted noise[40], they can now be effectively interpreted by neural networks[81–83].

### Extensions and future directions

Beyond our framework's primary applications to the evaluation and design of imaging systems, several promising directions could improve its capabilities, extend its theoretical foundations, and enable its application to new classes of design problems.

**Performance improvements.** Better probabilistic models could improve the estimator's accuracy. Transformer architectures [84] exhibit predictable scaling laws [52, 85], enabling either deployment of larger models for more accurate estimates or extrapolation of true information content using scaling relationships on smaller models. Alternatively, specialized architectures might achieve equivalent accuracy with fewer computational resources.

**Stochastic encoders.** Extending the framework to handle non-deterministic imaging systems would broaden its applicability. Two scenarios require this extension: systems with unpredictable variations like illumination fluctuations or mechanical drift, and quantum imaging regimes where image formation is inherently random [86–88]. We outline key challenges for this extension in the supplement (**Sec. S2.7**), though practical validation remains future work.

**Task-specific imaging.** Extending the framework to specialized tasks would address applications where maximizing total information may be suboptimal. While our approach captures performance well on reconstruction tasks that require recovering objects in complete detail, information estimates less accurately characterize performance when decoders perform tasks such as classification that prioritize specific object features. Modifying the objective function our estimator (**Sec. S2.8**) might address this limitation. Alternatively, optimization for general performance with IDEAL could provide a starting point for task-specific refinement using end-to-end design methods [12] that reduces labeled data requirements.

**Design in non-differentiable systems.** Information estimation could enable optimization of non-differentiable imaging systems where neither end-to-end design nor IDEAL would apply. This includes combinatorial design problems in simulation and real-time parameter adjustments in deployed imaging systems. Both applications require efficient objective functions that can be evaluated repeatedly during gradient-free optimization. Subsequent work has applied information estimation to non-differentiable imaging system design [89].

**Other sensing modalities.** Information-theoretic design principles could extend beyond imaging to optimize performance in electronic, biological, geological, and chemical sensors. Our method could potentially apply to any sensing system that can be modeled as a deterministic encoding with a known noise model, suggesting possibilities for systematic sensor design across diverse domains.

## Project website

https://waller-lab.github.io/EncodingInformationWebsite/

## Code

https://github.com/Waller-Lab/EncodingInformation.

## Acknowledgements

For helpful feedback and discussions about this work we thank T. Courtade, E. Aras, K. Lee, K. Bouman, A. Gao, M. Foxxe, C. Degher, S. Degher, D. Degher, S. Baker, J. Goodman, A. Ashok and the UC Berkeley Computational Imaging Lab.

## Author Contributions

**Conceptualization**: H.P.
**Methodology**: H.P., L.K., and J.J.
**Data, Experiments and Software**: L.K., H.P., and E.M.
**Funding**: L.W.
**Supervision**: H.P. and L.W.
**Visualization and figures**: H.P., T.C., L.K., E.M., and L.W.
**Writing**: The original draft was written by H.P. with assistance from T.C. and L.K.; all authors contributed to review and editing.

## Funding

L.W. is a Chan Zuckerberg Biohub investigator. L.K. was supported by the National Science Foundation Graduate Research Fellowship Program under Grant DGE 2146752.
This work was supported by STROBE: A National Science Foundation Science and Technology Center under Grant No. DMR 1548924, ONR Grant N00014-17-1-2401, and the U.S. Air Force Office Multidisciplinary University Research Initiative (MURI) program under award no. FA9550-23-1-0281.

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

# Supplementary materials

# Contents

# S1 Information encoding formalism

Information theory enables quantification of uncertainty and randomness. A formal mathematical model of uncertainty and randomness requires probability. Thus in order to use information theory to analyze imaging systems, we must model them probabilistically.

This section assumes familiarity with probability and information theory fundamentals. A tutorial introduction to these concepts can be found in previous work [90], as well as the textbooks [91, 80] and Shannon's original paper [79].

## S1.1 Minimal probabilistic model of an imaging system

The minimal probabilistic model we present has been widely used in information-theoretic analyses of imaging systems [13, 22, 30, 39] and builds on established principles in statistical optics [40].

This minimal imaging system model involves three random variables (**Fig. S1**):

- The **object** has physical properties we want to measure
- The **noiseless image** is created when the imaging system encodes object properties through physical processes (e.g., electromagnetic fields in optical systems, pressure waves in ultrasound)
- The **noisy measurement** results from detection noise corrupting the noiseless image

Each variable is modeled as random to capture our uncertainty about its true value.

Information can be lost at two stages: During encoding, information may be irretrievably lost when creating the noiseless image (e.g., spatial frequencies beyond the system's passband). During detection, measurement noise further corrupts information present in the noiseless image. The goal of an imaging system is to preserve as much object information as possible through both encoding and detection.

Mathematically, these variables form a Markov chain with joint distribution:

$$p(\mathbf{o}, \mathbf{x}, \mathbf{y}) = p(\mathbf{o})p(\mathbf{x} \mid \mathbf{o})p(\mathbf{y} \mid \mathbf{x})$$

where:

- $p(\mathbf{o})$ is the distribution of possible objects
- $p(\mathbf{x} \mid \mathbf{o})$ is a mapping from objects to noiseless images
- $p(\mathbf{y} \mid \mathbf{x})$ represents the detection noise process

The relationship between object, noiseless image, and noisy measurement can be understood in two directions:

- Forward: Object → Noiseless Image → Noisy Measurement
  $p(\mathbf{o}, \mathbf{x}, \mathbf{y}) = p(\mathbf{o})p(\mathbf{x}|\mathbf{o})p(\mathbf{y}|\mathbf{x})$
- Reverse: Noisy Measurement → Noiseless Image → Object
  $p(\mathbf{o}, \mathbf{x}, \mathbf{y}) = p(\mathbf{y})p(\mathbf{x}|\mathbf{y})p(\mathbf{o}|\mathbf{x})$

The Data Processing Inequality [80] reveals key limitations in each direction:

- Forward: $I(\mathbf{O}; \mathbf{X}) \geq I(\mathbf{O}; \mathbf{Y})$ - noise can only reduce information about the object.
- Reverse: $I(\mathbf{Y}; \mathbf{X}) \geq I(\mathbf{Y}; \mathbf{O})$ - the measurement's ability to carry information about the object is limited by the information it carries about the noiseless image.

In our model, encoding is deterministic - each object maps to exactly one noiseless image ($p(\mathbf{x} \mid \mathbf{o})$ is a Dirac delta function). This means the noiseless image's randomness comes entirely from object uncertainty. Consequently, any information the measurement contains about the object must equal the information preserved through noise: $I(\mathbf{Y}; \mathbf{X}) = I(\mathbf{Y}; \mathbf{O})$.

**Deterministic Information Transfer Theorem.** For a Markov chain $\mathbf{O} \to \mathbf{X} \to \mathbf{Y}$, if $\mathbf{X} = f(\mathbf{O})$ for some deterministic function $f$, then $I(\mathbf{O}; \mathbf{Y}) = I(\mathbf{X}; \mathbf{Y})$.

*Proof.* Since $I(\mathbf{O}; \mathbf{Y}) = H(\mathbf{Y}) - H(\mathbf{Y}|\mathbf{O})$ and $I(\mathbf{X}; \mathbf{Y}) = H(\mathbf{Y}) - H(\mathbf{Y}|\mathbf{X})$, we need to show $H(\mathbf{Y}|\mathbf{O}) = H(\mathbf{Y}|\mathbf{X})$.

From the deterministic condition $\mathbf{X} = f(\mathbf{O})$: knowing $\mathbf{O}$ completely determines $\mathbf{X}$. Therefore, conditioning on $\mathbf{O}$ is equivalent to conditioning on both $\mathbf{O}$ and $\mathbf{X}$:

$$H(\mathbf{Y}|\mathbf{O}) = H(\mathbf{Y}|\mathbf{O}, \mathbf{X}) \tag{5}$$

From the Markov property $\mathbf{O} \to \mathbf{X} \to \mathbf{Y}$:

$$H(\mathbf{Y}|\mathbf{O}, \mathbf{X}) = H(\mathbf{Y}|\mathbf{X}) \tag{6}$$

Therefore, $H(\mathbf{Y}|\mathbf{O}) = H(\mathbf{Y}|\mathbf{X})$, which implies $I(\mathbf{O}; \mathbf{Y}) = I(\mathbf{X}; \mathbf{Y})$. $\qquad\square$

This equality is powerful - it means we can evaluate imaging system performance by measuring information between noiseless images and noisy measurements ($I(\mathbf{X}; \mathbf{Y})$), without having to characterize the more challenging distribution over possible objects ($p(\mathbf{o})$). While simplified object models can provide valuable insights - as in the two-point resolution case (**Sec. S1.4.1**) - real objects are rarely known well enough to create accurate models. By focusing on $I(\mathbf{X}; \mathbf{Y})$, we can develop practical methods for estimating information in imaging systems.

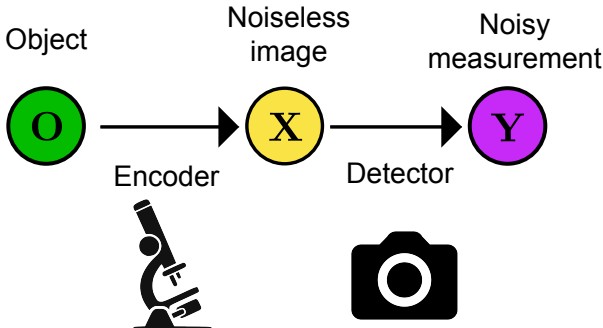

Figure S1: **A minimal probabilistic graphical model of an imaging system.** Three key variables describe the imaging process: (1) the object, which has unknown physical properties we want to measure, (2) the noiseless image, created when the imaging system encodes object properties through a deterministic process, and (3) the noisy measurement, produced when detection adds random noise to the noiseless image. The arrows show how information flows: the measurement can only reveal information about the object if that information was preserved in the noiseless image.

### S1.1.1 Separation of encoding and noise

This minimal model, which separates image encoding from detection noise, has been widely used in information-theoretic analyses of imaging systems [13, 22, 30, 39] and general imaging models [92]. For optical systems, this separation is justified by the "semi-classical" approach common in statistical optics [40]. This method treats light classically until it reaches the detector, where quantum effects introduce noise. Optical systems can experience two types of noise:

- **Classical noise** from fluctuations in light intensity (e.g., thermal light from a bulb or intensity variations in laser light)
- **Quantum "shot" noise** from the random arrival times of individual photons at the detector

In the visible spectrum, quantum noise typically dominates classical noise. This justifies treating light propagation as deterministic until it reaches the detector, where quantum effects become significant. This physical reality supports separating encoding and detection in optical systems.

### S1.1.2 Visualizing high-dimensional distributions in energy coordinates

The probability distributions over images ($p(\mathbf{y})$) and measurements ($p(\mathbf{x})$) are high-dimensional, and thus not possible to visualize directly. In microscopy, for example (**Fig. S2a**), a microscope transforms a distribution of possible cells into noiseless images, which detection then converts into noisy measurements. To provide insight into these complex distributions, we adapt a representation from [93] (**Fig. S2b**). This representation, which we call energy coordinates, offers a complementary perspective to the traditional spatial coordinate representation, enabling a more comprehensive understanding of image distributions and information flow in imaging systems.

Traditionally, images are represented in **spatial coordinates** – the normal way to show images, where each pixel's intensity is plotted at its corresponding position in a 2D space. In this representation, each displayed image is a single sample drawn from the underlying probability distribution of all possible images. While intuitive and familiar, this representation doesn't directly convey the statistical properties of image distributions, as it only shows one or a few instances at a time rather than the full range of possibilities and their relative probabilities.

Alternatively, image distributions can be visualized in the **energy coordinate** representation, which shows the probability mass over pixel values. In this representation, each image of $D$ pixels is a $D$-dimensional vector, and the $D$-dimensional probability density function describes how likely each image is. Low-dimensional projections of the full distribution, such as the marginal distribution $p(x_k)$ or the joint distribution $p(x_k, x_j)$ of two pixels can be plotted and provide insight into the full $D$-dimensional distribution's behavior.

Visualizing the distribution of noisy measurements for a single noiseless image (i.e., $p(\mathbf{y} \mid \mathbf{x})$) in energy coordinates demonstrates the effect of measurement noise (**Fig. S2c**). Noise spreads the probability mass further from the point corresponding to the noiseless image, which increases the overlap between noisy distributions of different images. This overlap makes it harder to determine the true object from a noisy measurement, with the extent of overlap determining how much object information is lost due to noise corruption.

The energy coordinate representation is particularly useful for comparing different imaging modalities (**Fig. S2d**). For example, when comparing two microscope illumination patterns with different spatial coherence, energy coordinates reveal how the encoders affect image distinguishability. An encoder using spatially incoherent illumination maps different objects to more similar images, resulting in overlapping measurement distributions that are difficult to distinguish. In contrast, an encoder using spatially coherent illumination creates more distinct images, leading to measurement distributions that remain separable even in the presence of noise. This visualization demonstrates how different illumination patterns affect the information content of the resulting measurements, with more distinct encoded images being more robust to noise corruption. This concept of distinguishability in the presence of noise is identical to the two-point resolution problem (See **Probabilistic two-point resolution**), only with spatial coherence as the variable parameter instead of signal-to-noise ratio and resolution. This demonstrates how the information-theoretic approach comprehensively captures imaging system performance while abstracting away the specific physical details of different imaging modalities.

### S1.1.3 Mathematical formalism for encoders

There is a family of encoders $\mathcal{E}$ consisting of functions $e_\theta : \boldsymbol{O} \to \boldsymbol{X}$, where $\boldsymbol{O}$ is the domain - the space of possible objects, $\boldsymbol{X}$ is the codomain - the space of possible noiseless images, and $\theta$ is the parameter(s) that define a particular encoder. For example, in the case of a linear, shift-invariant encoder, $\mathcal{E}$ would be the set of all linear, shift-invariant functions and $\theta$ would define a specific point spread function.

The action of an encoder is to take an object $\mathbf{o} \in \boldsymbol{O}$ and form a noiseless image $\mathbf{x} \in \boldsymbol{X}$ of it. This noiseless image will then undergo a measurement process, resulting in a noisy measurement $\mathbf{y} \in \boldsymbol{Y}$. The measurement process is modeled as a conditional probability distribution $\mathrm{P_{Y|X}}$, which describes the probability of observing a particular noisy measurement given a particular noiseless image.

The information carried by noisy measurements is determined by the distributions $\mathrm{P_{Y|X}}$ and $\mathrm{P_X}$. It is quantified by the mutual information between the noiseless image and the noisy measurement, $I(\mathbf{X}; \mathbf{Y})$, which can be expressed as:

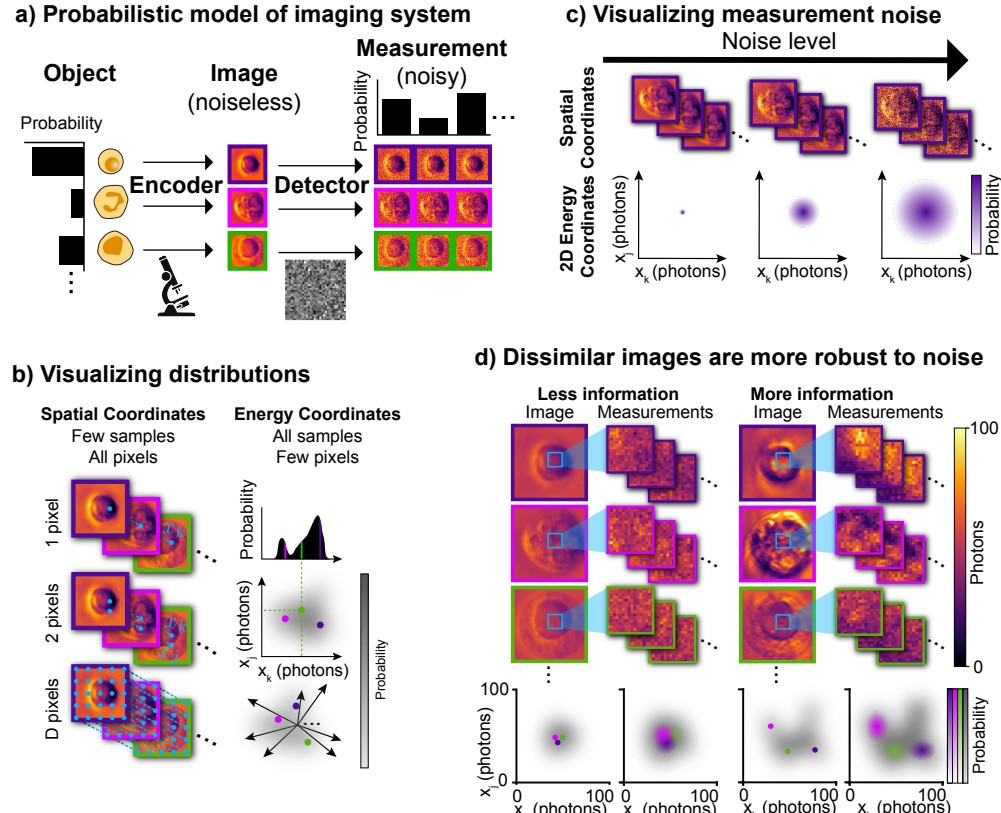

Figure S2: **Probabilistic modeling and visualization of imaging systems. a)** Probabilistic model of microscopy, in which a distribution of objects (cells) are encoded to a distribution of images by a microscope and detected as noisy measurements. **b)** Two complementary ways to visualize measurement distributions: spatial coordinates show all pixels for a few samples, while energy coordinates show the full probability distribution for selected pixels. **c)** Visualization of how measurement noise affects distributions: as noise increases, measurement probability spreads further from the noiseless value. **d)** Demonstration of how encoder design affects information preservation: more distinct noiseless images (right) create measurements that remain separable even with noise, while similar images (left) produce overlapping distributions that make object discrimination harder.

$$I(\mathbf{X}; \mathbf{Y}) = \mathbb{E}_{\mathbf{X}, \mathbf{Y}} \left[ \log \left( \frac{p(\mathbf{X}, \mathbf{Y})}{p(\mathbf{X}) p(\mathbf{Y})} \right) \right] \tag{7}$$

where $p(\mathbf{x}, \mathbf{y})$ is the joint probability of observing a particular noiseless image and a particular noisy measurement, and $p(\mathbf{x})$ and $p(\mathbf{y})$ are the marginal probabilities of observing a particular noiseless image and a particular noisy measurement, respectively.

Better encoders will produce noisy measurements that contain higher mutual information. It it thus of interest to investigate what limits the mutual information. One way of doing this is based on a decomposition of the mutual information into two terms:

$$I(\mathbf{X}; \mathbf{Y}) = H(\mathbf{X}) - H(\mathbf{X} \mid \mathbf{Y}) \tag{8}$$

$H(\mathbf{X})$ is the entropy of the noiseless images, and $H(\mathbf{X} \mid \mathbf{Y})$ is the conditional entropy of the noiseless images given the noisy measurements, which quantifies the uncertainty about the noiseless images that remains after observing the noisy measurements.

There are multiple ways of mathematically modeling the space of noiseless images $\boldsymbol{X}$, which depends on whether the images are continuous or discrete over space and in energy. For example, a model of noiseless images over continuous space and energy would be a space of continuous functions over space which output real numbers corresponding to an amount of energy, whereas a discrete model would be a space of finite-dimensional vectors that take discrete values (the number of photons) at a discrete set of locations (the pixels). Combinations of these are also possible, such as a continuous model over space and discrete model over energy, or a discrete model over space and continuous model over energy.

With any model, the space of noiseless images is ultimately finite in some sense. Energy cannot be infinitely concentrated in a single point[1], and the physics of wave-propagation effectively constrain electromagnetic waves to a finite number of degrees of freedom [94]. This means that even in the continuous/continuous case, any noiseless image can be represented to an arbitrary level of precision by a finite number of samples.

It is thus of interest to understand the limits of the entropy of the noiseless images, because this will determine the limits of the mutual information. The space $\boldsymbol{X}$ will have either finite volume or finite cardinality, as dictated by physical constraints. A uniform distribution over this space, in which all noiseless images are equally likely, will have the maximum possible entropy. However, due to their physical constraints, encoders will not in general be able to produce noiseless images that are uniformly distributed over $\boldsymbol{X}$, leading to an inefficiency in the amount of information that can be carried by the noisy measurements.

### S1.2 Probabilistic two-point resolution

To illustrate information-theoretic analysis of imaging systems, we first provide a simplified example: the classic two-point resolution problem. Typically, resolution is determined by the Abbe diffraction limit, which says that two diffraction-limited point sources will be "resolved" if they are sufficiently far apart (at least $\lambda/(2NA)$, where NA is the numerical aperture). The Abbe criteria, however, does not take into account the effects of noise, aberrations, coherence, or pixel sampling, among other factors. Our mutual information analysis can naturally incorporate all these effects into a single comprehensive metric.

We demonstrate with a simplified scenario, where the task is to distinguish between one point source or two closely spaced dimmer point sources, in the presence of measurement noise. We can derive the information content analytically for this idealized case (see **Section S1.4.1**) in which we model the object as either one point or two half-energy points, with equal probability [95, 96]. We assume that the imaging system has a diffraction-limited point spread function, with measurements corrupted by additive Gaussian noise (**Fig. S3a**). This simplified setup allow us to examine how information content accounts for both resolution and signal-to-noise ratio.

We seek to find the minimum numerical aperture needed to resolve two points that are spaced $\delta d$ apart. The Abbe diffraction limit says the numerical aperture should be at least $\lambda/(2\delta d)$. However, without noise we can theoretically resolve any arbitrarily spaced points. Conversely, if there is a lot of noise, we need a larger numerical aperture than the Abbe limit predicts to resolve these same two points. Below, we show how mutual information can quantify this interplay between resolution and noise (**Fig. S3c**), predicting the achievable classification accuracy (**Fig. S3b**).

With two equally probable object states, the maximum possible information is 1 bit. Since the optical system is deterministic, it introduces no randomness and $I(\mathbf{O}; \mathbf{Y}) = I(\mathbf{X}; \mathbf{Y})$ exactly, because the randomness in the object distribution is the only source of randomness in the noiseless image. This allows us to directly calculate the information preserved through the imaging process. Unlike the general case where we must estimate the entropy of measurements $H(\mathbf{Y})$ through a cross-entropy upper bound, in this scenario we can derive the probability density of measurements analytically:

$$p(\mathbf{y}) = \frac{1}{2}\mathcal{N}(\mathbf{y}; \mathbf{x}_1, \sigma^2 I) + \frac{1}{2}\mathcal{N}(\mathbf{y}; \mathbf{x}_2, \sigma^2 I)$$

---

[1]...without collapsing space into a black hole [94]

Here, $\mathbf{x}_1$ is a vector of pixels for a noiseless image of a single point, $\mathbf{x}_2$ is a vector corresponding to two points, and $\mathcal{N}(\mathbf{y}; \boldsymbol{\mu}, \boldsymbol{\Sigma})$ is the multivariate normal probability density function. $\sigma^2 I$ represents a diagonal covariance matrix with additive Gaussian noise with variance $\sigma^2$ at each pixel.

Using this probability density, we can generate $N$ samples from the distribution of measurements and use them to compute the entropy of measurements $H(\mathbf{Y})$ using a Monte Carlo approximation:

$$H(\mathbf{Y}) = \mathbb{E}\left[-\log p(\mathbf{Y})\right]$$
$$\approx -\frac{1}{N}\sum_{i=1}^{N} \log p(\mathbf{y}_i)$$

In this case of additive Gaussian noise, $H(\mathbf{Y}|\mathbf{X})$ also has a closed form analytical expression (**Sec. S2.3.2**). Subtracting these two quantities enables computation of mutual information with error converging to zero as the number of samples increases.

In **Figure S3c**, we plot the mutual information for a range of diffraction-limited resolutions and signal-to-noise ratios. As expected, there is a trend towards more information for higher resolution (higher numerical aperture) systems and higher signal-to-noise ratio measurements. However, a high-resolution measurement can contain less information than a low-resolution one, if the low-resolution one has a sufficiently better signal-to-noise ratio. When mutual information is equal to 1 bit, the measurements can be classified as coming from one or two point sources with perfect accuracy. When mutual information is zero, the measurement contains no information about which of the two scenarios it is, so the best classifier will only achieve 50 percent accuracy through random guessing. **Figure S3b** shows this relation between mutual information and classification accuracy, after deriving an optimal classifier analytically. As expected, the more information preserved in the measurement, the better an optimal classifier can perform.

This two-point resolution example demonstrates how our unified treatment with information theory can evaluate resolution limits with noise effects taken into consideration. Such analysis can be extended to more complex imaging scenarios, where many different factors influence the encoder (e.g. aberrations, filters) and the measurement quality (e.g. spectral sensitivity, coherence). While our analysis here can provide valuable intuition about when two points will be "distinguishable," real imaging scenarios involve high-dimensional measurements of complex objects. The key insight carries over: better encoders create measurements that are more distinguishable in the presence of noise. The geometric interpretation of these high-dimensional probability distributions provides additional insight into why information theory is well-suited for analyzing imaging systems (**Sec. S1.1.2**).

### S1.3   Encoder inefficiency in 1D

To understand how physical constraints limit information encoding, we expand on the 1D two-point resolution example to analyze a more general 1D model system of an imaging system. This example reveals a fundamental concept we call "encoder inefficiency" - the gap between theoretically optimal encoding and what physical constraints allow.

The family of encoders $\mathcal{E}$ studied were 1D bandlimited, nonnegative, linear-shift invariant, infinitely periodic point spread functions. **Figure S4a** shows the outputs of a representative encoder in this family (i.e. a specific point spread function) acting on a distribution of delta function objects. This system can be thought of as a simplified version of an imaging system that uses spatially incoherent illumination, such as in photography or microscopy [97].

The set of possible output signals $X$ for this family of encoders is identical to the set of possible point spread functions: bandlimited, nonnegative, and infinitely periodic signals (**Fig. S6b**). These output signals (which are analogous to the noiseless images in discussed in **Mathematical framework**) can be viewed either in spatial coordinates, in which their energy density is plotted as a function of space, or in energy coordinates (**Fig. S2**), with values found by integrating areas of the signal corresponding to "pixels."

Signals cannot have both finite bandwidth and finite extent in space, and they thus require infinite samples to represent with arbitrary accuracy in the absence of further constraints. This is typically handled in one of two ways: treating signals as both band-limited and spatially finite and sampling at

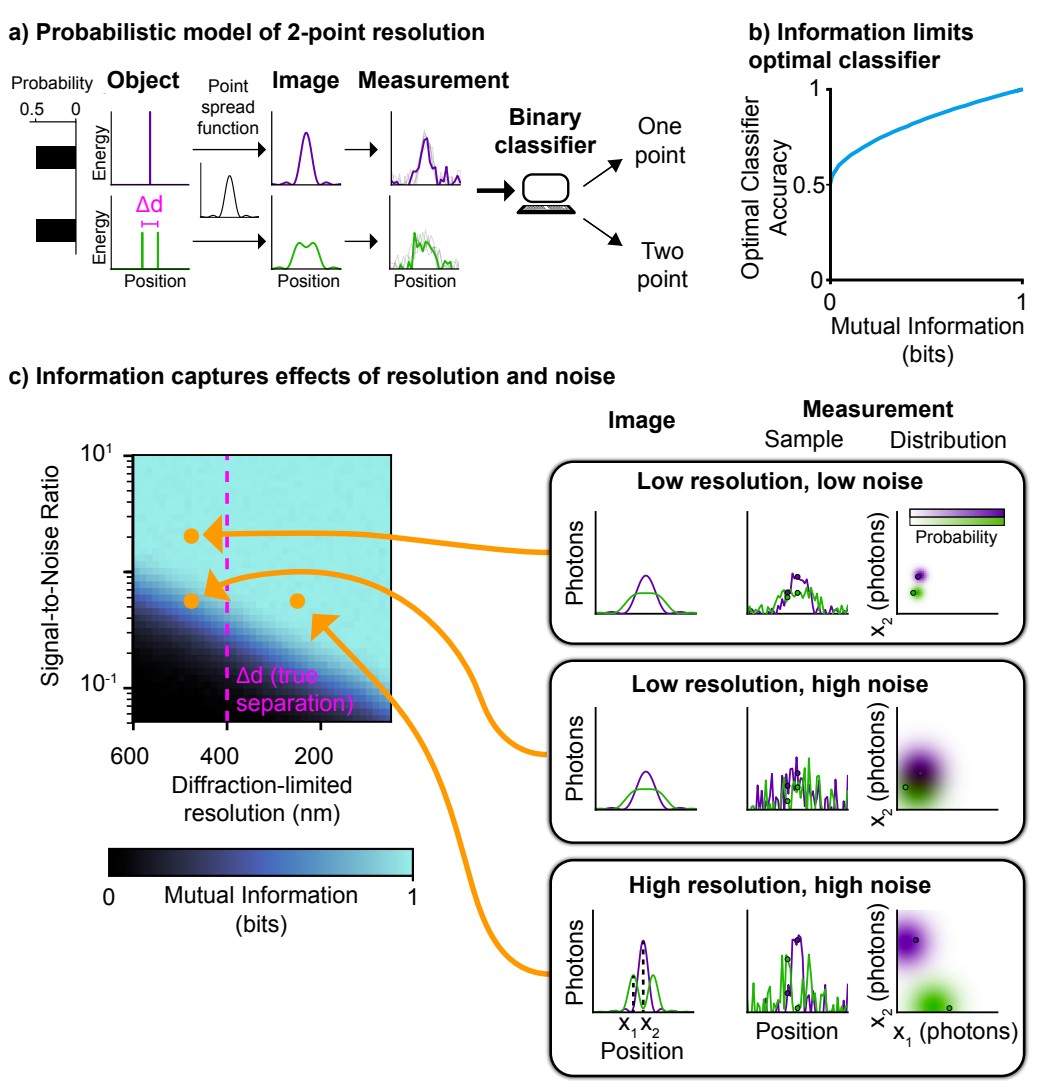

Figure S3: **Mutual information captures imaging trade-offs between resolution and noise for a simplified case of two-point resolution. a)** Probabilistic model of two-point resolution: an object (one point or two half-energy points) is blurred by a point spread function and corrupted by noise. A binary classifier attempts to determine the object type from the measurement. **b)** The accuracy of an optimal classifier is fundamentally limited by the information content of the measurements. **c)** Information analysis reveals how resolution and noise interact: (Left) information in measurement as a function of signal-to-noise ratio and diffraction-limited resolution. (Right) Example images, measurements, and measurement distributions (at pixel locations $x_1$ and $x_2$). The top row and bottom row achieve equivalent information with low-resolution, low-noise measurements and high-resolution, high noise measurements.

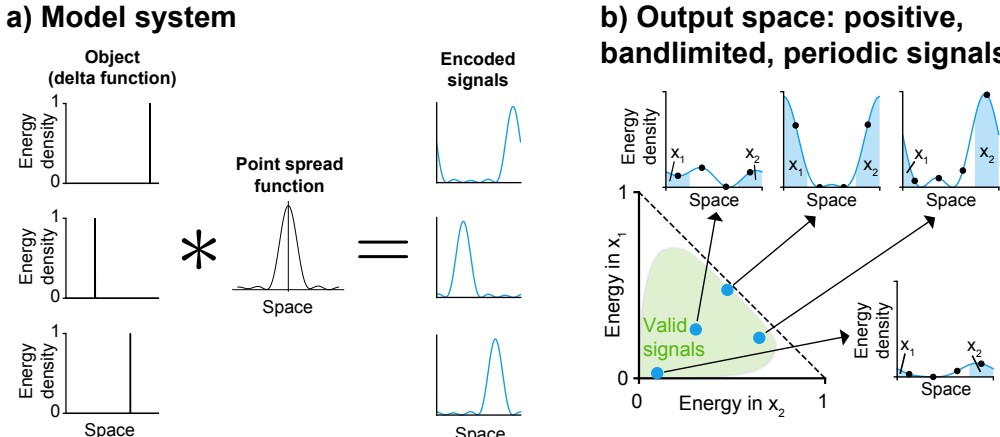

Figure S4: **1-dimensional model system. a)** An object distribution (e.g. a randomly-located delta function) and a linear, shift-invariant point spread function encoder. **b)** $M$-dimensional energy coordinate representation of the set of all encodable signals in which each signal corresponds to a single point.

finite density over finite extent; or considering band-limited signals with infinite spatial extent, which asymptotically have finite degrees of freedom and can thus be sampled at finite rates with arbitrary accuracy [94].

Here, we avoid these complications by considering band-limited signals that are infinitely periodic in space. This means the signal can be represented exactly by sampling at the Nyquist rate over a single period (because sampling beyond this period would yield the same values).

As a result of this simplification, there is a bijective mapping between the set of possible signals and non-negative vectors in $D$-dimensional space–that is, each signal in the set corresponds to a point in $D$-dimensional space (similar to the argument made in [93]). This allows us to computationally analyze this finite-dimensional space with insights that can be applied to the more complicated space of continuous signals.

As discussed in **Visualizing high-dimensional distributions in energy coordinates**, the effect of measurement noise in the energy coordinate representation is to turn a point (i.e. a signal/image) into a cloud of probability mass representing the possible noisy realizations of that signal/image. Here we assume, without loss of generality, that all measurement noise is additive Gaussian (See **Conditional entropy with additive Gaussian noise**). The amount of information that can be encoded is determined by how well dispersed the distribution of encoded signals can be in this space such that the noisy versions of different signals minimally overlap. Thus, the volume of the space of possible signals is critical: it determines how much room there is to map different objects to non-overlapping signals. Though there are an infinite number of signals in the set, only a finite number can be distinguished with a given level of certainty in the presence of noise.

What is the volume of the space of possible signals? Given that all signals have energy $\leq 1$, the vector that defines their representation in energy coordinates must have $L_1$ norm also $\leq 1$. Thus, all signals must correspond to points that lie inside the positive orthant of the $L_1$ unit ball. However, not every point in this space will correspond to a valid signal: for example, a vector with a single $1$ and the rest $0$s will not be possible, because this would entail concentrating all of the signal's energy within a single pixel.

It is unclear to us if there is an analytical expression that defines the set of possible signals, but we can investigate the size of this set empirically. To do so, we set up an optimization problem in which we pick a fixed object and a target energy coordinate representation of a signal (e.g. a vector with a single $1$ and the rest $0$s). We then find the optimal point spread function that brings the object closest to the target signal by optimizing an encoder to minimize this distance using gradient descent. Repeating this experiment over a grid of target signals shows which signals can be reached, and which cannot, thereby revealing the limits of the space of possible signals.

Repeating this experiment with different fixed objects illustrates an important insight about encoders: Their range is object dependent (**Fig. S5**). This is a direct result of their physical constraints. The 1D encoder in this simulation is representative of imaging systems governed by intensity point spread functions [97]. Such encoders have at least two important physical constraints: 1) they can only reduce energy (if they attenuate light) or preserve it. They cannot, for example, encode a dim object to a signal with greater energy. 2) They can only disperse, and not re-concentrate, energy. Every point spread function (under the constraints of non-negativity and linear shift invariance) can only map objects to signals that are blurrier versions of themselves.

**Figure S5** shows the consequence of the second constraint for three different objects with equal energy. The single delta function in the top row can be encoded to the broadest range of possible signals, since it is the most concentrated to begin with. More dispersed objects, like the 8 delta functions each with $\frac{1}{8}$ energy in the bottom row can only be encoded to a smaller volume of possible signals.

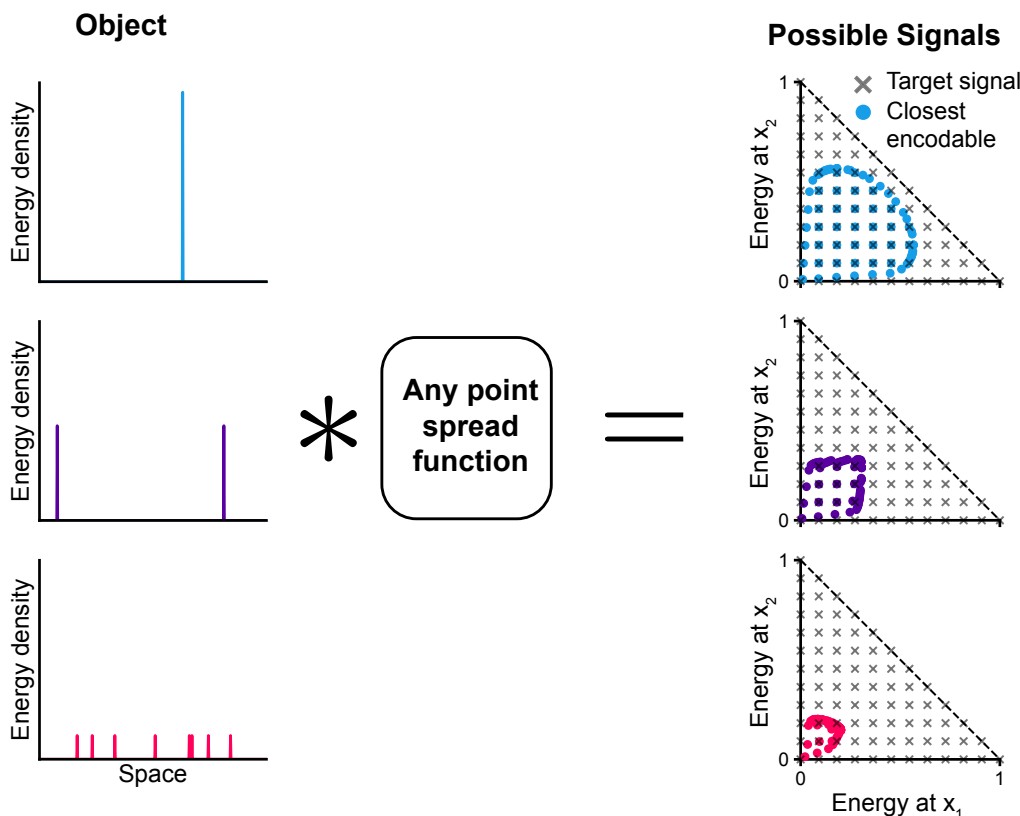

Figure S5: **Object-dependence of encoder range.** The set of signals encodable from an object depends on that object's properties. More dispersed objects like the 8 delta functions (bottom) can only reach a subset of the signals reachable from a concentrated delta function (top), despite equal energies. This is due to physical constraints preventing reconcentrating energy. The closest encodable signals to the target signals were found by solving an optimization problem to find the optimal encoder that encodes the object to the target signal.

The range of a family of encoders for a fixed object of a particular type determines the volume of the set of possible encoded signals, and thus places an upper limit on the amount of encoded information. Physical constraints limit the functions an encoder family can implement. As a result, even optimal encoders generally cannot achieve the theoretically ideal signal distribution (which would be uniform over all possible signals when dealing with additive Gaussian noise). An encoder must handle not just one object, but an entire distribution of objects. While we might find different encoders that each map a specific object to a desired signal, no single encoder can optimally map all objects simultaneously. This fundamental limitation prevents achieving the theoretically ideal signal distribution.

Given three constraints - an object distribution, an energy limit on signals, and a noise model - the optimal signal distribution maximizes mutual information between objects and encoded signals. However, physical constraints and object-dependence often prevent encoder families from achieving this optimum. We define "encoder inefficiency" as the gap between this theoretical optimum and the best distribution achievable by a given encoder family.

We can measure encoder inefficiency experimentally using a simplified test case with delta functions of unit energy at random positions as objects, and additive Gaussian measurement noise. Under these conditions, the distribution of signals carrying maximum information has uniform probability over the set of possible signals.

To find the best achievable distribution, we first use Information-Driven Encoder Analysis Learning (IDEAL) (See **Encoder design via information maximization with IDEAL**) to learn the optimal encoder. We then generate random objects and encode them with this optimal encoder to sample from the distribution of encoded signals, add noise, and estimate the mutual information. We use the PixelCNN-based estimator[2], treating the signals as 2D images to match the estimator's design.

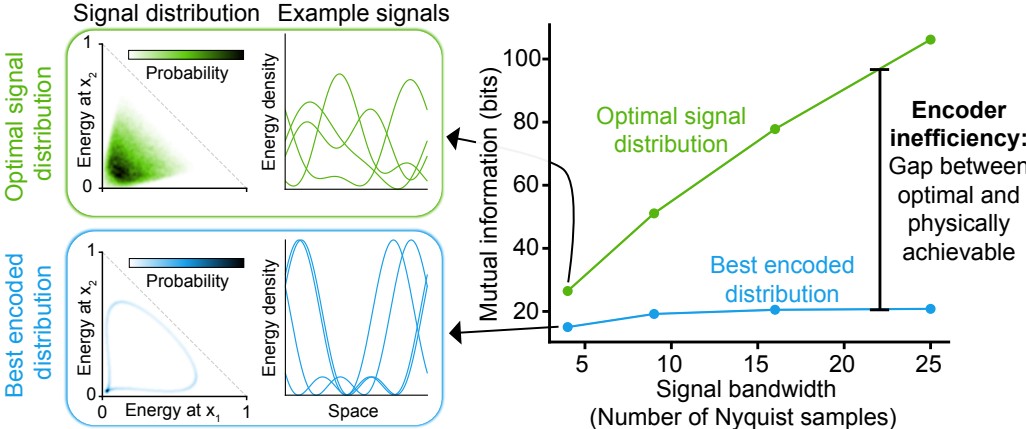

Figure S6: **Physical constraints limit the amount of information that can be encoded.** (Top left) The optimal distribution of energy limited signals for additive, signal-independent Gaussian noise is uniform (it appears non-uniform in this 2D projection of a 4D space). (Bottom left) The best encodable distribution using an optimal point spread function is far from uniform. (Right) The gap between the information in the optimal signal distribution and the best encodable distribution quantifies the inefficiency of the encoder family.

The results of this experiment are shown in **Figure S6**. The optimal point spread function encodes objects to only a small fraction of the total volume of possible signals, resulting in encoded signals that are significantly less random in appearance the the optimal uniform distribution of signals. (Note: the energy coordinate representation of these signals shown on the left side is a 2-dimensional projection of an $D$-dimensional space, with $D = 4$ for the picture shown, which is why the distribution appears non-uniform.) The mutual information estimates shown on the right side demonstrate the gap between the optimal distribution of signals and the best encodable distribution given the family of linear, shift-invariant encoders.

### S1.4   Effects of signal-to-noise ratio, resolution, and sampling on information

This same experimental setup can be used to investigate how different properties of the encoder/detection process, such as bandwidth, signal-to-noise ratio, and sampling density, affect the amount of information that can be encoded. These experiments provide insight into the tradeoffs between different system parameters as well as connections to the existing literature analyzing information in imaging systems.

---

[2]We used PixelCNN for convenience, though estimators designed for 1D signals [98, 99] might provide better accuracy.

**Signal-to-noise ratio**    Signal-to-noise ratio is a key parameter in imaging systems, and it is widely-appreciated that higher signal-to-noise ratio is a desirable characteristic. For the additive Gaussian noise measurement model, noise is fixed, and the signal-to-noise ratio is determined by the amount of energy in the signal. It is simplest to consider the average signal-to-noise ratio over each signal, which can be found by dividing the energy of the signal by the standard deviation of the noise. Choosing an maximum average signal-to-noise ratio defines a set of signals that can be encoded, which consists of all possible signals with this average signal-to-noise ratio or less. This set has a finite volume, and thus a finite maximum amount of information that can be encoded. Within this set, there are subsets of signals that each have the same signal-to-noise ratio.

### a) Experimental framework

### b) Effects of measurement properties on information

Figure S7: **Effects of signal-to-noise, space-bandwidth product, and sampling density on encoded information. a)** Samples from three different object distributions and the signals and noisy measurements to which they are encoded with an optimal point spread function. **b)** The amount of encoded information increases (Left) logarithmically with the average signal-to-noise ratio with object-dependent rates, (Middle) linearly with the space-bandwidth product of the signal with object-dependent rates. (Right) Sampling signals of fixed bandwidth at increasing densities increases the amount of encoded information, but with diminishing returns across different signal-to-noise ratios (for the 8 delta object distribution).

Sets of signals with higher average signal-to-noise ratios have increasing large volumes, and can thus carry larger amounts of information. As described in the previous section, physical constraints of encoders impose object-dependent limits on the the maximum amount of information that can be encoded. To test how the maximum average signal-to-noise ratio affects the amount of information that can be encoded, we repeated the procedure described on three different distributions of objects: single, randomly-located, unit energy delta functions, 8 randomly-located, delta functions each with $\frac{1}{8}$ energy, and unit energy white noise patterns (note, these are objects, not measurement noise) (**Fig. S7a**). The

results show that for all three object distributions, the best encodable distribution of signals grows logarithmically with the average signal-to-noise ratio, with different objects having different absolute amounts of information for a given signal-to-noise ratio (**Fig. S7b, left**). This is consistent with the intuition that higher signal-to-noise ratio allows for more information to be encoded, and that the amount of information that can be encoded is object-dependent.

**Space-bandwidth product**    Next, we tested the effect of signal bandwidth on information capacity. Optical imaging systems are often characterized in terms of their space-bandwidth product [100], and the space-bandwidth product is often used synonymously with the word "information." A more accurate term for the space-bandwidth product is "degrees of freedom," since it quantifies the potential complexity of an electromagnetic field wave propagating through the system [94]. Information (in Shannon's entropy/mutual information sense) also depends on the signal-to-noise ratio and the object-dependent ability of encoders to map to distinct signals. In our numerical simulation, the spatial extent of signals is fixed, so increasing the bandwidth of the signal increases the space-bandwidth product. We found that captured information increases linearly with the space-bandwidth product, with rate of increase depending on the object distribution (**Fig. S7b, center**).

**Sampling density**    Finally, we examined the effect of sampling density on the amount of information that can be encoded. For a fixed bandwidth, the sampling density determines the number of pixels in the signal. The Nyquist sampling theorem [101, 93] states that a bandlimited signal can be perfectly reconstructed from its samples if the sampling density is at least twice the bandwidth. However, in the presence of noise, even when sampling at the Nyquist rate, there remains residual uncertainty about the signal, and additional samples can reduce this uncertainty. Experimentally, we found that, as expected, increasing the sampling density increases the amount of information, even beyond the Nyquist rate (**Fig. S7b, right**). However, the additional increases in information were progressively smaller. These results were consistent across both different object distributions and different average signal-to-noise ratios (for a delta function object distribution).

### S1.4.1   Full mathematical treatment of two-point resolution

In this section we describe the example of 1-dimensional two-point resolution (See **Probabilistic two-point resolution**) in full mathematical detail. By making assumptions about the object distribution, the encoder, and the noise model, we can write down the exact probability density functions of the object, the noiseless image, and the noisy measurement. This enables writing an exact expression for the mutual information between object and noisy measurement, as well as the classification accuracy of the optimal binary classifier decoder that uses the noisy measurement to classify the object as being a single point source or two point sources.

**Object**    The object is a mixture of two possibilities that each occur with probability $\frac{1}{2}$: Either a single point source with energy 1 or two point sources with energy $\frac{1}{2}$ and separation distance $\Delta$ with their midpoint at the same location as the single point source. We represent these objects mathematically as $\mathbf{o}_1$ and $\mathbf{o}_2$ respectively. Using $r$ to denote spatial position, with the single-point source object located at $r = 0$:

$$\mathbf{o}_1(r) = \delta(r)$$
$$\mathbf{o}_2(r) = \frac{1}{2}\delta(r - \frac{\Delta}{2}) + \frac{1}{2}\delta(r + \frac{\Delta}{2})$$

where $\delta$ is the Dirac delta function.

The random object $\mathbf{O}$ has probability $\frac{1}{2}$ of being $\mathbf{o}_1$ and $\frac{1}{2}$ of being $\mathbf{o}_2$. Thus, its probability density function can be written as:

$$p(\mathbf{o}) = \frac{1}{2}\delta(\mathbf{o} - \mathbf{o}_1) + \frac{1}{2}\delta(\mathbf{o} - \mathbf{o}_2)$$

$\delta(\mathbf{o} - \mathbf{o}_1)$ is 1 when the object is $\mathbf{o}_1$ and 0 otherwise, and similarly for $\delta(\mathbf{o} - \mathbf{o}_2)$.

**Encoder**   The encoder is a 1-dimensional linear shift-invariant imaging system with a diffraction-limited intensity point spread function $h(r)$:

$$h(r) = \frac{\sin(\frac{2\pi \mathrm{NA}}{\lambda} r)}{\frac{2\pi \mathrm{NA}}{\lambda} r}$$

where NA is the numerical aperture of the system, $\lambda$ is the wavelength of light, and $r$ is the spatial coordinate.

The noiseless image is the convolution of the object with the point spread function:

$$\mathbf{x} = \mathbf{o} * h$$

This gives rise to two possible noiseless images, $\mathbf{x}_1$ and $\mathbf{x}_2$, corresponding to the two possible objects $\mathbf{o}_1$ and $\mathbf{o}_2$ respectively. The probability density function of the random noiseless image $\mathbf{X}$ is thus:

$$p(\mathbf{x}) = \frac{1}{2}\delta(\mathbf{x} - \mathbf{x}_1) + \frac{1}{2}\delta(\mathbf{x} - \mathbf{x}_2)$$

**Detector**   The noisy measurement is formed by adding independent Gaussian noise with variance $\sigma^2$ to each pixel of the noiseless image. We assume a pixel size much smaller than the minimum pixel size dictated by the Nyquist sampling theorem, so sampling effects minimally influence the results.

The random noisy measurement $\mathbf{Y}$ is thus a length $D$ vector of pixels. Its probability distribution is found by taking the mixture of two deltas distribution of the noiseless images and adding Gaussian noise to each pixel. This gives a mixture of two multivariate Gaussian distributions with mean vectors given by the noiseless images and a diagonal covariance matrix with variances equal to the noise variance. The probability density function of the noisy measurement is thus:

$$p(\mathbf{y}) = \frac{1}{2}\mathcal{N}(\mathbf{y}; \mathbf{x}_1, \sigma^2 I) + \frac{1}{2}\mathcal{N}(\mathbf{y}; \mathbf{x}_2, \sigma^2 I) \tag{9}$$

where $\mathcal{N}(\mathbf{y}; \mathbf{x}, \sigma^2 I)$ is the multivariate Gaussian distribution with mean vector $\boldsymbol{\mu} = \mathbf{x}$ and covariance matrix $\boldsymbol{\Sigma} = \sigma^2 \boldsymbol{I}$.

**Mutual information**   The mutual information between the object and the noisy measurement $I(\mathbf{O}; \mathbf{Y})$ is equal to the mutual information between the noiseless image and the noisy measurement $I(\mathbf{X}; \mathbf{Y})$, since the object is fully determined by the noiseless image. We focus on the mutual information between the noiseless and noisy images, which can be calculated by decomposing it into a difference of entropies:

$$I(\mathbf{X}; \mathbf{Y}) = H(\mathbf{Y}) - H(\mathbf{Y}|\mathbf{X})$$

where $H(\mathbf{Y})$ is the entropy of the noisy measurement and $H(\mathbf{Y}|\mathbf{X})$ is the conditional entropy of the noisy measurement given the noiseless image.

Under the additive Gaussian noise model, $H(\mathbf{Y}|\mathbf{X})$ is a constant that is a function of the the variance $\sigma^2$ and the number of dimensions (pixels) $D$. It can be analytically simplified as shown in Equation 24.

The entropy of the noisy measurement $H(\mathbf{Y})$ can be expanded as:

$$H(\mathbf{Y}) = -\mathbb{E}[\log p(\mathbf{y})]$$

Since we can easily generate samples from the distribution of the noisy measurements and we know the true probability density $p(\mathbf{y})$, we can estimate this entropy with a Monte Carlo approximation of $N$ samples:

$$H(\mathbf{Y}) \approx -\frac{1}{N}\sum_{i=1}^{N} \log p(\mathbf{y}_i)$$

**Decoder** Since the goal of the imaging system in this simple example is to classify the object as being a single point source or two point sources [95], the decoder is a binary classifier that takes in the noisy measurement and outputs a decision as to whether the object was a single point or two points.

The optimal decoder is the Bayes classifier, which chooses the object class that maximizes the posterior probability given the noisy measurement. In this case, since the prior probabilities of the two object classes are equal, the Bayes classifier is equivalent to the maximum likelihood classifier, which chooses the object class that maximizes the likelihood of the noisy measurement.

Given the probability density of the noisy measurement in Equation 9, the Bayes/maximum likelihood classifier decides the object is two points if:

$$\mathcal{N}(\mathbf{y}; \mathbf{x}_2, \sigma^2 I) > \mathcal{N}(\mathbf{y}; \mathbf{x}_1, \sigma^2 I)$$

Plugging in the expressions for the multivariate Gaussian distributions, this simplifies to:

$$\|\mathbf{y} - \mathbf{x}_2\|^2 < \|\mathbf{y} - \mathbf{x}_1\|^2$$

In other words, the noisy measurement $\mathbf{y}$ is classified as two points if its Euclidean distance to the noiseless image $\mathbf{x}_2$ is less than its distance to $\mathbf{x}_1$, and classified as one point otherwise.

Using the analytic expressions for mutual information and classification accuracy, they can be shown to have a monotonic relationship with each other (**Fig. S8**). This demonstrates the fundamental link between the information content of the measurements and the achievable performance of downstream decoding in this minimal example.

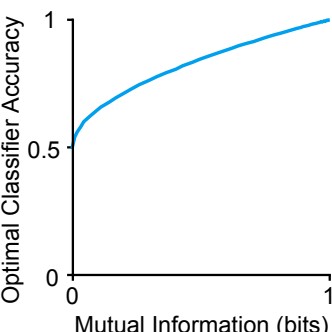

Figure S8: **Classification accuracy vs information in two-point resolution**. Performance of the optimal binary classifier decoder that uses the noisy measurement to classify the object as being a single point source or two point sources. The classification accuracy in this simple example has a monotic relationship with the mutual information between the object/noiseless image and the noisy measurement.

## S1.5 Expanded model

Our basic framework works for many imaging systems, including many optical systems whose purpose is to capture the object in as much detail as possible. Here we present an expanded model that accounts for specialized cases involving stochastic encoders and task-specific information, with detailed treatments provided in subsequent sections.

The minimal model assumes deterministic encoders where the same object always produces the same noiseless image. In specialized cases, this assumption may not hold due to quantum effects in image formation or classical system variations like mechanical drift and illumination fluctuations. We provide a theoretical framework for handling such stochastic encoders in **Section S2.7**.

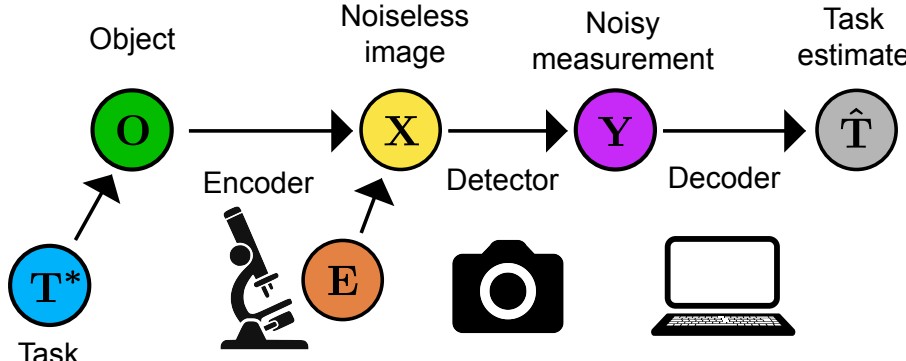

Figure S9: **An expanded probabilistic graphical model of an imaging system.** A generalization of the minimal probabilistic model in **Figure S1**, which in addition to modeling the object, noiseless image, and noisy measurement as random variables, also models randomness of the encoder, and the true and estimated values of the decoder task.

Additionally, the minimal model treats all object information as equally valuable, but many imaging applications focus on specific tasks that may require only a subset of the total object information. We develop extensions for task-specific information estimation in **Section S2.8**.

## S2   Estimating information: Theory

In this section, we develop the mathematical framework for estimating information in imaging systems.

### S2.1   Background

As described in **Mathematical framework**, our probabilistic model represents noiseless images as a random 2D array $\mathbf{X}$ with distribution $p(\mathbf{x})$, where each element quantifies the energy at a single pixel. The corresponding noisy measurements form array $\mathbf{Y}$ with distribution $p(\mathbf{y})$. These distributions determine how much object-relevant information survives the measurement process.

We estimate $I(\mathbf{X}; \mathbf{Y})$, the mutual information between noiseless images and noisy measurements. The units are bits, which have a concrete operational meaning: in ideal conditions, each bit enables perfect discrimination between $2^{I(\mathbf{X}; \mathbf{Y})}$ different objects. In practice, noise prevents such perfect discrimination—instead, each bit quantifies how well multiple object states can be partially distinguished from each other.

While imaging system performance typically varies across the field of view, we can quantify overall information content using a single value: the mutual information rate (information per pixel). This parallels how optical resolution is often characterized by a single number despite spatial variations [97].

### S2.2   Mutual information estimator

Mutual information estimation is a well-studied problem in many fields, including machine learning, neuroscience, and computational biology, and a number of different approaches have been proposed, some of which attempt to estimate mutual information directly [41–43], and others which try to infer its value by computing upper or lower bounds [51, 44]. It is a challenging statistical problem to solve in general, and many estimators and bounds are known to suffer from high bias and/or variance, particularly on high-dimensional problems [51, 44].

Many mutual information estimators rely on first estimating entropy, which is a measure of the uncertainty of a random variable. The outcome of random variables with higher entropy are more uncertain and harder to predict than the outcome of those with lower entropy, because they are more random. Equivalently, the higher a distribution's entropy is, the more spread out it is, and the harder

it would be to digitally compress samples from it. Mathematically, entropy is defined as the expected value of the negative log of the probability density/mass function:

$$\mathrm{H}\left(\mathbf{Y}\right) = -\mathbb{E}\left[\log p(\mathbf{Y})\right] \tag{10}$$

Like mutual information estimation, there is a large body of literature on entropy estimation and many different approaches have been proposed [102], including those that form an estimate of the probability density function $\hat{p}(\mathbf{y})$ and plug it into the definition of entropy and those that estimate entropy based on the similarity of samples from $p(\mathbf{y})$ [103–105]. Both approaches, however, face difficulties in high dimensions.

There are multiple ways of decomposing mutual information into a difference of entropies. Our approach is based upon the decomposition:

$$I(\mathbf{X}; \mathbf{Y}) = \mathrm{H}\left(\mathbf{Y}\right) - \mathrm{H}\left(\mathbf{Y} \mid \mathbf{X}\right) \tag{11}$$

Here, $\mathrm{H}\left(\mathbf{Y}\right)$ is the entropy of the noisy image distribution $p(\mathbf{y})$. Both variations in the object and measurement noise contribute to the randomness of measurements. We are interested in the information about the object, thus to quantify how much of these variations the measurements contain, we must subtract the entropy contributed by the noise, $\mathrm{H}\left(\mathbf{Y} \mid \mathbf{X}\right)$.

This decomposition is especially valuable for optical imaging because measurement noise typically acts independently at each pixel, greatly simplifying the calculation of the conditional entropy $\mathrm{H}\left(\mathbf{Y} \mid \mathbf{X}\right)$.

(A technical note: in optical systems, the outcomes of the random variables $\mathbf{X}$ and $\mathbf{Y}$ in our model are discrete, because they are 2D arrays of pixels, where each pixel takes on an intensity value that is an integer number of photons. However, for computational simplicity, we will approximate them as continuous random variables and use differential entropy instead of discrete entropy [80, 91]. These approximations break down at very low photon counts, less than $\sim 20$ photons, so in this paper, we use only data with photon counts greater than this. A possible direction of future work is to extend our approach to work with discrete random variables, which would allow it to be applied to data with lower photon counts.)

We begin by describing the simpler of the two terms to estimate, the conditional entropy $\mathrm{H}\left(\mathbf{Y} \mid \mathbf{X}\right)$.

### S2.3 Estimating conditional entropy of measurement noise

The conditional entropy of noisy measurements given noiseless images can be written out in terms of expectations over the logarithm conditional probability of noisy measurements given a noiseless image $p(\mathbf{y} \mid \mathbf{x})$:

$$\mathrm{H}\left(\mathbf{Y} \mid \mathbf{X}\right) = -\mathbb{E}_{\mathbf{X}, \mathbf{Y}}\left[\log p(\mathbf{Y} \mid \mathbf{X})\right] \tag{12}$$

By the Law of Total Expectation, this can be written as:

$$\mathbb{E}_{\mathbf{X}, \mathbf{Y}}\left[\log p(\mathbf{Y} \mid \mathbf{X})\right] = \mathbb{E}_{\mathbf{X}}\left[\mathbb{E}_{\mathbf{Y}}\left[\log p(\mathbf{Y} \mid \mathbf{X}) \mid \mathbf{X}\right]\right] \tag{13}$$

$$= \mathbb{E}_{\mathbf{X}}\left[\mathbb{E}_{\mathbf{Y}}\left[\log p(\mathbf{Y} \mid \mathbf{X})\right]\right] \tag{14}$$

$p(\mathbf{y} \mid \mathbf{x})$ embodies the various sources of noise in the detection process, including photon shot noise, detector read noise, etc. Here, we utilize established analytic models of detection noise in optical imaging [40]. Empirical results suggest that the true noise in systems deviates from these models in low-light conditions [106], but since our experiments are conducted in the high-light regime, we will assume that these models are accurate for the purposes of this paper. A possible direction for future work is to learn more accurate noise models from the data, as was done in [107].

Assuming we have access to a dataset of $N$ samples from the distribution of noiseless images $\{\mathbf{x}^{(1)}, \mathbf{x}^{(2)}, \ldots \mathbf{x}^{(N)}\}$, the outer expectation can be estimated through Monte Carlo approximation:

$$-\mathbb{E}_{\mathbf{X}}\left[\mathbb{E}_{\mathbf{Y}}\left[\log p(\mathbf{Y} \mid \mathbf{X})\right]\right] \approx -\frac{1}{N}\sum_{i=1}^{N}\mathbb{E}_{\mathbf{Y}}\left[\log p(\mathbf{Y} \mid \mathbf{x}^{(i)})\right] \tag{15}$$

$$= -\frac{1}{N}\sum_{i=1}^{N}\mathrm{H}\left(\mathbf{Y} \mid \mathbf{x}^{(i)}\right) \tag{16}$$

Here, $\mathrm{H}\left(\mathbf{Y} \mid \mathbf{x}^{(i)}\right)$ is the conditional entropy of the distribution of noisy images given the $i$th noiseless image.

### S2.3.1   Conditionally independent noise at each pixel

$\mathrm{H}\left(\mathbf{Y} \mid \mathbf{x}^{(i)}\right)$ is a function of the noise introduced in the detection process, which is modeled by the probability distribution $p(\mathbf{y} \mid \mathbf{x}^{(i)})$. In optical imaging, common analytic noise models like additive Gaussian and Poisson shot noise typically assume that the noise at each pixel is conditionally independent of the noise at other pixels, given the true (noiseless) pixel value. When this is true, $p(\mathbf{y} \mid \mathbf{x}^{(i)})$, which is a joint distribution over all $D$ pixels in the noisy measurement, can be simplified by factoring it into a product of scalar distributions for each pixel, where $y_k$ and $x_k^{(i)}$ are the intensity values at the $k$th pixel in the noisy measurement and the $i$th noiseless image, respectively:

$$p(\mathbf{y} \mid \mathbf{x}) = p(y_1, y_2, \ldots y_D \mid x_1^{(i)}, x_2^{(i)}, \ldots x_D^{(i)})$$

$$= \prod_{k=1}^{D} p(y_k \mid x_k^{(i)})$$

This factorization simplifies the calculation of conditional entropy, because it is much easier to compute $D$ scalar conditional entropies than a single joint conditional entropy over $D$ variables. Mathematically, this simplification can be seen by plugging the factorized distribution into the definition of conditional entropy:

$$\mathrm{H}\left(\mathbf{Y} \mid \mathbf{x}^{(i)}\right) = -\mathbb{E}_{\mathbf{Y}}\left[\log \prod_{k=1}^{D} p(y_k \mid x_k^{(i)})\right] \tag{17}$$

$$= -\mathbb{E}_{\mathbf{Y}}\left[\sum_{k=1}^{D} \log p(y_k \mid x_k^{(i)})\right] \tag{18}$$

$$= -\sum_{k=1}^{D}\mathbb{E}_{\mathbf{Y}}\left[\log p(y_k \mid x_k^{(i)})\right] \tag{19}$$

$$= \sum_{k=1}^{D}\mathrm{H}\left(\mathrm{Y}_k \mid x_k^{(i)}\right) \tag{20}$$

Here, $H(\mathrm{Y}_k \mid x_k^{(i)})$ is the conditional entropy of the $k$th pixel in the noisy image given the intensity of the $k$th pixel in the noiseless image.

This is a scalar quantity and can be calculated analytically for many common noise models. We will discuss two such models here: additive Gaussian noise and Poisson noise.

### S2.3.2   Conditional entropy with additive Gaussian noise

Additive Gaussian noise is a simple noise model often used in optical imaging, especially in low-light conditions where the read noise of the detector is the dominant source of noise. In this model, the noise at each pixel is drawn from a Gaussian distribution with mean zero and variance $\sigma^2$. Mathematically:

$$Y_k = X_k + N_k$$

$$N_k \sim \mathcal{N}(0, \sigma^2)$$

The entropy of a (scalar) Gaussian distribution $\mathcal{N}(\mu, \sigma)$ is [80]:

$$H(N_k) = \frac{1}{2} \log_2(2\pi e \sigma^2)$$

Since the noise is independent of the noiseless image, the conditional entropy of the noise at each pixel is the same, and the full conditional entropy (Equation 20) simplifies to:

$$H\left(\mathbf{Y} \mid \mathbf{x}^{(i)}\right) = \sum_{k=1}^{D} H\left(Y_k \mid x_k^{(i)}\right) \tag{21}$$

$$= \sum_{k=1}^{D} H(N_k) \tag{22}$$

$$= \frac{D}{2} \log_2(2\pi e \sigma^2) \tag{23}$$

Plugging this result into Equations 16 and 12 yields:

$$H(\mathbf{Y} \mid \mathbf{X}) = \frac{D}{2} \log_2(2\pi e \sigma^2) \tag{24}$$

To summarize, the conditional entropy of the noisy image given the noiseless image is a constant, independent of the intensity values of the noiseless images, and is equal to the number of pixels in the image times the entropy of the noise distribution at each pixel.

### S2.3.3 Conditional entropy with shot noise

Images with high photon counts are fundamentally limited by shot noise - randomness in photon arrival times due to the quantum nature of light. This shot noise follows a Poisson distribution with rate parameter equal to the expected number of photons at each pixel. When shot noise is the dominant source of noise, it can be accurately approximated by a Gaussian distribution with equal mean and variance [40].

Thus, the conditional entropy of the noise at pixel $k$ for the $i$th noiseless image can be approximated with the entropy of a gaussian distribution:

$$H\left(Y_k \mid x_k^{(i)}\right) = \frac{1}{2} \log_2(2\pi e x_k^{(i)}) \tag{25}$$

Once again making use of the fact that the measurement noise at each pixel is independent of the noise at other pixels conditional on the intensity of the noiseless image at that pixel (Section S2.3.1), we can write the conditional entropy for noiseless image $\mathbf{x}^{(i)}$ as:

$$H\left(\mathbf{Y} \mid \mathbf{x}^{(i)}\right) = \sum_{k=1}^{D} H\left(Y_k \mid x_k^{(i)}\right) \tag{26}$$

$$= \sum_{k=1}^{D} \frac{1}{2} \log_2(2\pi e x_k^{(i)}) \tag{27}$$

The full conditional entropy (Equation 20) simplifies to:

$$H\left(\mathbf{Y} \mid \mathbf{X}\right) \approx \frac{1}{N} \sum_{i=1}^{N} \sum_{k=1}^{D} \frac{1}{2} \log_2(2\pi e x_k^{(i)}) \tag{28}$$

To summarize, the conditional entropy under a Poisson noise model can be approximated as a sum of the log of the intensity values of the noiseless image at each pixel, averaged over $N$ noiseless images. This approximation is accurate when the photon counts are high, and breaks down when the photon counts are low, and is discussed further in **Section S3.5**.

## S2.4  Estimating entropy of noisy images

The second term in the mutual information decomposition, $H\left(\mathbf{Y}\right)$, presents a greater estimation challenge. Unlike the conditional entropy calculated in **Section S2.3**, the joint probability distribution $p(\mathbf{y})$ cannot be factored into independent distributions for each pixel, as pixels throughout both noisy and noiseless images exhibit complex dependencies.

We estimate this entropy by computing an upper bound, an approach that has proven more accurate than alternative bounds in high-dimensional settings [51]. Our method fits a parametric model $p_\theta(\mathbf{y})$ to the empirical distribution of noisy images using maximum likelihood estimation. The optimal parameters $\hat{\theta}_{\mathrm{MLE}}$ are found by minimizing the negative log likelihood of the observed data:

$$\hat{\theta}_{\mathrm{MLE}} = \arg\min_{\theta} -\mathbb{E}\left[\log p_\theta(\mathbf{Y})\right] \tag{29}$$

This loss function, $-\mathbb{E}\left[\log p_\theta(\mathbf{Y})\right]$, is also known as the cross-entropy between the model distribution $p_\theta(\mathbf{y})$ and the empirical distribution $p(\mathbf{y})$.

In practice, it is fit using a dataset of $N$ samples from the empirical distribution $p(\mathbf{y})$:

$$\hat{\theta} = \arg\min_{\theta} -\frac{1}{N} \sum_{i=1}^{N} \log p_\theta(\mathbf{y}^{(i)}) \tag{30}$$

When the model distribution $p_\theta(\mathbf{y})$ is identical to the empirical distribution $p(\mathbf{y})$, the data has been fit perfectly, and the value of the cross-entropy loss function is equal to the entropy of the noisy images, $H\left(\mathbf{Y}\right)$.

$$-\mathbb{E}\left[\log p_\theta(\mathbf{Y})\right] = -\mathbb{E}\left[\log p(\mathbf{Y})\right] \tag{31}$$
$$= H\left(\mathbf{Y}\right) \tag{32}$$

In practice, the model will not be able to fit the true distribution exactly, and the average value of the loss function will be greater than the entropy of the noisy images. The gap between the entropy that we are interested in estimating and the cross-entropy loss function is given by the Kullback-Leibler divergence between the empirical distribution and the model distribution:

$$-\mathbb{E}\left[\log p_\theta(\mathbf{Y})\right] = H\left(\mathbf{Y}\right) + D_{KL}(p \parallel p_\theta) \tag{33}$$

The Kullback-Leibler divergence is a measure of the difference between two probability distributions. It is always non-negative and is zero only when the two distributions are identical. Thus, the cross-entropy loss function is an upper bound on the entropy of the noisy images. The better our model fits the data, the tighter this bound will be. Finding the right model that balances the accuracy of this bound with the computational cost of fitting the data is an important choice that is discussed further in **Section S3.2.**

In practice, the cross-entropy loss function is evaluated on separate test set of samples from the empirical distribution, to avoid overfitting to a subset of the data and generating a model that is overly optimistic about its performance.

This process is mathematically equivalent to data compression in information theory[3]. In data compression, the goal is to map each outcome to a bit string while minimizing the average string length. Optimal compression assigns shorter bit strings to more probable outcomes and longer ones to less probable outcomes. The achievable compression - measured by the average bit string length - has a fundamental lower bound equal to the entropy of the data distribution. Just as no compression scheme can achieve a shorter average bit length than that dictated by the true distribution's entropy, no model we fit can achieve a lower entropy than the true distribution - any mismatch between our model and reality can only increase our entropy estimate.

### S2.5 Probabilistic models

In this section, we describe three probabilistic models $p_\theta(\mathbf{y})$ used to estimate the entropy of noisy measurements via upper bounds, as described in **Section S2.4**. The models offer different tradeoffs between accuracy and computational efficiency.

The simplest model assumes stationary Gaussian statistics, providing fast estimation with minimal data requirements. A more complex full-covariance Gaussian model removes the stationarity assumption, offering improved accuracy while maintaining computational efficiency. The most sophisticated model, based on the PixelCNN neural network architecture, provides the tightest bounds but requires substantially more computation time and training data.

Other models could be used within this framework, provided they allow direct evaluation of likelihood functions. Recent neural network architectures such as transformers [84], normalizing flows [108], and diffusion models [109, 110] offer promising directions for future work.

While generative models are often evaluated by the visual quality of their samples, this metric does not necessarily correlate with cross-entropy performance [111] - though some state-of-the-art models can achieve both [112]. We selected PixelCNN for its demonstrated effectiveness in minimizing test set cross-entropy [55], combined with its relative simplicity and computational efficiency. However, the rapid advancement of generative models suggests that more efficient architectures providing tighter entropy bounds for a given computational efficiency are likely to emerge.

#### S2.5.1 Full Gaussian process

The full Gaussian process model approximates the distribution of noisy images with a multivariate Gaussian distribution, which is specified by a mean vector $\boldsymbol{\mu}$ and covariance matrix $\boldsymbol{\Sigma}$. The mean vector contains $D$ parameters describing the average value at each pixel, while the covariance matrix requires $\frac{D(D+1)}{2}$ parameters to capture all possible correlations between pixels.

We fit this model by directly estimating the mean and covariance from the data.

#### S2.5.2 Stationary Gaussian process

To reduce model complexity and quantify performance over the full field of view with a single scalar quantity, we can make the simplifying assumption that images are stationary stochastic processes. This is the probabilistic analog of assuming constant optical resolution across the field of view [97].

A stationary stochastic process with distribution $p(\mathbf{y})$ is one in which the joint distribution of any set of pixels is invariant to translations across the field of view. For a 1D process (i.e., a single row of pixels), this means that the joint distribution of pixels depends only on their relative locations, not absolute positions.

Mathematically, for any vector of pixels $(y_1, y_2, \ldots, y_D)$, the joint distribution remains unchanged when all pixels are offset by a constant amount $k$:

---

[3]Technically, this is only completely true when the model distribution is discrete, because continuous data cannot be losslessly compressed. For example, it would take an infinite number of bits on a computer to represent an arbitrary real number exactly. But the intuition remains the same.

$$p(y_1, y_2, \ldots, y_D) = p(y_{1+k}, y_{2+k}, \ldots, y_{D+k}) \tag{34}$$

This stationarity assumption constrains our Gaussian model in two ways. First, the mean vector must be constant across all pixels, reducing it from $D$ parameters to just one. Second, the covariance between any two pixels must depend only on their relative positions. For a 1D process, this makes the covariance matrix Toeplitz (constant along diagonals). For 2D processes, it becomes a doubly Toeplitz matrix: blocks of Toeplitz matrices arranged in a Toeplitz pattern (**Fig. S10**). This structure reduces the number of covariance parameters from $\frac{D(D+1)}{2}$ to just $D$.

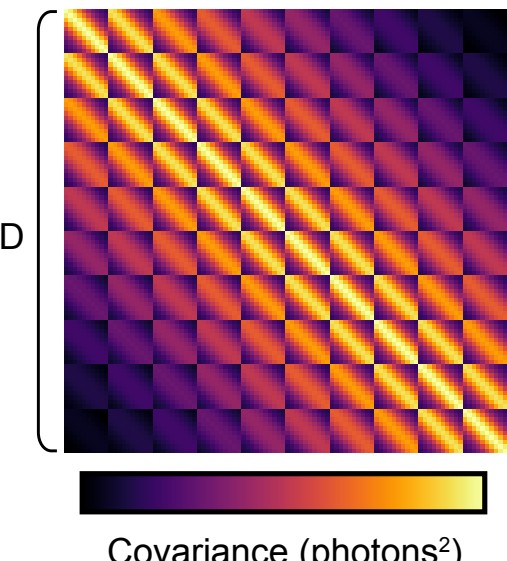

Covariance (photons²)

Figure S10: **The doubly Toeplitz D × D covariance matrix of a stationary 2D Gaussian process.**

Stationary processes have another useful property: a constant entropy per pixel, called the entropy rate [80] (explained in Figure 8 of [90]). This rate can be defined in two equivalent ways:

$$\lim_{D \to \infty} \frac{1}{D} H(y_1, y_2, \ldots, y_D) = \lim_{D \to \infty} \mathrm{H}\left(y_D \mid y_1, y_2, \ldots, y_{D-1}\right) \tag{35}$$

The second formulation reveals why the entropy rate decreases with $D$: as we observe more pixels, it is easier to predict the value at the next pixel, because more context about neighboring pixels is available.

### S2.5.3 PixelCNN

Our most flexible model is the PixelCNN [53–55], which uses neural networks to capture complex pixel dependencies through an autoregressive approach.

Autoregressive models factorize a joint distribution into a product of conditional distributions:

$$p(\mathbf{y}) = \prod_{k=1}^{D} p(y_k \mid y_1, y_2, \ldots, y_{k-1}) \tag{36}$$

$$= p(y_1)p(y_2 \mid y_1)p(y_3 \mid y_1, y_2) \ldots p(y_M \mid y_1, y_2, \ldots, y_{D-1}) \tag{37}$$

This factorization does not require specific assumptions about the joint distribution, so in theory it can be applied to any distribution. Creating the full model requires $D$ conditional distribution models,

each taking between 1 and $D - 1$ previous pixel values as input and outputting a 1D probability distribution for the next pixel. While this could be implemented with $D$ separate models, PixelCNN uses masked convolutions in a single neural network to model all conditionals simultaneously, dramatically reducing computational complexity.

Following previous work [53, 54], our architecture uses a series of masked convolutions to maintain the autoregressive ordering, producing a conditional probability distribution at each pixel.

However, we make one crucial modification: instead of the standard softmax output layer, we use a mixture of Gaussians parameterized by the network (a mixture density network [113]). This strategy is capable in theory of approximating any conditional probability distribution, in the same way that a neural network is capable of approximating any function [114]. This modification is essential for estimating entropy in our framework, because it means the output distributions will be continuous probability densities, instead of discrete probability mass functions. Since our noise models $p(\mathbf{y} \mid \mathbf{x})$ are continuous (Gaussian), our entropy estimates for $H(\mathbf{Y})$ must also use continuous distributions to allow proper subtraction of the conditional entropy $H(\mathbf{Y} \mid \mathbf{X})$.

### S2.6 Sampling and likelihood evaluation

While our models' primary purpose is likelihood evaluation for entropy estimation (**Section S2.4**), generating samples provides valuable insight into model behavior. For simple models like the stationary Gaussian process, sample quality correlates well with likelihood accuracy, making visual inspection a useful diagnostic tool. However, this correlation breaks down for more complex models like PixelCNN [111], though samples remain helpful for understanding model behavior.

Both PixelCNN and stationary Gaussian models are trained on fixed-size patches ($\sqrt{D} \times \sqrt{D}$ pixels), limiting their ability to capture longer-range dependencies. While increasing patch size allows modeling of longer-range correlations, both models can handle arbitrary-sized images despite their fixed patch size. This is possible because their stationary nature allows them to be factored into conditional distributions, enabling evaluation over larger images by sliding the fixed-extent window (though this iterative process is substantially slower than processing native-sized patches). For a 1-dimensional model with $D$ pixels, this factorization means the joint distribution can be written as:

$$p(\mathbf{y}) = \prod_{k=1}^{D} p(y_k \mid y_1, y_2, \ldots, y_{k-1}) \tag{38}$$

$$= p(y_1)p(y_2 \mid y_1)p(y_3 \mid y_1, y_2) \ldots p(y_D \mid y_1, y_2, \ldots, y_{D-1}) \tag{39}$$

Taking the log likelihood turns this product into a sum:

$$\log p(\mathbf{y}) = \sum_{k=1}^{D} \log p(y_k \mid y_1, y_2, \ldots, y_{k-1}) \tag{40}$$

$$= \log p(y_1) + \log p(y_2 \mid y_1) + \log p(y_3 \mid y_1, y_2) + \cdots + \log p(y_D \mid y_1, y_2, \ldots, y_{D-1}) \tag{41}$$

Computing the likelihood over an image larger than the length $D$ model the pixel was trained on can be accomplished by adding additional terms to this sum. For example, computing the log likelihood of an $D + 1$ length image would require adding an additional term to the sum of the log likelihood of the the $D + 1$ pixel conditioned on the previous $D - 1$ pixels (the maximum extent of the model) that preceded it:

$$\log p(\mathbf{y}) = \left( \sum_{k=1}^{D} \log p(y_k \mid y_1, y_2, \ldots, y_{k-1}) \right) + \log p(y_{D+1} \mid y_2, y_3, \ldots, y_D) \tag{42}$$

Details of the likelihood computations for PixelCNNs are described in [53]. For stationary Gaussian processes, there is a closed form solution for finding the mean $\mu_D$ and variance $\sigma_D^2$ of a 1-dimensional

Gaussian distribution conditioned on $D-1$ previous values. This involves decomposing the covariance matrix into a top left $(D-1) \times (D-1)$ block $\boldsymbol{\Sigma}_{1,1}$, a top right $(D-1) \times 1$ column vector $\boldsymbol{\Sigma}_{1,2}$, and a bottom left $1 \times (D-1)$ row vector $\boldsymbol{\Sigma}_{2,1}$:

$$\boldsymbol{\Sigma} = \begin{bmatrix} \boldsymbol{\Sigma}_{1,1} & \boldsymbol{\Sigma}_{1,2} \\ \boldsymbol{\Sigma}_{2,1} & \sigma_M^2 \end{bmatrix} \tag{43}$$

Given $D-1$ previous values $\mathbf{y}_{D-1}$, the mean and variance of the $D$th value can be computed as:

$$\mu_D = \mu + \boldsymbol{\Sigma}_{2,1} \boldsymbol{\Sigma}_{1,1}^{-1} (\mathbf{y}_{D-1} - \mu) \tag{44}$$

$$\sigma_D^2 = \sigma^2 - \boldsymbol{\Sigma}_{2,1} \boldsymbol{\Sigma}_{1,1}^{-1} \boldsymbol{\Sigma}_{1,2} \tag{45}$$

The likelihood the the $D$th pixel can then be evaluated using the probability density function of a 1D Gaussian distribution with mean $\mu_D$ and variance $\sigma_D^2$.

Similarly, sampling images larger than the patch size on which the models were trained can be accomplished by iteratively sampling each pixel conditioned on the previous $D-1$ pixels.

## S2.7  Stochastic encoders

The deterministic encoder assumption underlying our framework breaks down in two scenarios:

First, in quantum imaging regimes, image formation becomes inherently random and cannot be predicted from Maxwell's equations. These are beyond the scope of our present work.

Second, in systems with *encoder uncertainty*–deterministic but unpredictable variations like illumination fluctuations or mechanical drift that cause the same object to produce different noiseless images across measurements.

The most effective approach to reduce encoder uncertainty is to minimize system variations through careful engineering: building mechanically stable optical systems, using temperature control to reduce thermal drift, and implementing precise electronic control of components like illumination sources. Nonetheless, some level of system variation often remains unavoidable in practice.

Here we describe how our framework can be expanded to stochastic encoders to account for encoder uncertainty.

### S2.7.1  Encoder uncertainty

Stochastic encoders affect information estimates two ways. First, different encoder states preserve different amounts of object information. For example, defocus destroys sharp features in images. Average object information thus depends on the focus distribution.

Second, encoder variations create measurement diversity beyond object variations. With focus drift, measurements vary in both content and sharpness. Without modification, our method would interpret both as object information, thereby upwardly biasing its estimates.

One potential way to adapt our method is using calibration data to identify encoder-induced variations. This would likely require either additional compute or simplifying assumptions to handle the fact the the measurement diversity induced by encoder variations may be object dependent. Mathematical formulation

We model encoder variations with state variable $\mathbf{E}$:

$$(\mathbf{O}, \mathbf{E}) \to \mathbf{X} \to \mathbf{Y}$$

Now $\mathbf{X} = f(\mathbf{O}, \mathbf{E})$ for some deterministic function $f$. For example, for focus drift, $f$ would describe how defocus $\mathbf{E}$ blurs object $\mathbf{O}$ to produce image $\mathbf{X}$.

The quantity of interest in the original deterministic setting is now written with the encoder explicit $I(\mathbf{O}; \mathbf{Y} | \mathbf{e})$.

In the stochastic case, averaging over all encoder states gives:

$$\bar{I}(\mathbf{O}; \mathbf{Y}) = \sum_{\mathbf{e}} p(\mathbf{e}) I(\mathbf{O}; \mathbf{Y}|\mathbf{e})$$

Where $\bar{I}(\mathbf{O}; \mathbf{Y})$ denotes the average object information across all encoder states.

Extending the deterministic information transfer theorem to $\mathbf{X} = f(\mathbf{O}, \mathbf{E})$:

$$I(\mathbf{X}; \mathbf{Y}) = I(\mathbf{O}, \mathbf{E}; \mathbf{Y})$$

Rewriting using the chain rule of mutual information:

$$I(\mathbf{X}; \mathbf{Y}) = \bar{I}(\mathbf{O}; \mathbf{Y}) + I(\mathbf{E}; \mathbf{Y}|\mathbf{O})$$

The first term captures effect one (average object information across encoder states); the second captures effect two (diversity from encoder variations).

**Example: random focal drift** Consider a microscope with mechanical drift causing focus to vary between perfect focus and various degrees of defocus. Compared to a stable, in-focus system, we have $\bar{I}(\mathbf{O}; \mathbf{Y}) < I(\mathbf{O}; \mathbf{Y}|\mathbf{e}_{\text{in-focus}})$ because defocus destroys high-frequency information. Meanwhile, $I(\mathbf{E}; \mathbf{Y} \mid \mathbf{O}) > 0$ because focus variations add to measurement diversity.

As shown in the Deterministic Information Transfer Theorem, a deterministic in-focus system has $I(\mathbf{X}; \mathbf{Y}) = I(\mathbf{O}; \mathbf{Y}|\mathbf{e}_{\text{in-focus}})$. With stochastic focus variations, $I(\mathbf{X}; \mathbf{Y})$ can increase or decrease relative to this baseline depending on which of the two effects dominates.

### S2.7.2 Estimating information with stochastic encoders

When encoders vary stochastically, our method estimates $I(\mathbf{X}; \mathbf{Y}) = \bar{I}(\mathbf{O}; \mathbf{Y}) + I(\mathbf{E}; \mathbf{Y}|\mathbf{O})$. To isolate object information $\bar{I}(\mathbf{O}; \mathbf{Y})$, we must estimate and subtract the encoder contribution.

The central challenge is that $I(\mathbf{E}; \mathbf{Y}|\mathbf{O})$ depends on the object—different objects may reveal encoder variations differently. We propose two approaches:

**Approach 1**: Fix object, vary encoder states, estimate $I(\mathbf{E}; \mathbf{Y}|\mathbf{O} = \mathbf{o})$ for multiple objects. This would be feasible in IDEAL using increased compute, but in experimental systems only possible if we can fix objects while varying encoder states (e.g., imaging a static sample while inducing focus drift).

**Section S4.1** shows an example of non-object information in the measurement.

**Approach 2**: Assume $I(\mathbf{E}; \mathbf{Y}|\mathbf{O}) \approx I(\mathbf{E}; \mathbf{Y})$. Computationally and experimentally simpler, but may give incorrect results if encoder effects are highly object-dependent.

Taking Approach 2 as an example:

$$I(\mathbf{E}; \mathbf{Y}) = H(\mathbf{Y}) - H(\mathbf{Y}|\mathbf{E})$$

Our method already estimates $H(\mathbf{Y})$. To estimate $H(\mathbf{Y}|\mathbf{E})$, we fit conditional models $p_\theta(\mathbf{y}|\mathbf{e})$ instead of $p_\theta(\mathbf{y})$, where $\mathbf{e}$ represents encoder state information. For focus drift, conditioning on timestamps would help predict measurements since focus varies smoothly with time.

For Approach 1, we would additionally need to take an outer expectation over objects. In IDEAL, since we know the true object, the models could be modified to condition on it directly. In experimental conditions, we would likely need static objects to take multiple measurements of.

By estimating both terms we can recover the quantity of interest, object information in the measurement.

$$\bar{I}(\mathbf{O}; \mathbf{Y}) = I(\mathbf{X}; \mathbf{Y}) - I(\mathbf{E}; \mathbf{Y}|\mathbf{O})$$

## S2.8 Task-specific information

While information estimates strongly predict reconstruction performance, they may correlate less strongly with specialized tasks that rely only on specific features of measurements. Systems that preserve specific task-relevant information can outperform those capturing more total information indiscriminately [31, 76]

Total information naturally decomposes into task-specific and task-irrelevant components: $I(\mathbf{Y}; \mathbf{O}) = I(\mathbf{Y}; \mathbf{T}) + I(\mathbf{Y}; \mathbf{O}|\mathbf{T})$, where $\mathbf{T}$ represents task variables like classification labels. This enables weighted objectives $\alpha I(\mathbf{Y}; \mathbf{T}) + \beta I(\mathbf{Y}; \mathbf{O}|\mathbf{T})$ for specialized applications:

- **Balanced systems** ($\alpha, \beta > 0$): Control relative importance of task-specific versus general information

- **Compressed sensing** ($\alpha > 0, \beta < 0$): Maximize task-relevant while suppressing task-irrelevant information for bandwidth-limited applications

- **Privacy-preserving** ($\alpha < 0, \beta > 0$): Suppress task-specific information while preserving general scene information

Estimation requires computing $I(\mathbf{Y}; \mathbf{T}) = H(\mathbf{Y}) - H(\mathbf{Y}|\mathbf{T})$. Our existing approach estimates $H(\mathbf{Y})$, while $H(\mathbf{Y}|\mathbf{T})$ can be estimated by conditioning models on task information. **Section S4.2** provides experimental evidence of when information estimates do not fully predict task performance.

# S3 Estimating information: Experiments

Having developed a framework for estimating information in imaging systems, we now validate its performance through experiments on both simulated and real-world data.

## S3.1 Fitting stationary Gaussian processes

Fitting a stationary Gaussian process to $N$ image patches ($\sqrt{D} \times \sqrt{D}$ pixels) requires estimating both the mean vector $\boldsymbol{\mu}$ and covariance matrix $\boldsymbol{\Sigma}$. While the mean is easily computed by averaging all pixels across patches, estimating the covariance matrix presents a greater challenge.

As described in **Section S2.5.2**, the covariance matrix must be both symmetric positive definite and doubly Toeplitz, with repeated patterns along diagonal blocks. Our initial approach was to compute the sample covariance matrix from vectorized patches and enforce the Toeplitz structure by averaging along diagonals and blocks. However, this averaging operation sometimes produced matrices with negative eigenvalues, violating the positive definite requirement and creating invalid covariance matrices.

To address this issue, we initially tried enforcing a minimum positive value (floor) for the eigenvalues. However, this approach proved problematic for two reasons. First, modifying eigenvalues disrupted the doubly Toeplitz structure, violating the stationarity requirement. More critically, since the likelihood evaluation used for entropy estimation is highly sensitive to small eigenvalues in the covariance matrix, the arbitrary choice of eigenvalue floor significantly impacted our entropy bounds. This sensitivity varied across datasets, making the approach unreliable for comparing different imaging systems.

To overcome these limitations, we developed an iterative optimization procedure that improves the covariance matrix fit while maintaining required properties. Given $N$ image patches (each $\sqrt{D} \times \sqrt{D}$ pixels) vectorized to length-$D$ vectors $\mathbf{y}^{(i)}{}_{i=1}^{N}$, we minimize the negative log likelihood of a multivariate Gaussian distribution $\mathcal{N}(\cdot; \hat{\boldsymbol{\mu}}, \boldsymbol{\Sigma})$:

$$\hat{\boldsymbol{\Sigma}} = \underset{\boldsymbol{\Sigma}}{\arg\min} - \sum_{i=1}^{N} \log \mathcal{N}(\mathbf{y}^{(i)}; \hat{\boldsymbol{\mu}}, \boldsymbol{\Sigma}) \tag{46}$$

Where $\hat{\boldsymbol{\mu}}$ is the fixed mean vector estimated from the data as described above.

We solved this optimization problem using proximal gradient descent with momentum, initializing from the direct sample estimate. Each iteration consisted of three steps:

- Computing the eigendecomposition of the covariance matrix and applying gradients to eigenvalues while keeping eigenvectors fixed
- Applying a proximal operator that enforced the doubly Toeplitz structure by averaging along diagonals and blocks
- Enforcing the minimum eigenvalue constraint

The procedure was regularized through early stopping: optimization terminated when the loss hadn't decreased for a fixed number of iterations, with parameters from the lowest-loss iteration selected as the final estimate.

Empirically, we found that the optimization procedure was unstable and prone to divergence. This seemed to be because the likelihood was very sensitive to small eigenvalues of the covariance, which tended to produce extremely large gradients for these eigenvalues. To account for this, we implemented gradient clipping to limit the magnitude of the gradients for the eigenvalues.

The likelihood of a probability distribution can be very sensitive to small changes in parameters because the Euclidean distance between parameter vectors does not always accurately reflect the dissimilarity of the resulting distributions (as described in section 2.3 of [115]). In theory this can be corrected by multiplying the gradient by the inverse of the Fisher information matrix to reorient it from the steepest direction in Euclidean space to the steepest direction in Riemannian space, which accounts for the natural geometry of the parameters. However, in practice we found this to be expensive to compute without obvious performance benefits over gradient clipping.

**Figure S11** shows the results when fitting stationary Gaussian processes to images from the BSCCM dataset [75]. The direct estimate contained several negative eigenvalues, which were then set to an arbitrary eigenvalue floor. The optimization procedure was able to correct these negative eigenvalues, and produced robust results for a variety of settings of the eigenvalue floor parameter (provided it was smaller than the true minimum eigenvalue) (**Fig. S11b**). As a consequence of the small, incorrect eigenvalues in the direct estimate, samples produced from the model that were larger than the patch size on which they were trained (**Sec. S2.6**) exhibited numerical instability that created large oscillation in the samples. In contrast, samples from the optimized model were stable and did not exhibit this oscillation (**Fig. S11b**). The optimization quickly converged in a few iterations (**Fig. S11c**), taking $\sim 3$ seconds to complete on an NVIDIA GeForce RTX 3090 GPU. Comparing estimates of a non-stationary Gaussian process, the direct estimate stationary Gaussian process, and the optimized stationary Gaussian process showed that the optimized stationary Gaussian process produced more robust results in terms of its eigenvalue distribution using smaller datasets (**Fig. S11d**).

### S3.2    Comparing stationary Gaussian process and PixelCNN estimates

We evaluated model performance through two approaches: negative log likelihood on held-out test data and qualitative assessment of model-generated samples.

An important consideration for both models is the choice of training patch size. Larger patches capture longer-range dependencies and improve per-pixel likelihood, but require more training data and computational resources. We therefore sought to determine the minimum patch size that would provide reliable entropy estimates while maintaining computational efficiency.

**Figure S12** shows samples produced when fitting models to patches of different sizes. As expected, larger patch sizes were able to capture more complex statistical relationships between pixels (**Fig. S12a**), decrease the per-pixel loss (**Fig. S12b**), and more tightly upper bound the entropy of the noisy images (**Fig. S12c**). The gains in performance were minimal beyond a patch size of $35 \times 35$ pixels for the Gaussian model, while the PixelCNN was able to continue to produce a more accurate estimate as patch size increased further. This is presumably due to its much greater flexibility in modeling complex dependencies between pixels. However, the magnitude of the gains was small relative to the differences between the estimates of mutual information for different contrast modalities (**Fig. S13**) for these experimental images.

When comparing the samples produced by the two models when fit to images with different illumination patterns on the LED array, both the stationary Gaussian process and PixelCNN models were

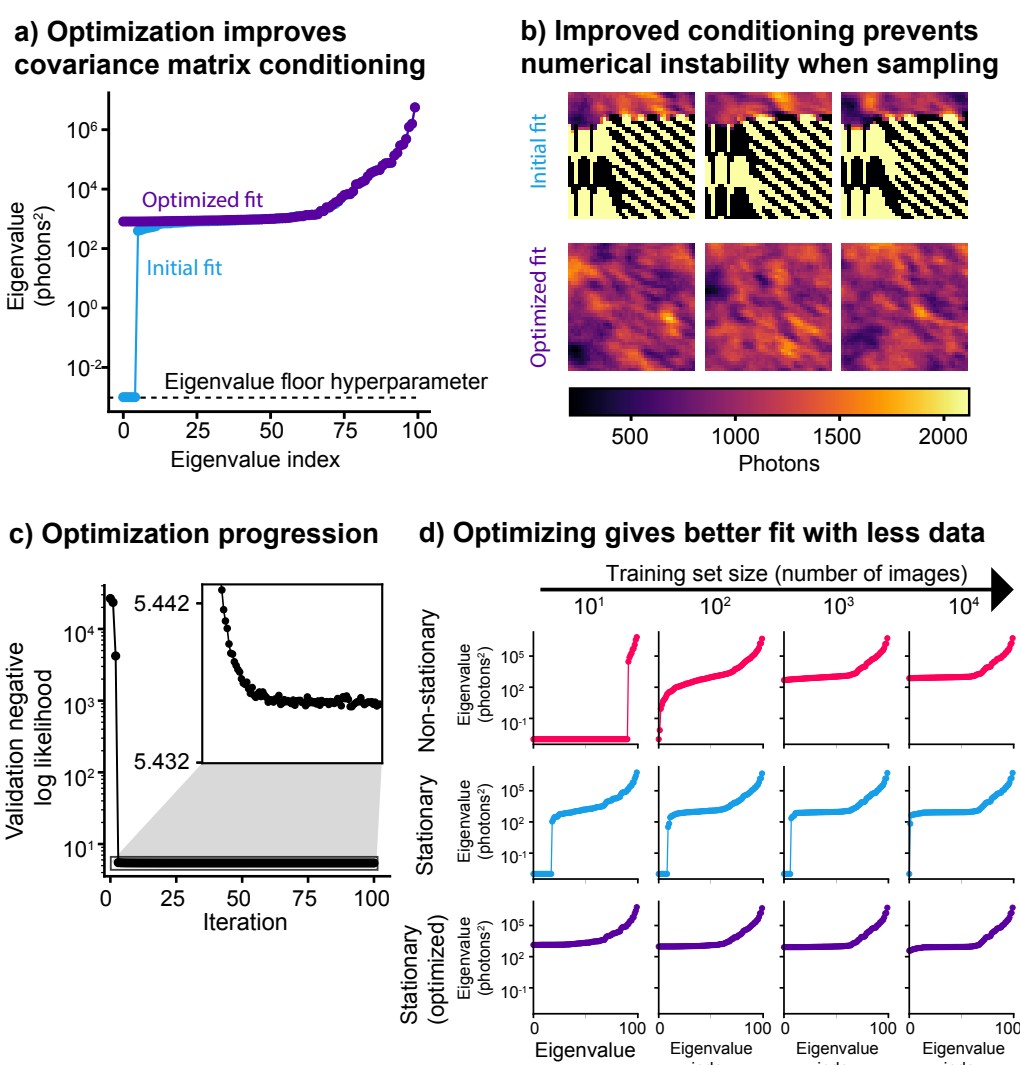

Figure S11: **Fitting Gaussian Process models. a)** Eigenvalues of the covariance matrix before and after optimization, indicating improved conditioning post-optimization. The conditioning of the covariance matrix of non-optimized fits is highly dependent on the choice of the eigenvalue floor hyperparameter. **b)** Comparison of numerical stability during sampling pre- and post-optimization, demonstrating that optimization prevents the instability that manifests in the initial fit. **c)** The optimization process takes only a few iterations to converge and shows an extremely large improvement in the negative log likelihood of the data for poor initial fits. **d)** Efficacy of the optimization across varying training set sizes, illustrating that optimized fitting requires fewer images to achieve a comparable or better fit than the non-optimized approach and outperforms the full Gaussian model when training data is limited.

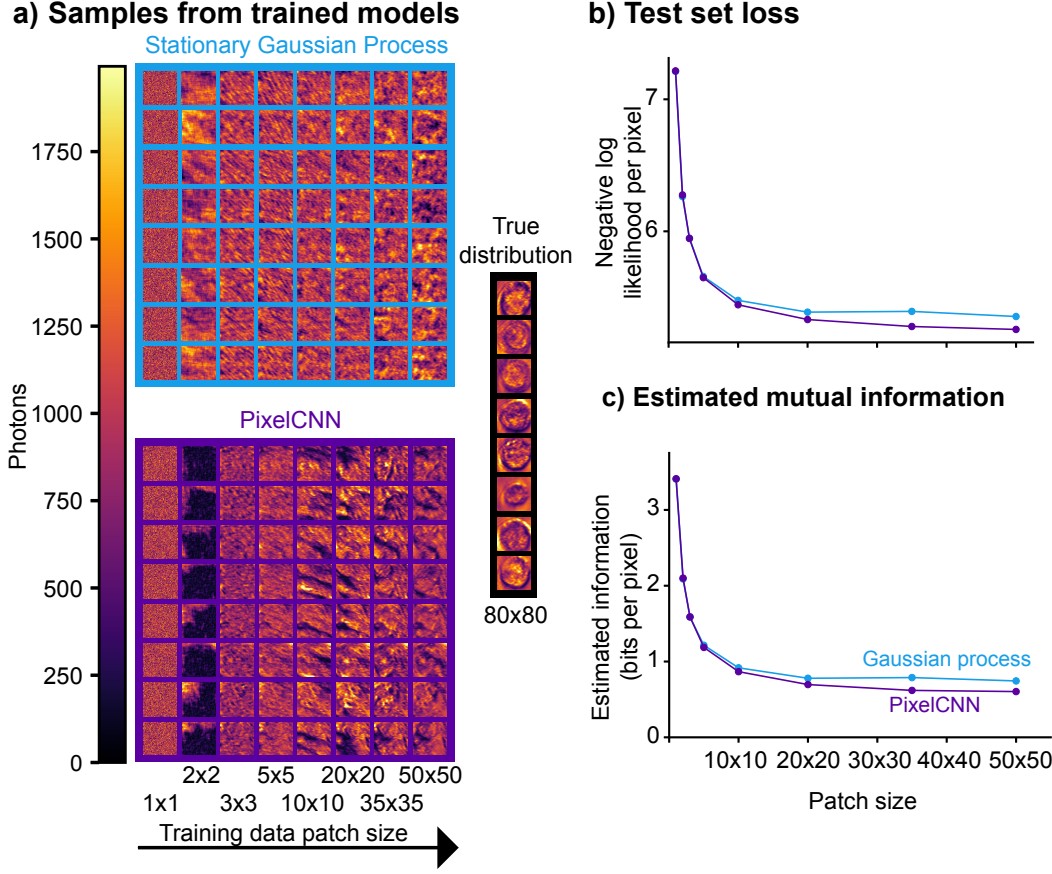

Figure S12: **The effects of patch size on model fit. a)** Samples generated from a Stationary Gaussian Process and PixelCNN, alongside samples from the the true distribution. As the patch size increases, the models are able to capture more long range dependencies between pixels, and **b)** achieve lower negative log likelihood per pixel and thus **c)** a tighter upper bound on mutual information.

able to capture statistical patterns to each type of contrast modality (**Fig. S14a**). Samples from the trained models appear to show that the stationary Gaussian model was able to learn the statistical patterns of the texture of the images of objects under different illumination, while the PixelCNN model additionally learned higher order structures like the edges of cells.

### S3.3  Failures of stationary Gaussian estimates on highly non-Gaussian data

The comparable performance of stationary Gaussian and PixelCNN models on the BSCCM dataset (single cells under coded-angle illumination) likely reflects the near-Gaussian statistics of these images. However, this similarity cannot be assumed for all datasets. This becomes evident when testing on the MNIST handwritten digits dataset [116] (GNU General Public License), where image statistics deviate significantly from Gaussian distributions due to the nearly bimodal nature of pixel values (predominantly 0 or 1).

To demonstrate this, we take the MNIST dataset and simulate a minimal optical encoder (i.e. a single lens) by convolving the images with a Gaussian point spread function. We then add Poisson noise to simulate noisy measurements, and fit both the stationary Gaussian process and PixelCNN models to the data. Sampling from the fitted models shows that the PixelCNN model can produce images that appear similar to true measurements, unlike the Gaussian process model (**Fig. S15a**).

This occurs because the best approximating stationary Gaussian fit is very dissimilar to the true distribution of the measurements. This can be seen by looking at the histograms of image pixels of

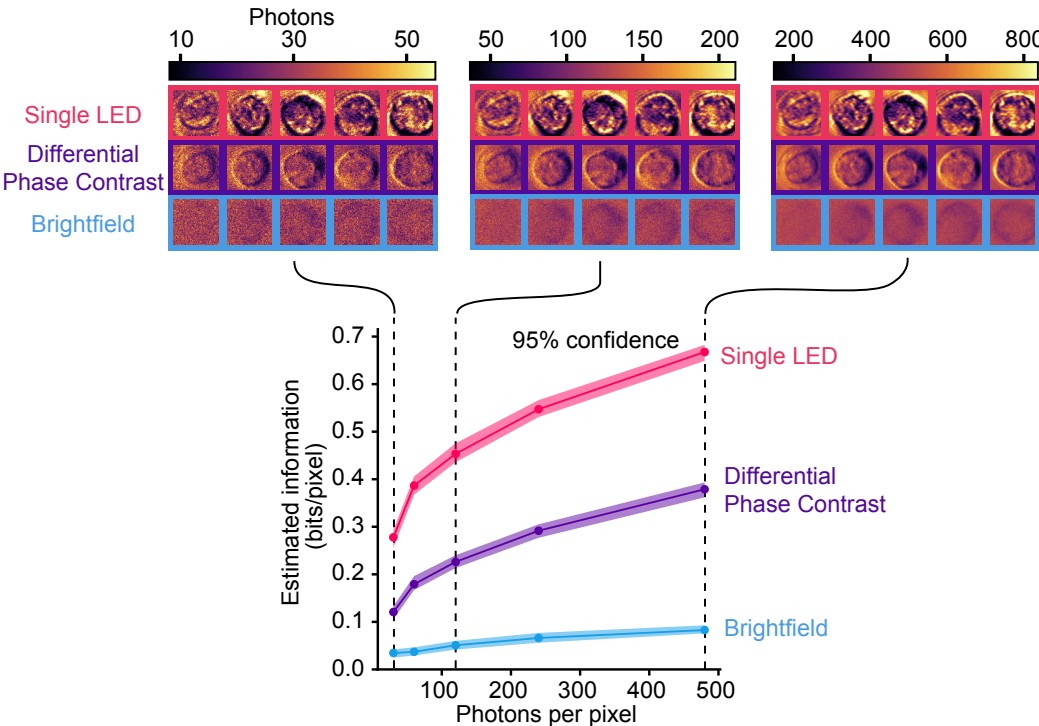

Figure S13: **Mutual information and photon count.** As the average number of photons per pixel increases, the signal-to-noise ratio and mutual information both increase. (Top) Example images of three different contrast modalities at varying photon counts. (Bottom) The mutual information per pixel for each contrast modality as a function of photon count.

the true data and samples from the models, pooled across all many images (**Fig. S15b**). The true data and PixelCNN samples have similar histograms, while the histogram from the stationary Gaussian process samples has a very different shape. As a result of this poor fit, the upper bound given by the mutual information estimator is quite loose for the stationary Gaussian model compared to the PixelCNN model (**Fig. S15c**).

These findings demonstrate that a stationary Gaussian process model is inadequate to capture certain image distributions. Since the tightness of the upper bound in our information estimation procedure depends on accurately fitting the distribution of noisy measurements, this suggests that achieving a tight upper bound/accurate estimate in many cases may require more flexible probabilistic models, like the PixelCNN.

An important and desirable property of an estimator is its consistency: whether it converges to the true value of the parameter being estimated given enough data. This has practical implications for determining how much data is needed to achieve a desired accuracy level.

To evaluate our estimators on high-dimensional data, we first tested them on samples from a known distribution: a stationary multivariate Gaussian with independent additive Gaussian noise at each pixel (Equation 24). In this case, $H(\mathbf{Y} \mid \mathbf{X})$ is constant, and $H(\mathbf{Y})$ can be computed analytically by adding the noise variance to the diagonal of the data covariance matrix and analytically calculating the entropy of the resultant multivariate Gaussian distribution.

**Figure S16a** shows that all three estimators converge to the true value given sufficient samples, with the stationary Gaussian estimator being most accurate for a given sample size. This is expected since the test data perfectly matches the model's assumptions.

To evaluate performance on real microscopy data, we tested two different patch sampling strategies. When patches were taken from fixed locations across different images (**Fig. S16b**), the full Gaussian model outperformed the stationary model. This suggests the presence of location-specific image

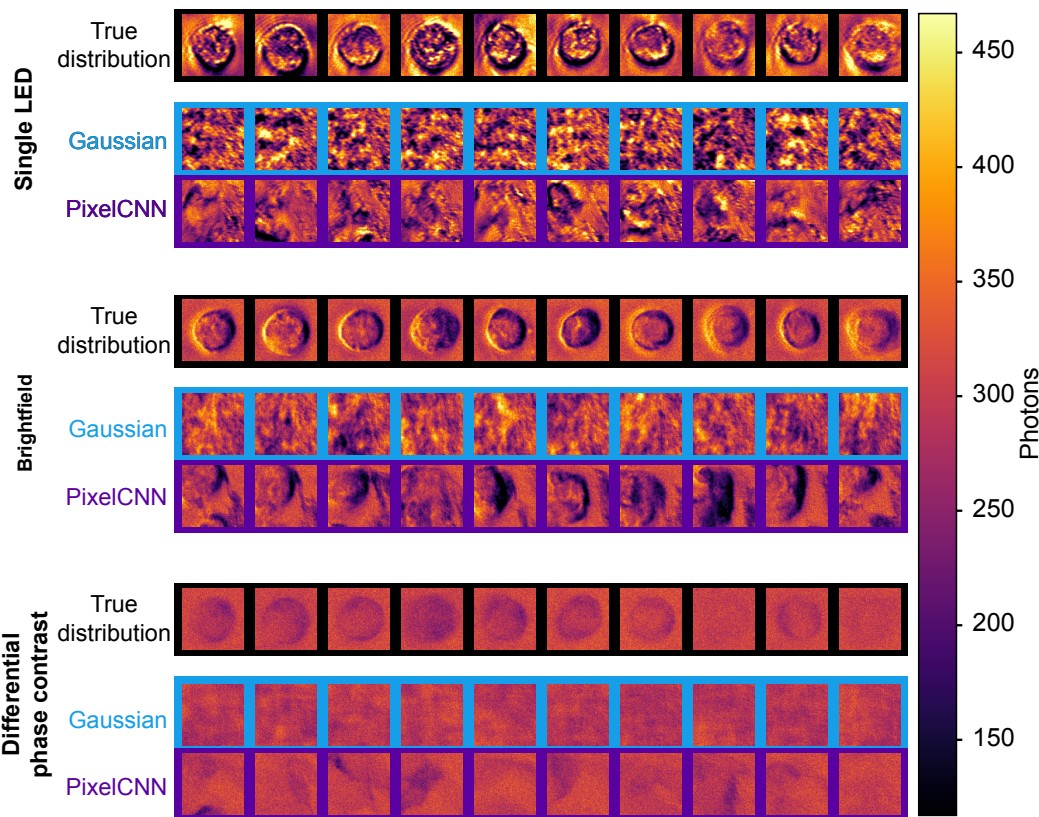

Figure S14: **Model samples for different contrast modalities.** Samples from the stationary Gaussian process and PixelCNN models for Brightfield, Differential phase contrast, and Single LED illumination.

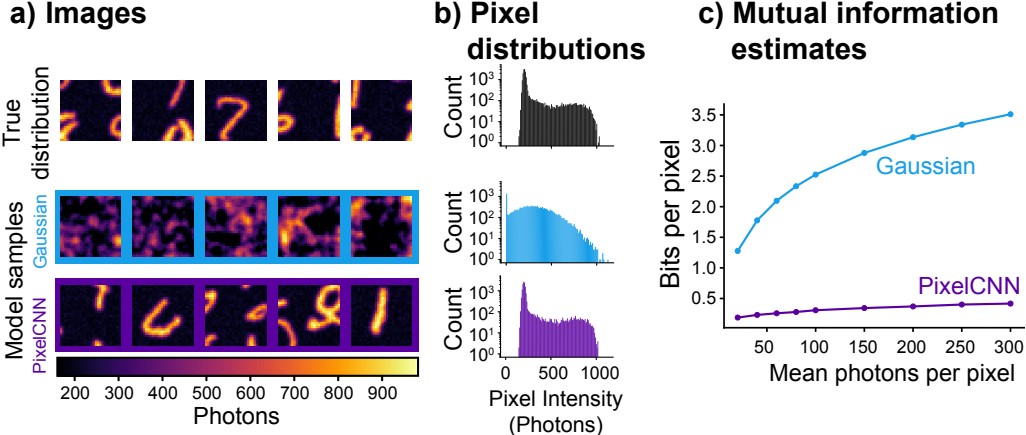

Figure S15: **PixelCNN can fit non-Gaussian data better than stationary Gaussian processes.** **a)** Samples from the stationary Gaussian process and PixelCNN models for MNIST digits. The Gaussian process samples do not resemble the true distribution, indicating a poor fit. **b)** Histograms of pixel values for the true distributions and samples from the two models indicate that the marginal distribution of pixels is non-Gaussian, which the PixelCNN, unlike the Gaussian model, is able to fit. **c)** As a result of the poor fit, the upper bound on mutual information given by the Gaussian process estimator is much looser than the PixelCNN estimator.

statistics that violate the stationarity assumption. However, when patches were sampled from random locations (**Fig. S16c**), the stationary Gaussian model performed better than the full Gaussian model, as the random sampling effectively enforces stationarity in the data distribution.

The PixelCNN model demonstrated good performance across all scenarios due to its flexibility, though the stationary Gaussian model was able to outperform it for small training dataset sizes.

### a)  Stationary Gaussian data

### b) Real data (fixed location patches)

### c) Real data (random location patches)

Figure S16: **Estimator consistency. a)** On simulated data from a stationary Gaussian process, all estimators converge to the true mutual information value, with the stationary Gaussian being most accurate per sample. **b)** With fixed-location patches from real data, the full Gaussian model outperforms the stationary model due to location-specific image statistics. **c)** With random-location patches, the stationary model performs better as sampling enforces effective stationarity in the data distribution.

## S3.4 Model training times

The three models offer different tradeoffs between accuracy and computational efficiency. As shown in **Figure S17**, training times on a typical $20 \times 20$ pixel patch dataset vary significantly: the full Gaussian process is fastest ($\sim 0.1$s), while the stationary Gaussian process requires longer ($\sim 10$s) due to its iterative optimization procedure. The PixelCNN is most computationally intensive ($\sim 100$s), reflecting the cost of its greater modeling flexibility.

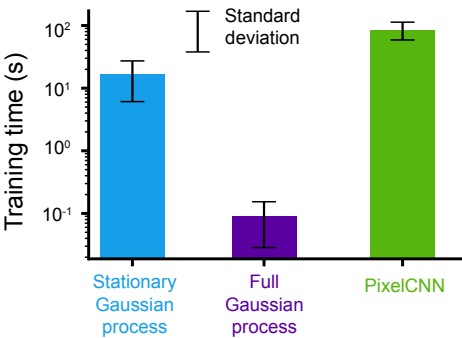

Figure S17: **Model training times.** Comparison of training times for each model type on a typical dataset of 20×20 pixel patches on an NVIDIA RTX A6000 GPU. Error bars show standard deviation across multiple training runs.

## S3.5 Conditional entropy estimates on noisy data

The accuracy of mutual information estimates depends on both $H(\mathbf{Y})$ and $H(\mathbf{Y} \mid \mathbf{X})$. For additive Gaussian noise, the conditional entropy is independent of the noiseless images and can be computed analytically (**Sec. S2.3.2**). However, for signal-dependent Poisson noise (**Sec. S2.3.3**), estimation becomes more challenging.

Using simulated data derived from the BSCCM [75] dataset, we first examined the consistency of our conditional entropy estimator. We created ground truth data by applying a $3 \times 3$ median filter to produce noiseless images, then added simulated shot noise. Our estimator rapidly converges to the true value (**Fig. S18a**), with variations on the order of $10^{-2}$ differential entropy per pixel - significantly smaller than the variations in $H(\mathbf{Y})$ estimation.

In experimental settings, noiseless images are typically unavailable. Instead, we must estimate conditional entropy directly from noisy measurements, replacing noiseless pixel values in Equation 28 with noisy ones:

$$\mathrm{H}\left(\mathbf{Y} \mid \mathbf{X}\right) \approx \frac{1}{N} \sum_{i=1}^{N} \sum_{k=1}^{D} \frac{1}{2} \log_2(2\pi e x_k^{(i)}) \tag{47}$$

$$\approx \frac{1}{N} \sum_{i=1}^{N} \sum_{k=1}^{D} \frac{1}{2} \log_2(2\pi e y_k^{(i)}) \tag{48}$$

To evaluate this approximation, we compared mutual information estimates using either noiseless or noisy images for conditional entropy estimation across different imaging modalities (**Fig. S18b**). The estimates show good agreement except at very low photon counts (below $\sim$40 photons per pixel). At these low counts, using noisy measurements leads to slight overestimation of mutual information due to underestimation of conditional entropy. While this bias could potentially be corrected, other approximations in our framework (such as the Gaussian approximation to Poisson noise (**Sec. S2.3.3**)) also break down in this regime.

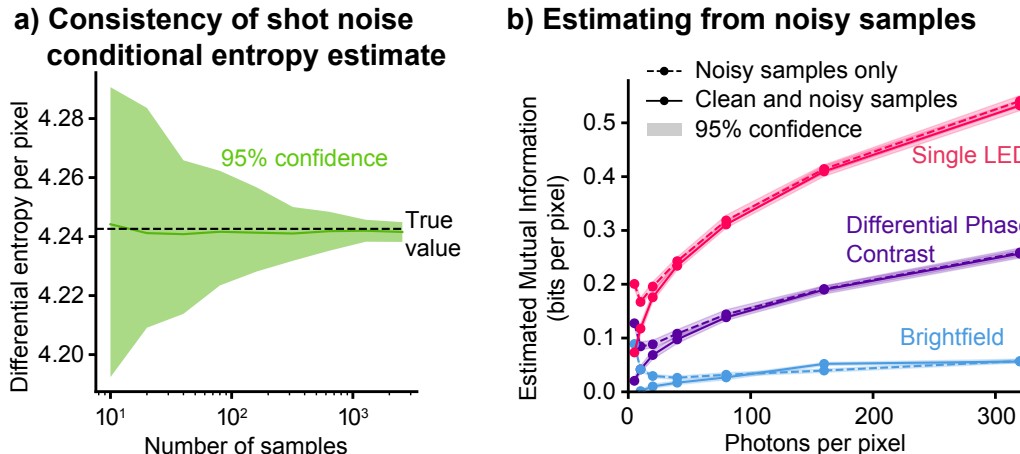

Figure S18: **Conditional entropy estimation. a)** The estimator converges rapidly to the true conditional entropy for shot noise as sample size increases, with variations much smaller than those seen in entropy estimation (**Figure S16**). **b)** Comparison of mutual information estimates using noisy versus noiseless measurements for conditional entropy estimation across different imaging modalities, showing good agreement except at very low photon counts.

## S3.6 Comparing analytic and upper bound entropy estimates

For Gaussian models, we can estimate entropy in two ways: by computing an upper bound using test set negative log likelihood, or by analytically calculating the entropy of the fitted model. While the upper bound approach provides theoretical guarantees, the relationship between analytic estimates and true entropy is less clear.

We compared both approaches empirically across three imaging modalities in the BSCCM dataset using three estimators: analytic stationary Gaussian entropy, stationary Gaussian upper bound, and PixelCNN upper bound (**Fig. S19**). While we lack theoretical guarantees for the analytic stationary Gaussian estimates, they consistently track close to the PixelCNN bounds, which represent our best estimate of the true entropy. The differences between estimation methods remain small compared to the variations across imaging modalities and photon counts. This close agreement is particularly relevant for our IDEAL framework (See **Encoder design via information maximization with IDEAL**), where we use the analytic form in our optimization loss function.

## S4 Additional decoder experiments

### S4.1 Experimental evidence of stochastic encoder effects

We examined LED array microscopy with different illumination patterns to demonstrate how sensitive systems capture both object and system information. By estimating mutual information on empty image patches, we measured information arising purely from system variations.

As shown in **Figure S21**, single-LED illumination exhibited higher background information than differential phase contrast or brightfield illumination. This demonstrates that more sensitive encoders capture more system information alongside object information. When substantial encoder uncertainty is present, the theoretical framework from **Section S2.7** may be needed to separate these contributions.

### S4.2 Task-specific information in classification

For tasks like deconvolution that aim to recover full image structure, any additional information helps reduce reconstruction error. However, simpler tasks like classification may only need specific features. We tested this using a 10-class object classification task on our lensless imaging measurements (**Fig. S22**). Unlike deconvolution, where higher information consistently predicted better

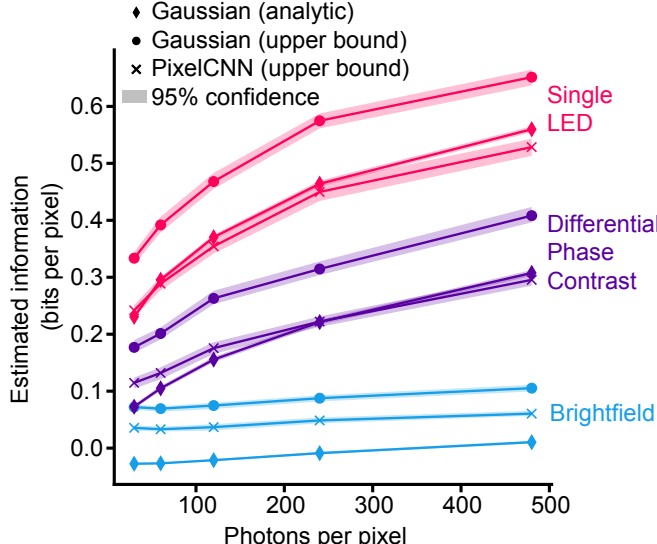

Figure S19: **Comparison of entropy estimation methods.** Three approaches to entropy estimation - analytic stationary Gaussian, stationary Gaussian upper bound, and PixelCNN upper bound - across different illumination patterns and photon counts. The error bars for the upper bound methods (Gaussian and PixelCNN) are computed by bootstrapping both the marginal and conditional entropy estimates from the test set. For the analytic estimates, multiple Gaussian models were fit to bootstrapped versions of the full dataset to estimate $H(\mathbf{Y})$, while $H(\mathbf{Y} \mid \mathbf{X})$ was bootstrapped as before. While the relationship between analytic estimates and true entropy lacks theoretical guarantees, they remain close to our best estimates across conditions.

performance across all encoders, classification accuracy varied significantly between encoders with similar total information content.

This disparity arises because classification requires only the information that distinguishes between classes. Most of the measured information ($\sim$307 bits for our 32×32 pixel images at 0.3 bits/pixel) captures within-class variations irrelevant to the classification task, which requires at most $\log_2 10 \approx 3.32$ bits. Different encoders may preserve this class-discriminative information more or less effectively, even while capturing similar total information.

## S5   Comparison to other frameworks

Mathematical assumptions about object structure underlie several established frameworks for imaging system design. Like our approach, estimation theory [117] and compressed sensing [118] offer ways to design imaging systems without requiring ground truth data or human-interpretable measurements. Estimation theory assumes objects can be described by a few key parameters and uses the Cramér-Rao lower bound [119, 120] to optimize system performance. Compressed sensing assumes objects are sparse in some basis, enabling efficient measurement with theoretical guarantees.

These frameworks take a deductive approach: they start with specific assumptions about objects, encoders, and decoders to prove performance guarantees. While powerful, this approach has inherent limitations. The theoretical guarantees only apply when all assumptions are met—a condition that can be difficult or impossible to verify in practice. Moreover, these frameworks often focus on best or worst-case scenarios for individual objects, which may not reflect the average performance over a range of objects.

Our information-based framework can complement these approaches. Like estimation theory and compressed sensing, it can be used deductively to derive theoretical limits when specific assumptions hold, as demonstrated in our analysis of two-point resolution (See **Probabilistic two-point resolution**). However, it also offers a unique advantage: the ability to work inductively from experimental data. By estimating information content directly from measurements, our approach automatically

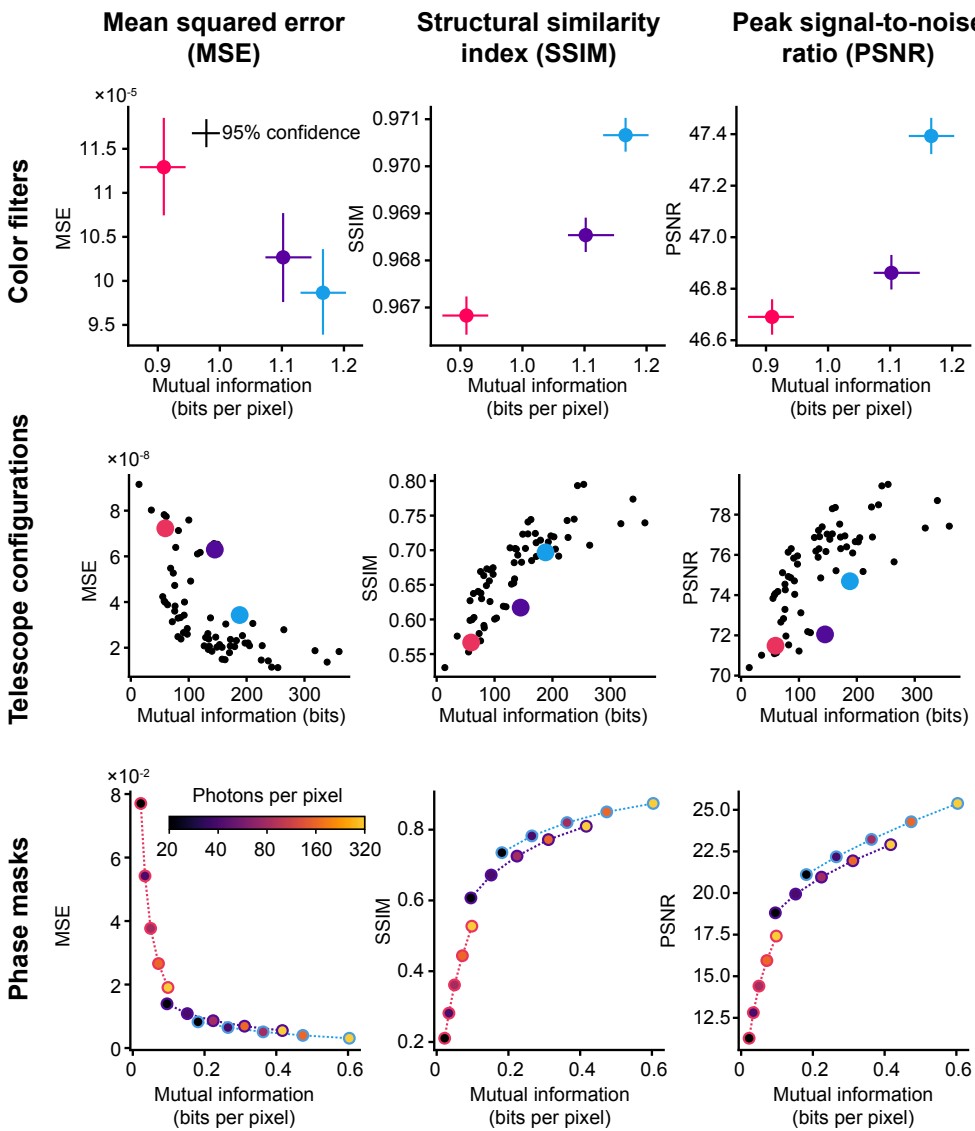

Figure S20: **Performance across multiple metrics correlates with information estimates.** Comparison of mean squared error (MSE), structural similarity index (SSIM), and peak signal-to-noise ratio (PSNR) for the color photography, radio telescope, and lensless imaging systems shown in **Fig. 2a-c**. For the radio telescope configurations (middle row), colored dots correspond to the three example configurations shown in the main text, while black dots show additional tested configurations. Each metric shows consistent correlation with information estimates across different imaging modalities.

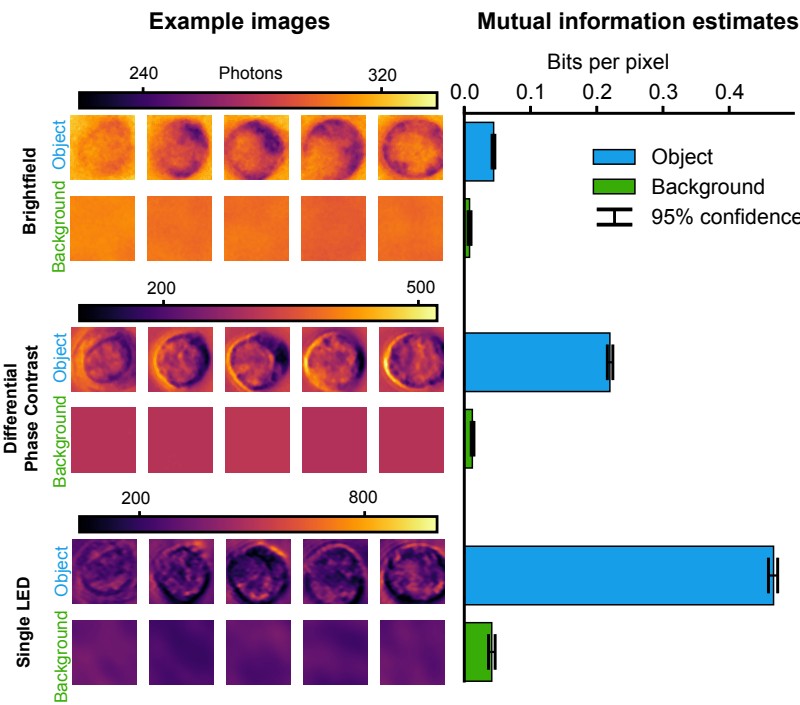

Figure S21: **The effect of stochastic encoders on information estimates.** By computing mutual information estimates on empty image patches with no objects in them, the information about the system present in the measurements can be estimated. This can be seen by comparing Single LED, Differential Phase Contrast, and Brightfield illumination patterns on an LED array microscope, and estimating the mutual information between noiseless image and noisy measurement with and without an object. Encoders that capture more information about the object also capture more information about the system.

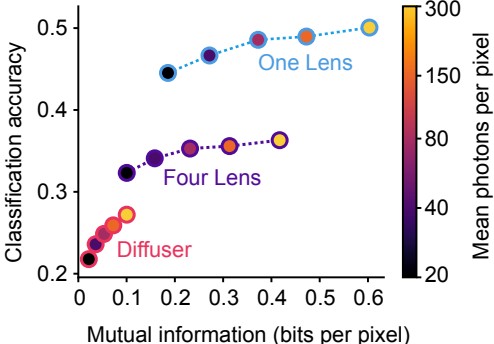

Figure S22: **Classification accuracy vs. information content in lensless imaging.** Unlike deconvolution **(Fig. 2c)**, classification performance does not show a consistent relationship with total information content across different optical designs, suggesting the importance of preserving task-specific information. Points represent median over five classification networks with different random weight initializations.

adapts to the actual properties of objects being imaged, without requiring explicit mathematical models of the imaging process. The only assumption we rely on is a noise model, which itself could be learned from data rather than specified analytically.

This flexibility enables novel capabilities: comparing different imaging modalities, identifying subtle performance tradeoffs, and understanding complex relationships between encoding physics and measurement quality. By learning from data rather than relying on predetermined assumptions, our framework can evaluate and optimize a broader range of imaging systems.

Detailed comparisons between our approach and these established frameworks are provided in **Sections S5.1** and **S5.2**.

## S5.1   Estimation theory and the Cramér-Rao lower bound

Estimation theory provides tools for optimizing imaging systems by minimizing measurement uncertainty. Its central tool, the Cramér-Rao lower bound [117], defines the minimum achievable error when estimating object parameters from noisy measurements. This bound enables systematic comparison of different optical designs by identifying those with the lowest theoretical error.

While both estimation theory and our information-based framework aim to reduce measurement uncertainty (**Sec. S5.1.4**), our approach offers several key advantages:

- **Data-driven adaptation**: Rather than requiring explicit mathematical models, our framework learns system behavior directly from measurements, automatically capturing complex optical effects.
- **Task flexibility**: Estimation theory requires expressing imaging goals as parameter estimation problems (e.g. modeling point source localization in terms of position coordinates). Many practical imaging tasks lack such tractable parameterizations.
- **Minimal assumptions**: We require only a noise model, which can be measured empirically, reducing the risk of model misspecification.
- **Decoder generality**: The Cramér-Rao bound traditionally applies only to unbiased estimators, which are rarely optimal in practice. While extensions to biased estimators exist, they introduce additional complexity without clear benefits (**Sec. S5.1.3**). Our framework places no restrictions on decoder type.

Our framework's flexibility allows it to operate in both deductive and inductive modes. When analytical object models are available, it can provide theoretical guarantees similar to estimation theory. However, it can also learn directly from experimental data when such models are impractical. This dual capability enables our framework to handle a wider range of imaging scenarios, from well-modeled systems to those with complex, difficult-to-model behavior.

### S5.1.1   Background: The Cramér-Rao lower bound in imaging

The Cramér-Rao lower bound has guided the design of numerous imaging systems. Its most successful application has been in fluorescence microscopy, where it has helped optimize point spread functions for localizing single molecules [121, 119, 122–124, 120, 125]. Similar approaches have proven valuable in phase microscopy [126] and wavefront sensing [127].

At its core, the Cramér-Rao bound provides a fundamental limit on measurement precision. Consider measuring a single parameter $\theta$ (such as a particle's position) from noisy data $X$. The measurement process is described by a probability distribution $p(x; \theta)$, which captures both the imaging physics and measurement noise. For instance, in fluorescence microscopy, this distribution might describe how likely different intensity patterns are when imaging a point source at position $\theta$.

The bound states that any unbiased estimator $\hat{\theta}(X)$ (one that gets the right answer on average) must have a mean squared error greater than or equal to the inverse of the Fisher information:

$$\mathbb{E}\left[(\hat{\theta}(X) - \theta)^2\right] \geq \frac{1}{I_F(\theta)} \tag{49}$$

where $I_F(\theta)$ is the Fisher information of $\theta$, defined as:

$$I_F(\theta) = E\left[(\nabla_\theta \log p(x;\theta))^2\right]$$

For many common distributions, the Fisher information behaves like the inverse of the distribution's variance. Intuitively, this means that noisier measurements (higher variance) make it harder to accurately estimate the true value of $\theta$.

This mathematical framework has direct practical applications. In single-molecule microscopy, for example, the Cramér-Rao bound quantifies how precisely a point source's position can be determined from an image. By calculating this bound for different optical designs (such as various point spread functions), researchers can systematically identify and optimize systems that enable more precise localization.

It's important to note that Fisher information, despite sharing the word "information," is distinct from Shannon's mutual information and entropy used elsewhere in this work. While both concepts quantify aspects of measurement uncertainty, they arose from different fields—estimation theory and information theory—and were developed independently. Though theoretical connections exist between them in certain cases [117], they serve different purposes in analysis and design.

### S5.1.2  Cramér-Rao lower bound-based design and its limitations

The approach of designing physical systems based on the Cramér-Rao lower bound has inherent limitations that prevent its application outside of a narrow set of applications in which the decoding problem is limited to estimation of one or a few parameters. Furthermore, even when it can be applied, it requires developing a complete mathematical model of the physics of image formation with respect to the parameter of interest. These limitations have thus far prevented its application outside a narrow range of problems, which include estimation of the position or mass of a single particle [121, 119, 122–124, 120, 128], or estimating a single optical aberration such as defocus [127]. Below, we discuss these limitations in more detail.

**The difficulties of creating parametric models**  Creating parametric models is essential for calculating the Cramér-Rao lower bound, but it becomes challenging when dealing with high-dimensional objects due to the difficulty of parameterizing objects in a way that captures important aspects of decoders, while also retaining computational tractability.

The computation of the Cramér-Rao lower bound requires calculation of Fisher Information, which in turn requires a parametric model of the data distribution, denoted as $p(x;\theta)$. This parametric model stipulates the probability of a specific outcome $x$, given a certain parameter $\theta$.

In certain scenarios, $p(x;\theta)$ can be analytically defined. For example, in the point localization problem, $\boldsymbol{\theta}$ denotes a 3-dimensional vector that marks the XYZ location of a point emitter (the bold type indicates it is a vector of parameters), and $x$ denotes the intensity of a specific pixel. Given known equations for the system's noise characteristics and image formation mechanism, $p(x;\boldsymbol{\theta})$ can be precisely defined. This can be used to compute Fisher Information and the Cramér-Rao lower bound for a single pixel, and the process can be repeated over each pixel $x$ to get an averaged bound.

However, creating a straightforward equation-based model for imaging most types of objects is usually very challenging due to the difficulty and increasing complexity of adding additional parameters. Most objects being imaged lack the simplicity of a single point emitter, which can be fully described by its XYZ location. One potential way to circumvent this issue is a more general purpose parameterization of the object in which it is represented by a discrete array of pixel values [126]. However, unbiased estimators are rarely used on high-dimensional estimation problems like this. Furthermore, the Cramér-Rao lower bound is in terms of mean squared error, which, when applied to an array of pixels, generally does not effectively capture semantically meaningful information about objects [129].

**Generalizing from the simple case**  In light of the the difficulties of extending this approach to objects with high-dimensional parameterizations, another possibility is to design imaging systems on simple classes of objects with the hope that they will generalize to other classes of objects. Taking such an approach requires making additional assumptions with unknown effects on results. Our experiments in minimal 1-Dimensional simulations show that the object-dependence of imaging systems can be readily demonstrated  (**Sec. S1.3**), which may in part explain why empirical solutions to particle localization problems deviate from theoretical predictions [125].

**Unbiased estimators are usually suboptimal** In addition to the practical difficulties of formulating complex parametric models and computing corresponding Cramér-Rao lower bounds, there remains the limitation that the (standard) Cramér-Rao lower bound quantifies the minimum mean squared error only of unbiased estimators. While constraining an estimator to be unbiased is arguably a reasonable choice for simple, low-dimensional parameters like the location of a single point in 3D space, most estimators used in practice are in fact biased [130, 131] and state-of-the-art methods for solving image processing tasks almost exclusively use biased estimators to achieve high performance.

The bias-variance tradeoff [132] provides a useful perspective: biased estimators, while possessing higher bias, can have lower variance, thereby potentially reducing overall error. Thus, with bias, estimators with better error than the standard Cramér-Rao lower bound can be achieved. A classic illustration of this principle is the James-Stein estimator [133], which improves the estimation of multiple parameters simultaneously by shrinking individual estimates towards a common mean. It is based on the counterintuitive principle that, under certain conditions, an estimator that partially pools the data towards a central value can produce overall estimates that are closer to the true values than those obtained by estimating each parameter independently, especially when dealing with small sample sizes or high-dimensional data. It results in lower average error than the unbiased sample mean approach for Gaussian random variables when the number of dimensions is $\geq 3$.

Furthermore, the advantages of biased estimators are clear on empirical problems, particularly high-dimensional ones. State of the art methods on image-to-image estimation problems like denoising and deconvolution are usually achieved using deep neural networks [134] or with iterative optimization procedures that use regularization to bias the estimates towards certain classes of solutions.

Since computational imaging relies heavily on biased estimators for most image processing tasks, designing systems focused solely on minimizing the error of unbiased estimators, or assessing empirical performance based on this criterion, can provide only a narrow range of guarantees, and it is unclear if the conclusions reached by these guarantees can be generalized to a broader range of applications. This raises of the question of what additional theoretic tools can be used to address this more general case.

### S5.1.3 The challenges of generalizing estimation-theoretic design

This section explores alternative approaches to estimation theory, focusing on the use of biased estimators and Bayesian Cramér-Rao lower bounds to address the limitations of the Cramér-Rao lower bound.

**The biased Cramér-Rao lower bound** Though the (standard) Cramér-Rao lower bound only pertains to unbiased estimators, there are variants and related inequalities that can be applied more broadly.

The first is the biased version of the Cramér-Rao lower bound:

$$\mathbb{E}\left[(\hat{\theta}(X) - \theta)^2\right] \geq \frac{(1 + \nabla_\theta b(\theta))^2}{I_F(\theta)} + b(\theta)^2$$

Where $b(\theta)$ is the bias of the estimator as a function of $\theta$:

$$b(\theta) = \mathbb{E}\left[\hat{\theta}\right] - \theta$$

While this form may appear promising, it is in practice challenging for similar reasons to those described in section S5.1.2. Computing the gradient of the expectation of the estimator in high dimensions is a challenging statistical problem in its own right, and this process would need to be repeated for each value of $\theta$, necessitating another high-dimensional integration. Furthermore, this bound is not universal–it changes depending on the bias of the estimator in question. This makes it more difficult to determine the best performance of an ideal theoretical estimator, and thus more difficult to determine how close a real estimator come to achieving that performance.

**The Bayesian Cramér-Rao lower bound (van Trees inequality)** One way of addressing the limitations of the (standard) Cramér-Rao lower bound for unbiased estimators and its biased estimator

variant can be found by generalizing the estimation problem to consider not just a single value of the parameter $\theta$, but instead consider it to also be random (the upper case $\Theta$ is used to denote the corresponding random variable).

Like the standard Cramér-Rao lower bound, the van Trees inequality (also known as the Bayesian Cramér-Rao lower bound) [135, 130] provides a lower bound on the squared error that can be achieved in a parameter estimation problem. Mathematically:

$$\mathbb{E}\left[(\hat{\theta}(X) - \Theta)^2\right] \geq \frac{1}{\mathbb{E}\left[I_F(\Theta)\right] + J(\Theta)} \tag{50}$$

Compared to the standard Cramér-Rao lower bound (Equation 49), this inequality makes two important changes. First, the Fisher information of a particular parameter value $\theta$ has been replaced with a probability-weighted average over all possible values of $\theta$. Second, there is an additional term $J(\Theta)$, defined as:

$$J(\Theta) = \mathbb{E}\left[(\nabla p(\Theta))^2\right] \tag{51}$$

where $p(\cdot)$ is the probability of a particular value of $\Theta$. This can be approximately understood as quantifying how concentrated the distribution of the random variable $\Theta$ is. The more concentrated the distribution of the parameter $\Theta$ is, the more precisely it can be estimated from noisy measurements. Biasing estimates towards more probable values of $\Theta$ enables estimation error to be lowered on average.

This inequality formalizes an important intuition: The theoretical limit of the average error with which a parameter of interest can be estimated (such as some property of an object being imaged) is dependent upon the distribution of that parameter. Changing the distribution of the parameter can change the theoretical limits of performance, as well as the the form of optimal estimators.

### S5.1.4   Connections between estimation and information theory

Originally, information-theoretic quantities like entropy and mutual information were developed for noise-affected message transmission. However, these tools have significant theoretical links to estimations of the precision of noise-corrupted random variables. Though these connections are mostly known only for simpler analytical cases such as Gaussian random variables, they nonetheless provide insights into the relationships between these fields [117]. Here we highlight some of these connections.

An important insight is that where these connections are recognized, designing imaging systems using either estimation or information measures tends to have similar objectives. However, estimation measures have inherent limitations in their applicability, as previously discussed. In contrast, information theory tools don't share these restrictions, positioning them as potentially universal tools for designing physical imaging systems across a variety of applications. Several known inequalities capture the known relationships between these findings and goals of information and estimation theory.

Much of estimation theory centers on limits defined in terms of mean squared error of signals. While mean squared error has many appealing properties, its shortcomings, particularly in the context of quantifying the perceptual and semantic quality of images are readily apparent [129]. This has, for example, motivated work on alternative ways of quantifying error [136, 137].

**Efroimovich inequality**   In contrast, rate distortion theory, a branch of information theory, can be used to understand the fundamental limits and behavior of a wide variety of loss functions. The connection from the lower bounds used in estimation theory and information theory can be readily seen in the Efroimovich inequality [138], which generalizes the van Trees inequality (Equation 50) from providing a bound on only mean squared error to providing a bound on a more general way of quantifying uncertainty, the entropy of a parameter $\theta$ given a noisy measurement $X$:

$$\frac{1}{2\pi e}e^{2h(\Theta|X)} \geq \frac{1}{\mathbb{E}\left[I_F(\Theta)\right] + J(\Theta)}$$

Here, $h(\Theta \mid X)$ is the differential entropy of a parameter given data $X$. This inequality can be used to derive bounds on loss functions other than mean squared error in terms of information-theoretic quantities [131, 139].

**I-MMSE formula**    Another known relationship with particular relevance to this work is known as the I-MMSE formula (short for Information - minimum mean squared error), which states that [140, 117]:

$$I(X;Y) = I(X; \sqrt{s}X + N) = \frac{1}{2} \int_0^s \mathrm{mmse}(X \mid \gamma X + N)d\gamma$$

In this equation, $X$ is a signal of interest, and $Y = \sqrt{s}X + N$ is a noisy measurement of that signal, created by adding independent Gaussian noise $N$ to the original signal such that the resultant signal-to-noise ratio is $\sqrt{s}$. $\mathrm{mmse}(\cdot)$ is the minimum mean squared error of estimating $X$ given $Y$ (i.e. the Bayesian Cramér-Rao lower bound shown in Equation 50). From this formula, the mutual information $I(X;Y)$ is equal to the minimum mean squared error of the optimal estimator, averaged over all achievable signal-to-noise ratios.

This formula shows that mutual information quantifies the same operational idea as the (Bayesian) Cramér-Rao lower bound in the case of additive Gaussian noise, which strongly suggests that the quantities may serve similar purposes under more general noise models. This relationship has been leveraged to develop task-specific information estimators [31].

This relationship can be visualized in the signal coordinate representation, which provides further intuition as to why these quantities are closely related. **Figure S23** shows the distributions of noisy measurements for 6 different signals with a measurement system that imparts additive Gaussian noise. Since mutual information is operationally defined as the number of signals that can be reliably distinguished in the presence of noise, decreasing the maximum signal-to-noise ratio lowers the mutual information. Simultaneously from the estimation theory perspective, it impedes the ability to estimate the original signal from a noisy measurement of it, because it increases the ambiguity as to which input signal gave rise to the measurement.

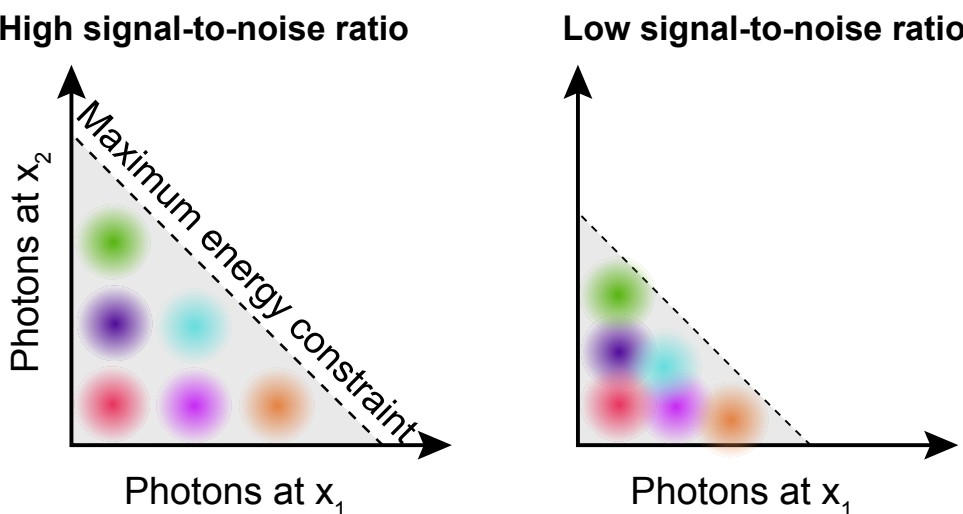

Figure S23: **Visualizing the connection between information and estimation**. Shown are noisy measurements of 6 distinct signals with high and low signal-to-noise rations. With less noise, the signals remain more distinguishable, increasing mutual information. Simultaneously, lower noise reduces uncertainty in inferring the true signal, improving estimability.

This theoretical connection, combined with the developments in the present work that enable estimation of mutual information across many types of imaging systems provides a means of generalizing

the successes of estimation-theoretic design criteria to a wide variety of imaging systems, without the requirements for detailed, system-specific mathematical modeling.

## S5.2 Compressed Sensing: Assumptions and Limitations

Compressed sensing theory showed that certain signals can be accurately reconstructed from fewer measurements than traditionally thought possible. While conventional sampling theory (Nyquist) requires sampling at twice the highest signal frequency, compressed sensing demonstrates that sparse signals—those with few nonzero components in some basis—can be recovered from fewer measurements.

In its original formulation, the framework relies on specific assumptions:

- **Signal sparsity**: Most signal components must be zero in some basis
- **Incoherence:** The measurements of the signal should *not* be sparse. The degree of incoherence between the measurement matrix and the sparsity basis gauges the level to which this requirement is met.
- **Gaussian noise**: Measurement noise must be normally distributed and signal-independent
- **Specific reconstruction algorithms**: Recovery requires optimization methods with provable guarantees

When these assumptions hold, compressed sensing approaches theoretical performance limits [141]. However, real-world imaging often violates these assumptions. Natural objects rarely exhibit perfect sparsity, and measurement noise may not be Gaussian. Studies have shown that compressed sensing strategies become suboptimal when objects don't match the assumed structure [39, 38].

Recent work has shown that incorporating additional object knowledge can improve upon basic compressed sensing. When signals can be generated by passing low-dimensional vectors through nonlinear models, reconstruction accuracy can improve by an order of magnitude compared to traditional compressed sensing approaches [142–144].

More broadly, probabilistic models offer a framework for describing signal structure beyond sparsity. These models enable analysis of average-case performance and can incorporate various forms of prior knowledge about signals [145, 146]. Theoretical work has shown how measurement limits depend fundamentally on the complexity of the objects being measured [147], and how probabilistic approaches can achieve optimal performance for specific signal classes [148, 149].

# S6 Methods

## S6.1 Model training

Memory, computational, and data constraints often made it impractical to estimate information from full images. We therefore randomly sampled fixed-size patches from across the images for training.

**Full Gaussian model**   The full (non-stationary) Gaussian model estimates a complete covariance matrix directly from vectorized image data without assuming stationarity. The mean vector is computed as the average across all training samples. The covariance matrix is estimated using the empirical covariance of the centered data. To ensure numerical stability and prevent degenerate solutions, we enforce positive definiteness by applying an eigenvalue floor - any eigenvalues below this threshold (typically $10^{-3}$) are set to the floor value. If numerical issues persist, the floor is automatically increased until the covariance matrix becomes positive definite. This model offers greater flexibility than the stationary version but requires more training data and computation due to the larger number of parameters. For discrete-valued data, small uniform noise is added during training to prevent infinite likelihoods.

**Stationary Gaussian process model**   The stationary Gaussian process model enforces translation invariance through a doubly Toeplitz structure in the covariance matrix. The model is initialized by computing the empirical covariance matrix from vectorized image patches and averaging along diagonals and block diagonals to impose the Toeplitz constraint. The mean is estimated as the

average pixel value across all patches. We then refine these estimates through an optimization procedure that maximizes the data likelihood. The optimization uses a parameterization based on eigendecomposition, where only the eigenvalues are updated while keeping the eigenvectors fixed. We employ stochastic gradient descent with momentum and gradient clipping to prevent numerical instability. After each gradient step, a proximal operator ensures the eigenvalues remain above a minimum floor value (typically $10^{-3}$) to maintain positive definiteness. The optimization typically runs for 60 epochs with early stopping based on validation likelihood, using batch sizes of 12 images and a learning rate of 100. During both training and evaluation, small uniform noise is added to discrete-valued data to enable proper likelihood computation with the continuous model. The stationary assumption significantly reduces the number of parameters compared to the full Gaussian model, enabling more efficient training and better generalization with limited data.

**PixelCNN model**   The images were modeled using a PixelCNN architecture adapted from prior work [53, 54]. The model consists of vertical and horizontal masked convolution layers arranged in a stack, enabling modeling of the conditional distribution of each pixel given the previous pixels in raster scan order. The network includes seven gated masked convolution layers with alternating dilation rates (1,2,1,4,1,2,1) to increase the receptive field while maintaining computational efficiency. Each gated layer combines information from vertical and horizontal stacks through masked convolutions followed by a gating mechanism using tanh and sigmoid activations.

We modified the original categorical output distribution to instead use a Gaussian mixture density [113] with 40 components at each pixel. The mixture parameters (means, standard deviations, and mixing weights) were computed through separate dense layers. The Gaussian means were initialized randomly uniformly between the training image minimum and maximum values and clipped to this range during inference. The standard deviations were computed as the softplus output of a dense layer, clipped to be between 1 and the training set standard deviation to avoid degenerate solutions.

During training, small uniform noise was added to the discrete-valued images to account for modeling them with a continuous distribution. The noise prevents the likelihoods from going to infinity and overfitting to the exact training values. Input images were normalized by subtracting the mean and dividing by the standard deviation of the training set. The model was trained using the Adam optimizer with a learning rate of 0.01 on image patches, with early stopping based on validation likelihood to prevent overfitting. All models were implemented in JAX/Flax for efficient training on GPUs.

## S6.2   Confidence intervals

Our framework supports estimation of confidence intervals through bootstrapping to quantify uncertainty from finite test set size. For a specified confidence level (e.g., 90%), we repeatedly resample the test set with replacement and recompute both the marginal entropy (using the trained model) and conditional entropy. The mutual information is computed for all combinations of these resampled estimates. The confidence interval is then determined by taking appropriate percentiles of these bootstrap estimates - for example, the 5th and 95th percentiles for a 90% confidence interval. This approach captures uncertainty in both the marginal and conditional entropy estimates.

## S6.3   Sample generation

Images are generated using ancestral sampling, where pixels are drawn sequentially in raster scan order (left to right, top to bottom). For each new pixel, its value is sampled from a probability distribution conditioned on all previously generated pixels within its local neighborhood. For the Gaussian models, this is a Gaussian distribution whose parameters are computed using the Schur complement of the covariance matrix. For PixelCNN, this is a mixture of Gaussians whose parameters are computed by the neural network. To ensure physical plausibility in imaging applications, generated values can optionally be constrained to be non-negative by clipping negative values to zero. While slower than parallel generation methods, ancestral sampling maintains the local statistical dependencies captured by the models while enabling generation of arbitrarily large images.

### S6.4 Generation of synthetic experimental noise

Experimental data was simulated to have been collected with fewer photons by adding simulated photon shot noise. However, since the experimental data already contains some shot noise, it is necessary to determine how noisy the images already are, and then only add additional noise as needed.

To simulate a lower photon count, each pixel with photon count $p$ was multiplied by a fraction $f$ to reduce the photon count to $fp$. Assuming the dominant source of noise in the original image is photon shot noise, the variance of the noise in the original image is approximately $p$, and the variance in the reduced photon count image is approximately $f^2p$. The desired variance of the reduced photon count image is equal to its mean, $fp$. Since the sum of two independent Gaussian random variables with variances $a$ and $b$ is a Gaussian random variable with variance $a + b$, we can add noise to the reduced photon count image to achieve the desired variance. Additional zero-mean Gaussian noise was added with variance $fp - f^2p$. The standard deviation of the added noise was then $\sqrt{fp(1-f)}$.

### S6.5 Applications

#### S6.5.1 Color Filter Array

**Color imaging dataset**   We conducted experiments using the Gehler-Shi dataset [62, 63] (implied license via explicit permission to use), which comprises 568 high-quality natural images. The dataset was partitioned into 461 training, 51 validation, and 56 test images, with each image subdivided into $24 \times 24$ pixel patches. To establish a consistent baseline for illumination, we computed a white channel by summing the red, green, and blue channels, then normalized the dataset to achieve a mean of 1000 photons per pixel in the white channel, based on the training split mean. Signal-dependent Poisson noise was subsequently applied to all patches.

**Information content analysis**   For mutual information estimation, we sampled $100,000$ patches from the test set for fitting a PixelCNN, reserving $10,000$ for evaluation. Conditional entropy calculations were performed on clean images prior to synthetic Poisson noise application. We evaluated three distinct color filter array (CFA) designs: a Bayer pattern, a random configuration, and a learned pattern using the architecture in [61]. For each CFA design, we selected the minimum information estimate across the 20 replicates (since our estimator provides an upper bound) and the best prediction performance (lowest negative log likelihood on test data) to characterize system performance. Each replicate took ~3.5GB of memory. All replicates were trained in ~5 hours on an Nvidia RTX A6000 GPU.

**Neural network architecture and training**   Color image reconstruction was implemented using a bifurcated network architecture as proposed by Chakrabarti [61]. This approach processes $24 \times 24$ pixel measurement patches through two parallel paths. The first path performs multiplicative operations in log-space with linear mixing, while the second employs convolutional layers with 128 filters. Each path generates 24 estimates per color value, which are then combined through learned weights for final reconstruction.

The network implementation features an $8 \times 8$ repeated CFA pattern incorporating four channels (RGB and panchromatic). Training used the Adam optimizer ($\beta_1 = 0.9$, $\beta_2 = 0.999$) with a learning rate of $1 \times 10^{-5}$ and a batch size of 128 patches, running for a maximum of $100,000$ training steps with checkpoints every $5,000$ steps. Training progress was monitored using a validation set comprising $10\%$ of the training patches ($100,000$ patches total). The checkpoint achieving the lowest validation loss was selected for final reconstruction performance evaluation.

#### S6.5.2 Black hole imaging

**Black hole dataset**   For our site selection experiments, we generated a dataset of 100,000 synthetic black holes based on the physical models for black hole dynamics from [150]. Each synthetic black hole had a static envelope, right ascension, and source declination matching that of M87. The dynamic portion forming the accretion disk surrounding each black hole was modeled as a Gaussian random field generated by a different solution to an underlying anisotropic spatio-temporal diffusion partial differential equation with fixed diffusion and advection fields. Each black hole was formed

by applying a Gaussian random field as a multiplicative perturbation to the static envelope, and normalized to have total flux of 1 Jansky. To simulate black hole imaging we selected combinations of four telescopes from the eight original telescope locations used in the Event Horizon Telescope 2017 array. We also included comparisons between reconstruction quality and information estimates for all 70 possible combinations of four telescopes (**Fig. S20**). For each combination we simulated radio telescope measurements including thermal noise using the methods described in [151].

**Information content analysis**    For mutual information estimation from complex-valued radio telescope measurements we used the full Gaussian process model. Each complex-valued measurement with thermal noise was separated into real and imaginary components and concatenated into a single vector before being input to the model. The full Gaussian process was fit with 50,000 measurements and evaluated on a test set of 10,000 measurements with 100 bootstrapped estimates to form a 95% confidence interval. The conditional entropy was calculated based on the thermal noise statistics for each telescope array, modeled as Gaussian noise with independent and varying standard deviations at each measurement sample.

**Imaging inverse problem**    Black hole images were reconstructed from simulated radio telescope measurements using a standard regularized maximum likelihood method with total variation regularization [151], with hyperparameters matching the implementation in [152]. Images were reconstructed for a 2,000 measurement subset of the test set used for information estimation. Each reconstruction was performed on the CPU and took $\sim 10$s.

### S6.5.3   Lensless imaging

**Natural image dataset**    The CIFAR10 dataset [153] (implied license via permission to use) was used for the lensless imaging experiments. This dataset consists of 60,000 total images across 10 classes. To convert CIFAR10 image pixel values to synthetic photon counts, the dataset was scaled to have mean value equivalent to the desired photons per pixel. Encoded measurements for each lensless imaging system were generated by convolving (valid region, no zero padding) with the corresponding point spread function. The point spread functions for the one lens and four lens encoders were modeled as Gaussian-blurred points. For the diffuser encoder, the point spread function is a downsampled version of an experimentally captured point spread function from DiffuserCam [67]. After convolution, synthetic Poisson noise was added to form the noisy encoded measurements.

**Information content analysis**    For mutual information estimation, images from the CIFAR10 dataset were randomly selected in sets of 9 and tiled into 3x3 grids before convolution with the encoding point spread functions. This tiling was implemented to prevent intensity falloff at the edges of a measurement, which occurs when convolving a non-tiled image due to the zero padding necessary to maintain measurement size. With tiling, image content was brought in uniformly at every point in the field of view, forming a spatially consistent texture. Information content was estimated using $32 \times 32$ image patches, the same size as the original non-tiled CIFAR10 images. Patches were randomly sampled over the $65 \times 65$ region corresponding to the valid convolution output and 10,000 patches were used for information estimation. Conditional entropy was calculated using clean images without synthetic Poisson noise. For each encoder and photon count combination, we trained four separate PixelCNN models to estimate information content, each including 100 bootstrap estimates with a test set of 1,500 images to form 95% confidence intervals. We selected the minimum information estimate across the replicated models to characterize system performance. Estimates were consistent across replicates, with less than 0.01 bits per pixel difference across replicates and confidence intervals smaller than the marker size in graphs.

**Image deconvolution**    For the deconvolution task, images from the CIFAR10 dataset were convolved with an encoder. Then, Wiener deconvolution with an automatically tuned regularization parameter was used to reconstruct the original image from each convolved measurement. This was performed on the CPU, taking $\sim 10 - 100$ ms per reconstructuion.

Deconvolution is an ill-posed process and successful reconstructions with Wiener deconvolution require finite image extent. Therefore, instead of random tiling, we use non-tiled images with zero padding for this task. For each encoder and photon count combination, we evaluated image reconstruction metrics on a test set of 1,500 images. The images in this test set correspond to the

center image of each of the $3 \times 3$ tiled measurements used in the information estimation test set. The mean value for each metric was reported and 95% confidence intervals generated from 100 bootstrap estimates with the test set were smaller than the marker size.

**Image classification**    In addition to the deconvolution task in the main paper, we study object classification using the same encoders. A simple CNN architecture, sufficiently powerful for regular CIFAR10 image classification is used. This consists of two convolutional layers (64 and 128 filters respectively with kernel size 5), each followed by a MaxPool, and two densely-connected layers, the first with 128 nodes and a ReLU activation, and the second with 10 layers and a Softmax activation for classification into the 10 classes. Classifiers were trained on an NVIDIA TITAN XP GPU using <5GB memory, ~10 minutes training time per model.

The random tiling process used in mutual information estimation is used for the images in this task as well. The label for classification is based on the center image in the $3 \times 3$ grid, for which maximum image content is present in the measurement. Classification is repeated 10 times for each photon count and encoder combination, from which the 90% confidence interval is generated.

### S6.5.4    LED array microscopy

**Cell imaging dataset**    We analyzed single leukocyte images and corresponding protein expression measurements from the Berkeley Single Cell Computational Microscopy (BSCCM) dataset [75] (CC0 1.0 Universal license). The dataset includes images acquired under multiple illumination conditions using an LED array microscope along with measurements of eight protein markers (CD3, CD19, CD56, CD123, HLA-DR, CD14, CD16, and CD45) obtained through antibody staining. To study performance at different signal levels, we simulated lower photon counts by adding synthetic Poisson noise to the original images.

**Neural network architecture and training**    We developed a neural network to predict protein expression levels from single-cell images. The network uses a DenseNet121 [154] backbone pre-trained on ImageNet, modified to accept single-channel input by averaging the first convolutional layer's RGB weights. The backbone feeds into a global average pooling layer followed by eight parallel fully-connected networks, each with two hidden layers. These networks output parameters for Gaussian mixture models corresponding to each protein marker, enabling direct evaluation of the negative log likelihood for target protein levels. Training used the Adam optimizer with a learning rate of $5 \times 10^{-5}$ and batches of 16 images. A composite loss function summed the negative log likelihood across protein markers, with missing values masked out. Training proceeded for 4,000 steps per epoch and employed early stopping when validation loss failed to improve for 20 epochs. Typical trainings took 1-2 days on NVIDIA TITAN Xp GPUs with 12GB memory. 2-3 networks were trained in parallel sharing a single GPU.

**Information content analysis**    To evaluate system performance, we performed 15 independent train-test splits of the data. For each split, we trained both a PixelCNN model to estimate information content and a protein prediction network. Information content was estimated using $40 \times 40$ pixel patches uniformly sampled from the normalized images. For each illumination condition and photon count, we selected the minimum information estimate across the 15 replicates (since our estimator provides an upper bound) and the best prediction performance (lowest negative log likelihood on test data) to characterize system performance.

### S6.6    Information-Driven Encoder Analysis Learning (IDEAL)

Using the Gehler-Shi dataset [62, 63] partitioned into training (461 images), validation (51 images), and test (56 images) sets, we extracted 100,000 patches of size $24 \times 24$ pixels for training, along with 10,000 patches from the validation set. To establish consistent illumination conditions, the dataset was normalized to achieve a mean of 1000 photons per pixel in the white channel (sum of RGB channels) based on the training set statistics.

The mutual information between noiseless and noisy measurements was calculated using an analytic Gaussian approximation to ensure differentiability of the loss function with respect to the learnable parameters [61]. At each optimization step, we process a batch of 2304 measurement patches ($24 \times 24$ pixels each), selected randomly from the Gehler-Shi dataset. These measurements were vectorized to

compute their covariance matrix, whose log eigenvalues are used to estimate $H(\mathbf{Y})$. For numerical stability, any non-positive eigenvalues were set to $10^{-8}$. The batch size was chosen to be 2304, 4 times the number of pixels in a patch (576 pixels), as we found a batch of at least this size produces the most stable estimation. $H(\mathbf{Y}|\mathbf{X})$ is calculated using the analytic formula detailed earlier (**Sec. 24**).

For the IDEAL optimization, we designed an $8 \times 8$ repeating color filter array (CFA) pattern with four possible filters at each pixel: red, green, blue, and white. To learn which filter to place at each location, we used a temperature-based softmax approach [61]:

$$m(n) = \text{Softmax}[\alpha_t w(n)] \tag{52}$$

Here, $m(n)$ is a 4-element vector that represents the probability of selecting each color filter at pixel location $n$. The learnable weights $w(n)$ determine these probabilities through a softmax function whose sharpness is controlled by the temperature parameter $\alpha_t = 1 + (\gamma t)^2$, where $t$ is the training iteration and $\gamma = 0.001$. Early in training when $\alpha_t$ is small, the softmax produces soft probabilities that allow gradient-based learning. As training progresses and $\alpha_t$ increases, these probabilities become increasingly binary, eventually forcing the network to select a single filter at each location.

The training procedure begins by passing clean image patches through the CFA pattern to generate noiseless measurements. Signal-dependent Poisson noise is then approximated using Gaussian noise with variance matching the signal intensity. We use the negative mutual information as our loss function, optimized using the AdamW optimizer (learning rate = $10^{-4}$, $\beta_1 = 0.9$, $\beta_2 = 0.999$) until convergence. Training took ~3 hours and ~1 GB on a Nvidia RTX A6000 GPU.

To evaluate IDEAL's ability to optimize a CFA, we computed the mutual information on the test set for the optimized pattern, as well as initial and intermediate patterns, using the estimation procedure described previously (See **Color Filter Array**). Reconstruction networks were subsequently trained for these three filters and evaluated following the protocol detailed above (See **Color Filter Array**).

### S6.7 1D simulations

We studied a simplified 1D imaging system with three key components: objects, encoders (point spread functions), and measurements. Objects were represented as periodic signals over a fixed spatial domain of 512 samples. Encoders were constrained to be bandlimited, non-negative, and infinitely periodic point spread functions that acted on objects through convolution. To ensure bandlimiting and enable stable optimization, encoders were parameterized in the Fourier domain using amplitude and phase components up to a specified bandwidth. The encoding process generated signals by convolving objects with these point spread functions. These signals were then integrated over fixed intervals ("pixels") to simulate detector sampling, with independent additive Gaussian noise applied to each pixel to model measurement noise.

Information content was estimated using our PixelCNN-based estimator, requiring 1D signals to be reshaped into 2D arrays. Training datasets were generated by encoding random object instances drawn from specified distributions. To characterize encoder constraints, we mapped the space of achievable signals by optimizing encoders to match target signals, defined as points in the space of possible pixel measurements. This optimization minimized the L2 distance between encoded and target signals. We analyzed how information capacity varied with key system parameters. Signal-to-noise ratio effects were studied by varying noise standard deviation while keeping signal energy fixed. Bandwidth impact was assessed by adjusting the number of nonzero Fourier components in the encoder. Sampling density analysis involved changing the number of pixels while maintaining fixed bandwidth. These relationships were examined across different object distributions including single delta functions, multiple delta functions, and bandlimited noise patterns.

Encoder optimization employed stochastic gradient ascent on a Gaussian-approximated mutual information objective, with a proximal operator enforcing non-negativity and energy constraints on the point spread functions. All simulations were implemented in JAX for automatic differentiation and GPU acceleration.

