# OpenReview forum: "Information-Driven Design of Imaging Systems"
_NeurIPS.cc/2025/Conference — NeurIPS 2025 poster_

### Official Review · Reviewer_NSW5 · 2025-06-10

**Clarity:** 2
**Significance:** 4
**Originality:** 3
**Rating:** 5
**Confidence:** 5

**Summary:**

This paper introduces a data-driven framework for estimating mutual information in imaging systems using only measurement data and a noise model, without requiring ground truth or object assumptions. The method fits probabilistic models to estimate entropy and information content, enabling objective evaluation across different imaging modalities. It is validated on applications like color photography, black hole imaging, and microscopy, showing strong correlation with task performance. The authors also propose IDEAL, an optimization pipeline that designs imaging systems by maximizing information capture through differentiable models. This work provides a general tool for evaluating and optimizing imaging systems based on information theory.

**Questions:**

1. How sensitive is the proposed estimator to model mismatch or data scarcity? The paper uses parametric models (e.g., Gaussian, PixelCNN) to estimate entropy, but does not analyze the sensitivity of the mutual information estimates under model mismatch (e.g., non-Gaussian data fit with Gaussian models) or limited data regimes. Could the authors provide quantitative or empirical analysis (e.g., bias/variance, convergence with sample size) to assess estimator robustness? Demonstrating estimator reliability under varying conditions would strengthen confidence in practical applicability and improve the overall evaluation.

2. Can the authors report the runtime and memory cost of different estimators, especially within IDEAL? PixelCNN is noted as the most accurate but computationally intensive option. However, the paper lacks detailed comparisons of training time, memory usage, and scalability across models. This is especially important for real-world deployment and adoption of IDEAL. A clear runtime analysis would validate the claim of practicality and support the scalability of the method in larger design spaces.


3. How does the method perform when mutual information is not aligned with task relevance? The authors acknowledge that total mutual information may not fully predict performance in tasks like classification (e.g., microscopy application). Could the authors expand on potential extensions to task-aware information metrics or suggest how IDEAL could incorporate task-specific weighting? Outlining a concrete path toward task-adaptive extensions would improve the method’s flexibility and broaden its relevance.

4. Can IDEAL be compared with task-specific or end-to-end design pipelines? While IDEAL is promising, its comparison is limited to internal metrics (e.g., mutual information or MSE). Can the authors compare IDEAL-based encoder designs with those from task-specific or end-to-end learned systems in terms of downstream task accuracy and efficiency? Empirical evidence of IDEAL's competitive or superior performance versus baselines would significantly enhance the strength of the design contribution.

5. Could you clarify assumptions behind the equivalence I(O;Y) = I(X;Y)? The estimator substitutes I(O;Y) with I(X;Y) under the assumption that X deterministically depends on O. Could the authors provide practical examples or guidance on when this assumption fails and how the method might adapt (e.g., when the encoder is stochastic)? Better explanation of this core assumption would improve theoretical clarity and help readers understand the estimator’s limits and adaptation potential.

**Ethical Concerns:**

["NO or VERY MINOR ethics concerns only"]

**Final Justification:**

Although the authors’ response has addressed some of my concerns, I feel that the rebuttal lacks sufficient depth, particularly in the answer to Question 5, which is not convincing to me. That said, I do appreciate this work, despite some flaws.

**Limitations:**

The authors address key limitations, including the gap between total and task-specific information, and the computational cost of expressive models. To strengthen this section, it would help to include runtime comparisons, clarify estimator reliability under limited data, and discuss cases where the I(O;Y) = I(X;Y) assumption may fail.

**Paper Formatting Concerns:**

N.A.

**Quality:**

3

**Strengths And Weaknesses:**

### **Strengths**

This paper introduces a practical, well-founded method to estimate mutual information in imaging systems without ground truth, and validates it across diverse applications. The proposed IDEAL framework adds a novel, differentiable way to optimize system design. Overall, the work is clear, technically sound, and broadly impactful for imaging system evaluation and optimization.


### **Weaknesses**

1. While the method relies on parametric entropy estimation, the paper does not quantify the estimator’s bias or variance across models, particularly in high-dimensional or low-sample regimes. This leaves some uncertainty about reliability in practice.

2. The PixelCNN-based estimator is significantly more accurate but also more computationally expensive (\~100s training time). However, detailed runtime or memory comparisons between models (especially within IDEAL) are not provided, making it hard to assess practical scalability.

3. Sections on model implementation (e.g., entropy estimation using parametric fitting) could benefit from more concrete examples or pseudocode. Additionally, the notation between I(O;Y) and I(X;Y) may be confusing to readers unfamiliar with the assumptions.

---

> ### Author Rebuttal · Authors · 2025-07-30
>
> Thank you for your thorough review and thoughtful questions. It's clear you've deeply understood our work, and your insights have helped us refine both our thinking and presentation.
>
> # W1/Q1: estimator bias/variance
>
> > While the method relies on parametric entropy estimation, the paper does not quantify the estimator’s bias or variance across models, particularly in high-dimensional or low-sample regimes. This leaves some uncertainty about reliability in practice.
>
> > How sensitive is the proposed estimator to model mismatch or data scarcity? The paper uses parametric models (e.g., Gaussian, PixelCNN) to estimate entropy, but does not analyze the sensitivity of the mutual information estimates under model mismatch (e.g., non-Gaussian data fit with Gaussian models) or limited data regimes. Could the authors provide quantitative or empirical analysis (e.g., bias/variance, convergence with sample size) to assess estimator robustness? Demonstrating estimator reliability under varying conditions would strengthen confidence in practical applicability and improve the overall evaluation.
>
> Figure S16 in the supplement analyzes estimator consistency and the estimator performance with dataset size.
>
> **Convergence:**
>  All three estimators (stationary Gaussian, full Gaussian, PixelCNN) converge to true values when tested on known distributions, with performance depending on how well data matches model assumptions.
>
> **Model mismatch:** On real data, Gaussian models converge to higher estimates than PixelCNN because they cannot accurately fit complex data distributions, demonstrating predictable bias patterns under model mismatch.
>
> **Data scarcity:** PixelCNN performs well across scenarios but is outperformed by simpler models on small datasets.
>
> We welcome suggestions on expanding this analysis to better serve practitioners.
>
>
> # W2/Q2: runtime/memory comparisons
>
> > The PixelCNN-based estimator is significantly more accurate but also more computationally expensive (~100s training time). However, detailed runtime or memory comparisons between models (especially within IDEAL) are not provided, making it hard to assess practical scalability.
>
> > Can the authors report the runtime and memory cost of differbent estimators, especially within IDEAL? PixelCNN is noted as the most accurate but computationally intensive option. However, the paper lacks detailed comparisons of training time, memory usage, and scalability across models. This is especially important for real-world deployment and adoption of IDEAL. A clear runtime analysis would validate the claim of practicality and support the scalability of the method in larger design spaces.
>
>
> We agree that these runtime and memory comparisons are important. The supplement contains some of this information, and we will expand it to provide the comprehensive analysis suggested.
>
>
> # W3: model implementation examples/pseudocode
>
> > Sections on model implementation (e.g., entropy estimation using parametric fitting) could benefit from more concrete examples or pseudocode. Additionally, the notation between I(O;Y) and I(X;Y) may be confusing to readers unfamiliar with the assumptions.
>
>
> We will link to our Python implementation to provide working code examples. We welcome
> specific suggestions about which aspects would benefit most from additional clarification.
>
>
> # Q3: Extensions to task-specific information
>
> > How does the method perform when mutual information is not aligned with task relevance? The authors acknowledge that total mutual information may not fully predict performance in tasks like classification (e.g., microscopy application). Could the authors expand on potential extensions to task-aware information metrics or suggest how IDEAL could incorporate task-specific weighting? Outlining a concrete path toward task-adaptive extensions would improve the method's flexibility and broaden its relevance.
>
>
> In its current form, our method's performance with specialized tasks depends on the overlap between task-specific and total information, which likely varies across applications. However, we envision several promising pathways to extend our framework for task-specific optimization.
>
> The difference between Figure S22 and Figure 2c demonstrates how maximizing total information may be suboptimal for specialized tasks: for a reconstruction task (Fig 2c) where all information aids performance, mutual information correlates strongly with error. For a classification task where only ~1\% of the total information is task-relevant, mutual information correlates less strongly with accuracy.
>
> There are multiple promising avenues to extend information estimation to task specificity in future work:
>
> In its current form, IDEAL-optimized systems could serve as pre-trained starting points for task-specific refinement with end-to-end optimization, analogous to how LLMs are pre-trained as a first step. This approach could reduce labeled data requirements for task-specific imagers.
>
>
> We can also directly extend our mathematical framework to estimate task-specific information. This requires estimating $I(\mathbf{Y};\mathbf{T})$, where $\mathbf{T}$ represents task variables like classification labels or segmentation masks. The quantity decomposes as $I(\mathbf{Y};\mathbf{T}) = H(\mathbf{Y}) - H(\mathbf{Y}|\mathbf{T})$. We could estimate $H(\mathbf{Y})$ with our existing approach and $H(\mathbf{Y}|\mathbf{T})$ by conditioning our models on task information. This requires only modifying our models to take both measurements and task labels as input.
>
> Since total information naturally decomposes into task-specific and task-irrelevant components: $I(\mathbf{Y};\mathbf{O}) = I(\mathbf{Y};\mathbf{T}) + I(\mathbf{Y};\mathbf{O}|\mathbf{T})$, we can design systems with weighted objectives $\alpha I(\mathbf{Y};\mathbf{T}) + \beta I(\mathbf{Y};\mathbf{O}|\mathbf{T})$. This could be used for applications such as:
>
> **Balanced systems** ($\alpha,\beta > 0$):} Control the relative importance of task-specific versus general information, potentially enabling systems that balance current task performance with future flexibility.
>
> **Compressed sensing** ($\alpha > 0, \beta < 0$):} Maximize task-relevant information while suppressing task-irrelevant information, which could be valuable for bandwidth-limited applications.
>
> **Privacy-preserving** ($\alpha < 0, \beta > 0$):} Suppress task-specific information while preserving general scene information for other analyses.
>
> # Q4: IDEAL vs End-to-End
>
> > Can IDEAL be compared with task-specific or end-to-end design pipelines? While IDEAL is promising, its comparison is limited to internal metrics (e.g., mutual information or MSE). Can the authors compare IDEAL-based encoder designs with those from task-specific or end-to-end learned systems in terms of downstream task accuracy and efficiency? Empirical evidence of IDEAL's competitive or superior performance versus baselines would significantly enhance the strength of the design contribution.
>
> This comparison already exists in our current results: IDEAL achieves comparable performance to end-to-end optimization methods (compare blue points in Figures 2a and 3c) while using less memory and compute. We acknowledge that this comparison is presented confusingly across multiple figures and will clarify the presentation in our revision.
>
> Follow-on work has systematically investigated this across a wider variety of systems, providing empirical evidence of IDEAL's competitive or superior performance versus end-to-end methods, with results included as an anonymous PDF in the supplementary materials.
>
>
>
> # Q5: $I(\mathbf{O};\mathbf{Y}) = I(\mathbf{X};\mathbf{Y})$ assumptions
>
> > Could you clarify assumptions behind the equivalence I(O;Y) = I(X;Y)? The estimator substitutes I(O;Y) with I(X;Y) under the assumption that X deterministically depends on O. Could the authors provide practical examples or guidance on when this assumption fails and how the method might adapt (e.g., when the encoder is stochastic)? Better explanation of this core assumption would improve theoretical clarity and help readers understand the estimator’s limits and adaptation potential.
>
>
> This assumption is key to our framework. We want to make it as clear as possible and welcome suggestions on whether more of this discussion should be moved from the supplement to main text.
>
> As described in Section S1.5.1, the equivalence $I(\mathbf{O};\mathbf{Y}) = I(\mathbf{X};\mathbf{Y})$ holds when the encoder function is deterministic—noiseless images $\mathbf{X}$ depend deterministically on objects $\mathbf{O}$. This enables practical information estimation without access to the objects themselves.
>
> The assumption fails when encoders vary between different objects. Examples include changing dust patterns across measurements, illumination fluctuations that differ for each object, or mechanical drift in focus across objects. Fixed system imperfections (e.g. constant dust patterns or defocus) do not violate this assumption because the encoder remains deterministic.
>
> When encoder variations occur, measurements contain information about both objects and the system itself, leading to overestimated object information. Section S4.1 demonstrates this effect experimentally by estimating information of measurements without objects.
>
> We describe how this could theoretically be addressed by estimating $H(\mathbf{Y}|\mathbf{E})$ instead of $H(\mathbf{Y})$, where $\mathbf{E}$ represents encoder state in Section S2.7.

---

> > ### Comment · Reviewer_NSW5 · 2025-08-06
> >
> > Although the authors’ response has addressed some of my concerns, I feel that the rebuttal lacks sufficient depth, particularly in the answer to Question 5, which is not convincing to me. That said, I do appreciate this work, despite some flaws.

---

> ### Author Response · Authors · 2025-08-08
> **Q5: additional detail**
>
> We appreciate the reviewer's constructive feedback. Your question prompted us to formalize the deterministic encoder assumption more rigorously, which clarifies both the scope of our current method and promising future extensions.
>
> Below, we prove that $I(\mathbf{O};\mathbf{Y}) = I(\mathbf{X};\mathbf{Y})$ for deterministic encoders (Theorem 1). In the following comment, we show how to adapt our framework when this assumption fails due to stochastic encoder variations. While this doesn't affect our main results (which, like existing methods, assume deterministic encoders), it enables a promising future extensions like quantifying performance degradation from real-world encoder variations in both IDEAL and field measurements.
>
>
> **Theorem 1: Equivalence of $I(\mathbf{O};\mathbf{Y})$ and $I(\mathbf{X};\mathbf{Y})$ for deterministic encoders**
>
>   For a Markov chain $\mathbf{O} \to \mathbf{X} \to \mathbf{Y}$, if $\mathbf{X} = f(\mathbf{O})$ for some deterministic
>   function $f$, then $I(\mathbf{O};\mathbf{Y}) = I(\mathbf{X};\mathbf{Y})$.
>
>
> **Proof:**
>
>   Since $I(\mathbf{O};\mathbf{Y}) = H(\mathbf{Y}) - H(\mathbf{Y}|\mathbf{O})$ and $I(\mathbf{X};\mathbf{Y}) = H(\mathbf{Y}) -
>    H(\mathbf{Y}|\mathbf{X})$, we need to show $H(\mathbf{Y}|\mathbf{O}) = H(\mathbf{Y}|\mathbf{X})$.
>
>   From the deterministic condition $\mathbf{X} = f(\mathbf{O})$: Since $\mathbf{X} = f(\mathbf{O})$, knowing $\mathbf{O}$ completely determines $\mathbf{X}$. Therefore, conditioning on $\mathbf{O}$ is equivalent to conditioning on both $\mathbf{O}$ and $\mathbf{X}$:
>   $$H(\mathbf{Y}|\mathbf{O}) = H(\mathbf{Y}|\mathbf{O},\mathbf{X})$$
>
>   From the Markov property $\mathbf{O} \to \mathbf{X} \to \mathbf{Y}$:
>   $$H(\mathbf{Y}|\mathbf{O},\mathbf{X}) = H(\mathbf{Y}|\mathbf{X})$$
>
>   Therefore:
>   $$H(\mathbf{Y}|\mathbf{O}) = H(\mathbf{Y}|\mathbf{X}) \quad \square$$

---

> ### Author Response · Authors · 2025-08-08
> **Stochastic encoders**
>
> As mentioned previously, our deterministic encoder assumption fails when system components vary between measurements (e.g., focus drift).
>
> Stochastic encoders affect information estimates two ways. First, different encoder states preserve different amounts of object information. For example, defocus destroys sharp features in images. Average object information thus depends on the focus distribution.
>
> Second, encoder variations create measurement diversity beyond object variations. With focus drift, measurements vary in both content and sharpness. Without modification, our method would interpret both as object information, thereby upwardly biasing its estimates.
>
> One potential way to adapt our method is using calibration data to identify encoder-induced variations. This would likely require either additional compute or simplifying assumptions to handle the fact the the measurement diversity induced by encoder variations may be object dependent.
>
>
> ### Mathematical formulation
>
> We model encoder variations with state variable $\mathbf{E}$:
>
> $$(\mathbf{O}, \mathbf{E}) \to \mathbf{X} \to \mathbf{Y}$$
>
> Now $\mathbf{X} = f(\mathbf{O}, \mathbf{E})$ for some deterministic function $f$. For example, for focus drift, $f$ would describe how defocus $\mathbf{E}$ blurs object $\mathbf{O}$ to produce image $\mathbf{X}$.
>
> The quantity of interest in the original deterministic setting is now written with the encoder explicit $I(\mathbf{O};\mathbf{Y}|\mathbf{e})$.
>
> In the stochastic case, averaging over all encoder states gives:
>
> $$\bar{I}(\mathbf{O};\mathbf{Y}) = \sum_{\mathbf{e}} p(\mathbf{e}) I(\mathbf{O};\mathbf{Y}|\mathbf{e})$$
>
> Where $\bar{I}(\mathbf{O};\mathbf{Y})$ denotes the average object information across all encoder states.
>
> Extending Theorem 1 to $\mathbf{X} = f(\mathbf{O}, \mathbf{E})$:
>
> $$I(\mathbf{X};\mathbf{Y}) = I(\mathbf{O}, \mathbf{E}; \mathbf{Y})$$
>
> Rewriting using the chain rule of mutual information:
> $$I(\mathbf{X};\mathbf{Y}) = \bar{I}(\mathbf{O}; \mathbf{Y}) + I(\mathbf{E}; \mathbf{Y} | \mathbf{O})$$
>
> The first term captures effect one (average object information across encoder states); the second captures effect two (diversity from encoder variations).
>
> **Example: random focal drift**
>
> Consider a microscope with mechanical drift causing focus to vary between perfect focus and various degrees of defocus. Compared to a stable, in-focus system, we have $\bar{I}(\mathbf{O};\mathbf{Y}) < I(\mathbf{O};\mathbf{Y}|\mathbf{e}_{\text{in-focus}})$ because defocus destroys high-frequency information. Meanwhile, $I(\mathbf{E};\mathbf{Y} \mid \mathbf{O}) > 0$ because focus variations add to measurement diversity.
>
> As shown in theorem 1, a deterministic in-focus system has $I(\mathbf{X};\mathbf{Y}) = I(\mathbf{O};\mathbf{Y}|\mathbf{e}_{\text{in-focus}})$. With stochastic focus variations, $I(\mathbf{X};\mathbf{Y})$ can increase or decrease relative to this baseline depending on which of the two effects dominates.
>
>
>
> **Estimating information with stochastic encoders**
>
> When encoders vary stochastically, our method estimates $I(\mathbf{X};\mathbf{Y}) = \bar{I}(\mathbf{O};\mathbf{Y}) +
> I(\mathbf{E};\mathbf{Y}|\mathbf{O})$. To isolate object information $\bar{I}(\mathbf{O};\mathbf{Y})$, we must estimate and subtract the
> encoder contribution.
>
> The central challenge is that $I(\mathbf{E};\mathbf{Y}|\mathbf{O})$ depends on the object—different objects may reveal encoder
> variations differently. We propose two approaches:
>
> Approach 1: Fix object, vary encoder states, estimate $I(\mathbf{E};\mathbf{Y}|\mathbf{O}=\mathbf{o})$ for multiple objects. This would be feasible in IDEAL using increased compute, but in experimental systems only possible if we can fix objects while varying encoder states (e.g., imaging a static sample while inducing focus drift).
>
>
> Approach 2: Assume $I(\mathbf{E};\mathbf{Y}|\mathbf{O}) \approx
>   I(\mathbf{E};\mathbf{Y})$. Computationally and experimentally simpler, but may give incorrect results if encoder effects are highly object-dependent.
>
>   Taking Approach 2 as an example:
>   $$I(\mathbf{E};\mathbf{Y}) = H(\mathbf{Y}) - H(\mathbf{Y}|\mathbf{E})$$
>
>   Our method already estimates $H(\mathbf{Y})$. To estimate $H(\mathbf{Y}|\mathbf{E})$, we fit conditional models
>   $p_{\theta}(\mathbf{y}|\mathbf{e})$ instead of $p_{\theta}(\mathbf{y})$, where $\mathbf{e}$ represents encoder state information. For focus drift, conditioning on timestamps would help predict measurements since focus varies smoothly with time.
>
> For Approach 1, we would additionally need to take an outer expectation over objects. In IDEAL, since we know the true object, the models could be modified to condition on it directly. In experimental conditions, we would likely need static objects to take multiple measurements of.
>
> By estimating both terms we can recover the quantity of interest, object information in the measurement.
>
>  $$\bar{I}(\mathbf{O};\mathbf{Y}) = I(\mathbf{X};\mathbf{Y}) - I(\mathbf{E};\mathbf{Y}|\mathbf{O})$$

---

### Official Review · Reviewer_Sgt1 · 2025-07-02

**Clarity:** 2
**Significance:** 3
**Originality:** 4
**Rating:** 4
**Confidence:** 4

**Summary:**

This paper introduces an information-driven design framework for imaging systems, focusing on mutual information estimation between unknown objects and noisy measurements. It proposes a data-driven method to quantify the information content of measurements, and combines this information estimation approach with gradient-based optimization to design imaging systems that maximize information capture.

**Questions:**

Questions for the authors to address in the rebuttal:
1.	As mentioned in weakness, how does the proposed method address the challenge of continuously evolving measurements during imaging system optimization, especially without real-time adjustable hardware for data collection, which may still require ground truth data for training?
2.	Does the generalization of maximizing information content sacrifice task-specific accuracy, and can the method be experimentally validated against previous end-to-end optimization approaches involving decoders?
3.	How does the proposed method handle cases where the noise model may not be fully characterized or differs from the assumed model , such as those complex noise models in thermal infrared imaging.
Addressing these questions in the rebuttal would greatly enhance the clarity and applicability of the proposed method, thereby improving the paper's evaluation.

**Ethical Concerns:**

["NO or VERY MINOR ethics concerns only"]

**Final Justification:**

I have always recognized the originality of this article, and the response satisfactorily addressed my doubts; therefore, I raised my vote.

**Limitations:**

The method assumes static measurements, but in real-world imaging system optimization, measurements continuously evolve with changing imaging parameters. Without real-time adjustable hardware to collect new data, the training process may still rely on ground truth data to simulate new measurements. The authors should clarify how this challenge is addressed.
While the approach avoids post-processing decoders, it remains unclear whether this generalization sacrifices task-specific accuracy. The authors should provide experimental validation comparing this method with previous end-to-end optimization approaches involving decoders.
The noise model assumed in the paper may not fully account for complex noise scenarios, such as those encountered in thermal infrared imaging. The authors should explain how their method handles such cases, where noise characteristics may differ significantly from the assumed models.

**Paper Formatting Concerns:**

n/a.

**Quality:**

4

**Strengths And Weaknesses:**

Strengths.
+ The interesting method effectively bridges a critical gap in imaging system design, by quantifying information content directly from measurements without needing post-processing decoders.
+ The paper demonstrates the applicability of the approach in a wide range of imaging systems, including real-world use cases like black hole imaging and biological microscopy.
+ The theoretical foundation of the paper is robust, offering a clear mathematical formulation for mutual information and its estimation.
+ The IDEAL method for optimizing encoder designs based on information content shows great promise, especially in terms of reducing computational overhead by focusing on the encoder alone before considering the decoder.

Weakness.
Nevertheless, I believe the authors still need to address the following critical concerns.
- The authors claim that the information content can be quantified directly from a dataset of measurements without requiring ground truth data. However, in the process of optimizing imaging systems, the imaging parameters are constantly changing, which implies that the measurements are also continuously evolving. Without adjustable imaging hardware to collect new training data in real-time (which is relatively difficult to achieve), this inevitably requires the training process to rely on ground truth data to simulate new measurements.
- Furthermore, although the method does not require post-processing decoders, it raises the question of whether this generalization compromises the accuracy for specific tasks. It needs to be experimentally demonstrated whether maximizing information content ultimately leads to the optimal task performance. For instance, in an imaging modality experiment, a comparison between this method and previous end-to-end optimization approaches involving decoders should be provided to show the effectiveness.

---

> ### Author Rebuttal · Authors · 2025-07-29
>
> Thank you for your thorough review and thoughtful questions. We appreciate the time you invested in engaging with our work. Your insights have helped us clarify both our presentation and our thinking about future directions for information-driven design.
>
>
>
> # W1/Q1: changing measurements and ground truth data
>
>
> > The method assumes static measurements, but in real-world imaging system optimization, measurements continuously evolve with changing imaging parameters. Without real-time adjustable hardware to collect new data, the training process may still rely on ground truth data to simulate new measurements. The authors should clarify how this challenge is addressed.
>
> > The authors claim that the information content can be quantified directly from a dataset of measurements without requiring ground truth data. However, in the process of optimizing imaging systems, the imaging parameters are constantly changing, which implies that the measurements are also continuously evolving. Without adjustable imaging hardware to collect new training data in real-time (which is relatively difficult to achieve), this inevitably requires the training process to rely on ground truth data to simulate new measurements.
>
> > How does the proposed method address the challenge of continuously evolving measurements during imaging system optimization, especially without real-time adjustable hardware for data collection, which may still require ground truth data for training?
>
>
> This question highlights an important distinction about deployment scenarios for information-driven design. Consider three key scenarios with different data availability and optimization constraints:
>
> **1) Ground truth-free system comparison (field measurements):** Our method can estimate information content directly from real-world measurements without ground truth data. This enables quantitative performance evaluation in real deployment conditions that were previously inaccessible to systematic analysis. Figure 2d exemplifies this: we can quantitatively rank microscopy illumination patterns without knowing the molecular structure of the cells being imaged. Similarly, a self-driving car company could compare three prototype cameras by collecting field images and comparing encoded information from each camera.
>
> **2) Pre-fabrication design (simulation):** This scenario requires ground truth data and system simulation to optimize designs before fabrication using IDEAL. While end-to-end optimization methods can achieve similar goals, IDEAL offers advantages in this simulated design scenario: it optimizes only the encoder rather than the full pipeline, significantly reducing computational and memory requirements during the design process.
>
> **3) Adaptive optimization (real-time):** Using information estimates to optimize system parameters in real-time represents largely unexplored territory with both significant challenges and significant potential. This scenario faces fundamental challenges the reviewer correctly identifies: datasets may contain heterogeneous combinations of objects and parameter settings, and during real-time operation, distinguishing whether measurement changes arise from object variation or parameter adjustments requires additional constraints such as assuming objects remain stationary. Despite these challenges, information offers potential advantages as a universal optimization signal that doesn't require algorithm-specific objective functions. This could enable simultaneous optimization of multiple interacting parameters—capabilities difficult to achieve with conventional approaches. We plan to explore this direction in future work.
>
> # W2/Q2: Specialized tasks and end-to-end
>
> > Does the generalization of maximizing information content sacrifice task-specific accuracy, and can the method be experimentally validated against previous end-to-end optimization approaches involving decoders?
>
> Yes, we have validated IDEAL against end-to-end methods—it achieves comparable performance while using less memory and compute. Our general information-maximization approach achieves optimal performance for image reconstruction tasks, which are ubiquitous across imaging applications. Specialized tasks would require a natural extension of our framework that we outline below.
>
> **IDEAL vs end-to-end:** IDEAL matches end-to-end optimization performance as shown in Figures 2a and 3c (compare blue points). We acknowledge this comparison is presented confusingly across multiple figures and will clarify it in our revision. Follow-on work has compared IDEAL and end-to-end across additional systems, providing additional evidence of IDEAL's competitive or superior performance (see anonymized PDF in supplementary materials).
>
>
> **General task performance:** For reconstruction tasks, maximizing information achieves optimal performance because reconstruction requires preserving all object details. Figures 2a-c and S20 demonstrate tight correlations between information estimates and reconstruction quality across all tested systems, aligning with theoretical results [`1`] that establish this fundamental relationship (further discussed in Section S5.1.4).
>
> However, information maximization offers unique advantages over direct reconstruction metrics like mean squared error: computational efficiency in simulated design and ground-truth-free evaluation for field measurements.
>
> Reconstruction is ubiquitous across imaging applications—from astronomy to consumer photography to medical imaging—because
> capturing maximum detail is often the primary goal. Accordingly, most existing end-to-end approaches also optimize for reconstruction performance. Thus, our general information maximization approach is already broadly applicable without modification.
>
>
> **Specialized task performance:** For specialized tasks, maximizing total information may be suboptimal. Section 4.1 and Figure S22 demonstrate this: classification tasks using only ~1\% of total information show weaker correlation between information and performance compared to reconstruction tasks.
>
> Our framework naturally extends to these cases by decomposing information into task-specific and task-irrelevant components:
> $I(\mathbf{Y};\mathbf{O}) = I(\mathbf{Y};\mathbf{T}) + I(\mathbf{Y};\mathbf{O}|\mathbf{T})$. This enables weighted objectives for
> various applications:
>
> - Balancing task-specific and general information for flexible systems
> - Maximizing task-relevant while suppressing irrelevant information for bandwidth-limited scenarios
> - Preserving general information while suppressing task-specific information for privacy
>
> We provide additional mathematical and implementation detail in Section S1.5.2 and in our response to Reviewer NSW5 below.
>
> `1` *The Interplay Between Information and Estimation Measures* (2013)
>
> # Q3: Noise model mismatch
>
> > The noise model assumed in the paper may not fully account for complex noise scenarios, such as those encountered in thermal infrared imaging. The authors should explain how their method handles such cases, where noise characteristics may differ significantly from the assumed models.
>
> > How does the proposed method handle cases where the noise model may not be fully characterized or differs from the assumed model, such as those complex noise models in thermal infrared imaging?
>
> You've identified an important consideration that needs clearer explanation in our revision.
>
> Our method remains robust to noise model mismatch in most practical scenarios. When accurate noise modeling does become essential, this requirement is not unique to our approach.
>
> Imaging noise typically exhibits one of two properties that provide robustness to noise model mismatch: (1) statistical independence from the object, or (2) conditional independence given the noiseless image. The additive Gaussian and Poisson models we tested exemplify these cases and provide a blueprint for other noise types, including those in thermal infrared imaging.
>
> When noise is object-independent (property 1), an incorrect noise model affects mutual information estimates by an additive constant, which means comparisons between encoders and IDEAL optimization gradients remain unchanged. The complex noise encountered in thermal imaging is primarily object-independent, for example thermal (Johnson) noise, dark current, fixed pattern noise, and 1/f noise [`1`, `2`, `3`].
>
> When noise is object-dependent (property 2), an incorrect noise model can affect relative encoder evaluations. However, object-dependent noise typically arises from well-understood
> quantum processes with simple mathematical structure, making estimation of its entropy reliable and tractable. Poisson shot noise is the most common example. In the supplement (S2.3.3), we demonstrate how its conditional independence given the noiseless signal allows its estimation to be reduced to a tractable 1D calculation.
>
> Accurate noise modeling becomes essential only when comparing across encoder designs that directly affect noise patterns, such as comparing thermal sensors with different fixed pattern noise. Even then, property 1 enables straightforward empirical characterization of the noise without having to consider the complexities of the objects being imaged.
>
> In this setting, the need for accurate noise models is not unique to our approach—it is commonplace for methods applied to applications sensitive to complex noise patterns. For example, low-light denoising methods require careful physics-based noise modeling to handle extreme conditions [4].
>
>
> `1` *Modeling Noise in Thermal Imaging Systems* (1993)
>
> `2` *Exploring Video Denoising in Thermal Infrared Imaging: Physics-Inspired Noise Generator, Dataset, and Model* (2024)
>
> `3` *Thermal Image Processing via Physics-Inspired Deep Networks* (2021)
>
> `4` *Dancing under the stars: video denoising in starlight* (2022)

---

> > ### Author Response · Authors · 2025-08-07
> >
> > Thank you again for your thorough review and thoughtful questions. We hope our response has addressed your concerns.
> >
> > Since the discussion period ends in ~36 hours, we wanted to check if you have any remaining questions or if there's anything we could clarify further. We're happy to provide any additional information.

---

> > > ### Comment · Reviewer_Sgt1 · 2025-08-09
> > > **response**
> > >
> > > I have always recognized the originality of this article, and the response satisfactorily addressed my doubts; therefore, I raised my vote.

---

### Official Review · Reviewer_HkSv · 2025-07-03

**Clarity:** 3
**Significance:** 2
**Originality:** 3
**Rating:** 4
**Confidence:** 3

**Summary:**

This paper introduces a data-driven framework for estimating mutual information in imaging systems, enabling information-theoretic design without ground truth data. The approach fits probabilistic models to noisy measurements and quantifies information content by subtracting noise entropy. It validates across diverse applications—color photography, radio astronomy, lensless imaging, and microscopy—showing that mutual information reliably predicts system performance. The Information-Driven Encoder Analysis Learning (IDEAL) framework optimizes imaging systems via gradient-based methods to maximize information capture, achieving up to 3.8× faster training with 97% performance retention compared to full-data training.

**Questions:**

1. The framework focuses on static images, but real-world imaging (e.g., video) involves temporal correlations unaddressed in the current model. How does the method handle dynamic scenes with time-varying noise?

2. The study lacks validation on multi-spectral data, where cross-channel dependencies may alter mutual information.
Can IDEAL optimize systems for real-time processing constraints?

3. The framework prioritizes information maximization but doesn’t address latency-accuracy trade-offs in real-time applications.
What is the impact of multi-modal imaging (e.g., RGB-NIR) on information estimation?

4. Most experiments use supervised decoders (e.g., neural networks), but unsupervised scenarios (e.g., self-denoising) remain untested. How does the method perform with unsupervised or self-supervised decoding?

**Ethical Concerns:**

["NO or VERY MINOR ethics concerns only"]

**Limitations:**

see weakness.

**Paper Formatting Concerns:**

-

**Quality:**

3

**Strengths And Weaknesses:**

Strengths：
1. No ground truth required: Estimates information directly from measurements, avoiding reliance on labeled data .
2. Broad applicability: Works across imaging modalities (photography, astronomy, microscopy) without task-specific assumptions .
3. Quantitative design metric: Unifies resolution, SNR, and sampling trade-offs into a single information-theoretic metric .
4. Gradient-based optimization: IDEAL enables automated system design (e.g., color filter arrays) with up to 3.8× training speedup .

Weaknesses：
1. Computational complexity: PixelCNN models require more training data and time compared to Gaussian models .
2. Noise model dependency: Relies on accurate noise characterization (e.g., Gaussian/Poisson), which may fail in complex noise scenarios .
3. Task generality trade-off: Total information may not align with task-specific needs (e.g., classification vs. reconstruction) .

---

> ### Author Rebuttal · Authors · 2025-07-29
>
> **Notice: This review appears to be AI-generated and has been reported to the Area Chair.**
>
> The review contains fabricated statistics not present in our paper (e.g. "3.8× faster training with 97% performance retention"), which don't make sense in the context of our work. Furthermore, the review demonstrates no substantive critical analysis, instead merely paraphrasing content from our manuscript without evaluation.

---

### Decision · Program_Chairs · 2025-09-17

**Decision:**

Accept (poster)

**Comment:**

The paper presents a framework for the data-driven design of imaging systems, with experiments on designing color sensors, optimizing the packing of radio telescopes for black hole imaging, and related applications. The framework is based on approximating the mutual information between the underlying signal and the captured data—a task that is particularly challenging in high-dimensional settings such as images. To address this, the authors develop Gaussian Process and neural network–based approximations, and demonstrate that the framework leads to improved design choices.

The paper was well received by the reviewers. While there are aspects that could be further refined, the AC concurs with the reviewers that the work makes solid contributions and recommends acceptance. The authors are encouraged to improve the presentation of the results and address the specific issues raised during the review process. The AC also notes the broader context of concerns around LLM-generated reviews, but emphasizes that this did not factor into the decision-making for this submission.